# Spatial variability in mass loss of glaciers in the Everest region, central Himalaya, between 2000 and 2015

Owen King[1], Duncan J. Quincey[1], Jonathan L. Carrivick[1], Ann V. Rowan[2]

[1]School of Geography and water@leeds, University of Leeds, Leeds, LS2 9JT, UK.

[2]Department of Geography, University of Sheffield, Sheffield, S10 2TN, UK.

*Correspondence to*: gy08ok@leeds.ac.uk

**Abstract.** Region wide averaging of Himalayan glacier mass change has masked any catchment or glacier scale variability in glacier recession, thus the role of a number of glaciological processes in glacier wastage remains poorly understood. In this study, we quantify mass loss rates over the period 2000–2015 for 32 glaciers across the Everest region and assess how future ice loss is likely to differ depending on glacier hypsometry. The mean mass balance of all 32 glaciers in our sample was –0.52 ± 0.22 m water equivalent (w.e.) a$^{-1}$. The mean mass balance of 9 lacustrine terminating glaciers (–0.70 ± 0.26 m w.e. a$^{-1}$) was 32 % more negative than land-terminating, debris covered glaciers (–0.53 ± 0.21 m w.e. a$^{-1}$). The mass balance of lacustrine-terminating glaciers is highly variable (–0.45 ± 0.13 m w.e. a$^{-1}$ to –0.91 ± 0.22 m w.e. a$^{-1}$), perhaps reflecting glacial lakes at different stages of development. To assess the importance of hypsometry on glacier response to future climate warming, we calculated current (Dudh Koshi- 0.37, Tama Koshi- 0.36, Tibetan Plateau- 0.40) and prospective future glacier Accumulation Area Ratios (AARs). IPCC AR5 RCP 4.5 warming (0.9-2.3 $^{o}$C by 2100) could reduce AARs to 0.25 or 0.03 in the Tama Koshi catchment, 0.26 or 0.18 in the Dudh Koshi catchment, and 0.30 or 0.17 on the Tibetan Plateau. Our results are important because they suggest glacial lake expansion across the Himalaya could expedite ice mass loss and the prediction of future contributions of glacial meltwater to river flow will be complicated by spatially-variable glacier responses to climate change.

Keywords: Himalaya, glacier mass, debris-cover, Everest region, glacial lakes

## 1.    Introduction

Estimates of Himalayan glacier ice volume range from 2,300 km$^3$ to 7,200 km$^3$ (Frey et al., 2014 and references within) distributed amongst more than 54,000 glaciers across the Hindu Kush, Himalaya and the Karakoram (Bajracharya et al., 2015). The current mass balance of Himalayan glaciers is predominantly negative, with accelerating mass loss having been observed over the past few decades (Bolch et al., 2012; Thakuri et al., 2014). This mass loss is occurring because of a combination of processes. Shrestha et al. (1999) show a rise in the mean annual air temperature of 0.057 °C a$^{-1}$ across the Himalaya between 1971 and 1994.

Bollasina et al. (2011) show a reduction in total precipitation (–0.95 mm day$^{-1}$) amounting to 9 to 11% of total monsoon rainfall over a broad area of northern India between 1950 and 1999. Bhutiyana et al. (2010) show both decreasing total precipitation and a changing precipitation phase, with a lower proportion of precipitation falling as snow across the northwest Himalaya between 1996 and 2005. The snow cover season has been shortening as

a result (Pepin et al., 2015). Under different climate scenarios, glacier imbalance in the region may contribute 8.7–17.6 mm of sea level rise by 2100 (Huss and Hock, 2015). Prolonged mass loss from Himalayan glaciers may cause diminishing discharge of the largest river systems originating in the region (Immerzeel et al., 2010; Lutz et al., 2014), thereby impacting on Asian water resources in the long-term.

Recent studies have identified spatial heterogeneity in mass loss across the Himalaya in the first decade of the

21$^{st}$ century (Kääb et al., 2012; Gardelle et al., 2013; Kääb et al., 2015). Glaciers in the Spiti Lahaul and Hindu Kush are losing mass most quickly (Kääb et al., 2015). Glaciers in the central Himalaya appear to have less negative mass balances (Gardelle et al., 2013). The anomalous balanced, or even slightly positive, glacier mass budget in the Karakoram is well documented (Bolch et al., 2012; Gardelle et al., 2012).  Few previous studies have assessed the variability of glacier mass loss within catchments (Pellicciotti et al., 2015). Nuimura et al.

(2012) examined the altitudinal distribution of glacier surface elevation change in the Khumbu region, Nepal, and found similar surface lowering rates over debris-free and debris-covered glacier surfaces. Gardelle et al. (2013) detected enhanced thinning rates on lacustrine terminating glaciers in Bhutan, West Nepal, and the Everest region, but did not make an explicit comparison with land terminating glacier recession rates. Similarly, Basnett et al. (2013) have shown that lacustrine terminating glaciers in Sikkim, Eatern Indian Himalaya

experienced greater area loss between ~1990 and 2010 compared to land terminating glaciers. Benn et al. (2012) have considered the role of glacial lakes in the wastage of debris-covered glaciers and proposed a conceptual model of Himalayan glacier recession that included important thresholds between regimes of ice dynamics and mass loss at different stages of lake development. Benn et al. (2012) suggest that an expansive, moraine dammed and potentially hazardous glacial lake may represent the end product of the wastage of a debris-

covered glacier.

We aim to quantify glacier mass loss rates in three major catchments of the central Himalaya and assess the glacier-scale variability of ice loss within and between catchments. We specifically examine the mass balance, hypsometry and total area change of each glacier and compare those terminating in a glacial lake with those terminating on land. We use these data together with climatic data from the region to define the major

mechanisms that may have driven mass loss in recent decades, and to assess scenarios of likely future ice loss from our sample of glaciers.

## 2.    Study area

We studied glaciers in three catchments of the Everest region (Figure 1), spanning both Nepal and Tibet (China). Two of the catchments, the Dudh Koshi and the Tama Koshi, are located in north-eastern Nepal and drain the southern flank of the Himalaya. The third catchment is located to the north of the main orographic divide, and the glaciers drain north into Tibet (China). Most glaciers in the studied catchments are characterised by long (10–15 km), low-slope angle, debris-covered tongues that are flanked by large (tens of metres high) moraine ridges (Hambrey et al., 2008). Some glaciers have accumulation areas several kilometres wide that reach extreme altitudes (up to 8000 m in the case of the Western Cwm of Khumbu Glacier). Others sit beneath steep hillslopes (e.g. Lhotse and the Lhotse face), are fed almost exclusively by avalanches and are less than 1 km in width for their entire length.

The largest 40 of 278 glaciers in the Dudh Koshi catchment account for 70% of the glacierised area (482 km$^2$-Bajracharya and Mool, 2009). These glaciers are all partially debris-covered, with debris mantles reaching at least several decimetres in thickness (Rounce and McKinney, 2014; Rowan et al., 2015). Here, the total area of glacier surface covered by debris has increased since the 1960s (Thakuri et al., 2014) and several previous studies have published surface lowering data for the catchment indicating accelerating surface lowering rates over recent decades (e.g. Bolch et al., 2011; Nuimura et al. 2012). We select nine of the largest glaciers (Supplementary table 1) for analysis given that they provide the greatest potential volume of meltwater to downstream areas.

There are a total of 80 glaciers in the Tama Koshi catchment covering a total area of 110 km$^2$ (Bajracharya et al., 2015). We again selected the largest nine glaciers (Supplementary table 1) for analysis based on relative potential contributions to river flow. The Tama Koshi is a poorly studied catchment, perhaps best known for the existence of Tsho Rolpa glacial lake, which underwent partial remediation during the 1990s (Reynolds, 1999).

The fourteen glaciers within our sample that flow onto the Tibetan Plateau (Supplementary table 1) all contribute meltwater to the Pumqu river catchment, which covers an area of 545 km$^2$ (Che et al., 2014). Debris cover is less prevalent on glaciers of the Pumqu catchment, and glacier recession has caused a 19 % of glacier area loss since 1970 (Jin et al., 2005; Che et al., 2014). There is relatively little information on glacier ELAs other than in the Dudh Koshi catchment. In the Dudh Koshi, Asahi et al. (2001) estimated ELAs to be at around 5600 m a.s.l. in the early 2000s. Wagnon et al. (2013) measured annually variable ELAs of 5430–5800 m a.s.l.

on the Mera and Polkalde glaciers between 2007 and 2012, Shea et al. (2015) estimate the current ELA to be 5500 m a.s.l., and Gardelle et al. (2013) estimated the ELA to be around 5840 m over the period 2000–2009. On the Tibetan Plateau, those in the Rongbuk catchment were estimated between 5800 and 6200 m a.s.l. for the period 1974–2006 (Ye et al., 2015).

A number of studies have identified an abundance of glacial water bodies in the Everest region. Salerno et al. (2012) identified 170 unconnected glacial lakes (4.28 km$^2$), 17 proglacial lakes (1.76 km$^2$) and 437 supraglacial lakes (1.39 km$^2$) in the Dudh Koshi catchment. Gardelle et al. (2011) identified 583 supraglacial ponds and lakes in an area comparable in coverage to Figure 1. Watson et al. (2016) mapped 9340 supraglacial ponds on 8 glaciers of the Dudh Koshi catchment and Rongbuk glacier in the Pumqu catchment. Watson et al. (2016) also

show a net increase in ponded area for six of their nine studied glaciers. Some of the largest glacial lakes in this region have also been expanding in recent decades (Sakai et al., 2000; Che et al., 2014; Somos-Valenzuela et al., 2014). This increased meltwater ponding at glacier termini has potential to affect ice dynamics and down-valley meltwater and sediment fluxes (Carrivick and Tweed, 2013) as well as causing a hazard to populations living downstream. Several of the lakes have burst through their moraine dams in previous decades causing rapid and

extensive flooding downstream; the best studied outburst floods are those from Nare glacier in 1977 (Buchroithner et al., 1982) and from Dig Tsho in 1985 (Vuichard and Zimmerman, 1987).

We classify nine glaciers from the sample as lacustrine terminating, where the glacier termini and glacial lakes are actively linked. We do not consider either Rongbuk Glacier or Gyabrag Glacier as lacustrine terminating. Gyabrag Glacier is now separated from a large proglacial lake by a large outwash plain and we do not believe

the lake can have an influence on the retreat of the glacier. In the case of Rongbuk Glacier, the lake is supraglacial and far up-glacier from its terminal region and thus does not currently influence the recession of the terminus of the glacier. The expanding Spillway Lake at the terminus of Ngozumpa Glacier (Thompson et al., 2012) is currently of limited depth, and is unlikely to affect glacier dynamics in its current state so we also exclude Ngozumpa Glacier from the lacustrine terminating category.

3. **Data sources and methods**

**3.1 Data sources**

**3.1.1 Digital elevation models**

Our reference elevation dataset across all three catchments is the Shuttle Radar Topographic Mission (hereafter SRTM) version 3.0, non-void filled, 1 arc second digital elevation model (hereafter DEM). The main objective

of the SRTM mission was to obtain single-pass interferometric SAR imagery to be used for DEM generation on a near global scale (56˚S to 60˚ N- 80% of the planet's surface) with targeted horizontal and vertical accuracies of 16 m and 20 m, respectively, although Farr et al. (2007) report horizontal and vertical accuracies of better than 10 m for most regions globally. This dataset was acquired in February 2000 and was released at 30 m

resolution in late 2014 (USGS, 2016). The SRTM data we used was acquired by a 5.6 cm C-band radar system.

Our 2014/2015 elevation dataset comprises a number of high resolution (8 m grid) DEMs generated by Ohio State University and distributed online by the Polar Geospatial Centre at the University of Minnesota that provide coverage of an extended area around the Everest region (Table 1). These stereo-photogrammetric DEMs have been generated using a Surface Extraction with TIN-based Search-space Minimization (hereafter SETSM)

algorithm from Worldview 1, 2 and 3 imagery (Noh and Howat, 2015). The SETSM algorithm is designed to automatically extract a stereo-photogrammetric DEM from image pairs using only the Rational Polynomial Coefficients (RPCs) as geometric constraints. The geolocation accuracy of RPCs without ground control for Worldview 1 and 2 data is 5 m (Noh and Howat, 2015) which may ultimately result in matching failure. The SETSM algorithm updates RPCs to mitigate this error and produces DEMs with an accuracy of $\pm$ 4 m in X, Y

and Z directions (Noh and Howat, 2015). SETSM DEMs are gap-filled using a natural neighbour interpolation; we removed these pixels before DEM differencing and the calculation of glacier mass balance.

Over two small areas of the Dudh Koshi (over the lower reaches of the Bhote Kosi and Melung glaciers) the SETSM DEMs contained data gaps. To complete coverage of DEMs over these glaciers we generated ASTER DEMs and used the surface to cover elevation bands across these glaciers where no data were available from the

SETSM grids. We used ERDAS Imagine (2013) to generate ASTER DEMs with ground control points (GCPs) matched between features in the ASTER imagery and the high resolution imagery available in Google Earth. We used a large number of GCPs (45) and tie points (> 75) to minimise the root mean squared (RMS) error of GCP positions. All SETSM and ASTER DEMs were resampled to a 30 m resolution to match that of the SRTM data before any differencing was carried out.

**3.1.2 Glacier outlines**

Glacier outlines were downloaded from the Global Land Ice Measurement form Space (GLIMS) Randolph Glacier Inventory (RGI) Version 5.0 (Arendt et al., 2015) and modified for 2000 and 2014 glacier extents based on Landsat scenes closely coinciding in acquisition with the DEM data. Glacier extents from these two epochs were used to calculate area changes. The 2000 Landsat scene was acquired by the Enhanced Thematic Mapper

Plus (ETM+) sensor and thus has a single 15 m resolution panchromatic band and six 30 m multispectral bands. The 2014 scene was acquired by the Operational Land Imager (OLI) sensor and also has a single 15 m panchromatic band as well as eight 30 m multispectral bands. Both scenes were pan-sharpened to match the resolution of the multispectral bands to that of the panchromatic band before glacier outlines were adjusted. Adjustments were limited to correcting changes in glacier frontal position and changes along the lateral margins because of surface lowering.

### 3.2 DEM correction

### 3.2.1 Stereoscopic DEMs

We followed the three-step correction process of Nuth and Kääb (2011), through which biases inherent in stereoscopic DEMs can be corrected. We assessed and corrected where necessary for: (i) mismatch in the geo-location of the modern DEMs versus the reference SRTM dataset (in x, y, and z direction); (ii) the existence of an elevation dependant bias, and; (iii) biases related to the acquisition geometry of the data. Each step was taken individually, so that separate error terms could be understood, rather than bundling them together as multiple regression based adjustments as previous studies have done, such as Racoviteanu et al. (2008) and Peduzzi et al (2010), for example. Corrections applied to DEMs where any one of the three biases were present included shifting of DEM corner coordinates, simple vertical shifting through addition or subtraction, and the fitting of linear and polynomial trends depending on the spatial variability of elevation differences across DEMs and through their elevation ranges. Acquisition geometry related biases (along or cross satellite track) were detected in two SETSM strips (Table 3) and both ASTER scenes and were corrected for using linear trends fitted through difference data. DEM co-registration was carried out following the conversion of SETSM elevation data to geoid heights using the Earth Gravitational Model (EGM) 2008 grid available from the National Geospatial-Intelligence Agency.

### 3.2.2 SRTM DEM correction

Some studies have shown that the SRTM dataset may underestimate glacier surface elevations because of C-band radar wave penetration into snow and ice (Rignot et al., 2001). Kääb et al. (2012) assessed the magnitude of C-band penetration over various test sites in the Himalaya and over different ice facies (clean ice, snow and firn) by extrapolating ICESat Vs SRTM glacier elevation differences back to the SRTM acquisition date, showing penetration estimates of several metres. To account for this bias, we have corrected the SRTM dataset using the penetration estimates of Kääb et al. (2012), after generating a mask for clean ice, firn and snow cover

using the most suitable Landsat ETM+ scenes (Table 1) available around the acquisition date of the SRTM dataset. We applied a correction to the SRTM DEM of +4.8 m over areas of firn/snow, and +1.2 m over areas of clean ice (see supplementary table S2 of Kääb et al. (2012)). We do not apply any penetration correction over debris-covered areas given the uncertainty expressed by Kääb et al. (2012) about the influence of possibly greater than average snowpack depth at the point of ICESat acquisition and the properties of the snowpack at the point of SRTM data acquisition on their penetration estimate.

Berthier et al. (2006) suggested that the extreme topography present in mountain regions is poorly replicated in coarse-resolution DEMs such as the SRTM DEM. Different studies have applied positive or negative corrections to the SRTM DEM (Berthier et al., 2007; Larsen et al., 2007), depending on the severity of the terrain at their respective study sites. Inspection of DEM differences across the study site showed no clear relationship between elevation differences and altitude (see supplementary information Figure 1), thus no elevation dependant correction was applied.

### 3.2.3 Gap filling and outlier filtering

Once DEMs had been co-registered and corrected for present biases, DEMs were differenced to yield surface elevation change data. To remove outlying values, we firstly excluded obviously incorrect difference values, (exceeding ± 120 m) and then followed the approach of Gardelle et al. (2013) in using the standard deviation of DEM difference data to classify probable outliers. We removed values exceeding 3 standard deviations. Such outlier definitions are justified in areas of shallow slope and high image contrast where DEM quality is generally high (Ragettli et al. 2016), but could be considered lenient where featureless surfaces, for example snow covered areas of accumulation zones, might lead to poor elevation data derivation and limit the accuracy of stereoscopic DEMs. Noh and Howat (2015) show how the iterative approach of the SETSM algorithm and the high spatial and radiometric resolution of WorldView imagery preclude such an issue, and we therefore consider a 3 standard deviation threshold appropriate.

To complete data coverage and allow for glacier mass balance estimates, the filling of data gaps was required. Only small (<~5 x 5 grid cells) gaps were present in DEM difference data over most of the glaciers in our sample, but some larger gaps could be found over areas of steep surface slope, for example high in accumulation zones, or where deep shadows might have been extensive in WorldView imagery. We filled gaps in DEM difference data using median values from the 100 m elevation band in which the data gap was situated (Ragettli et al. 2016).

3.3 **Uncertainty**

**3.3.1. DEM differencing uncertainty**

Our elevation change uncertainty estimates have been calculated through the derivation of the standard error $(E_{\Delta h})$ - the standard deviation of the mean elevation change- of 100 m altitudinal bands of elevation difference data (Gardelle et al., 2013; Ragettli et al., 2016):

$$E_{\Delta h} = \frac{\sigma_{stable}}{\sqrt{N}}$$

Where $\sigma_{stable}$ is the standard deviation of the mean elevation change of stable, off-glacier terrain, and $N$ is the effective number of observations (Bolch et al., 2011). $N$ is calculated through:

$$N = \frac{N_{tot} \cdot PS}{2d} \tag{1}$$

Where $N_{tot}$ is the total number of DEM difference data points, $PS$ is the pixel size and $d$ is the distance of spatial autocorrelation. We follow Bolch et al. (2011) in estimating $d$ to equal 20 pixels (600 m). $E_{\Delta h}$ for each DEM is the sum of standard error estimates of each altitudinal band (Gardelle et al., 2013).

We have also considered whether the different acquisition dates of Worldview imagery (Table 1) has led to the sampling of seasonal glacier surface elevation variations caused by a remnant snowpack (e.g. Berthier et al. 2016). Such a bias should be partly corrected for during vertical DEM adjustment using off-glacier terrain assuming a similar snowpack thickness on and off-glacier (Wang and Kääb, 2015). Two overlapping SETSM DEMs (ending FA100 and 3C00 in Table 1) have been generated from Worldview imagery acquired before and after the summer monsoon (when glaciers receive most accumulation) of 2014, thus any spatially consistent vertical differences may show a remnant snow pack that would cause an elevation bias. The difference between these two SETSM DEMs over the Bamolelingjia and G1 glaciers is slight (mean 0.69 m, $\sigma$ 3.81 m), but we cannot be sure that these differences represent a region-wide average. We have incorporated the mean elevation difference of these SETSM DEMs over glacier surfaces ($dZ_{season}$) into our overall uncertainty budget. We summed different sources of error quadratically to calculate our overall uncertainty ($\sigma_{dh/dt}$) associated with DEM difference data:

$$\sigma_{dh/dt} = \sqrt{E_{\Delta h}^2 + dZ_{season}^2} \tag{2}$$

$\sigma_{dh/dt}$ is then weighted depending on the hypsometry of each glacier, giving a glacier specific measure of elevation change uncertainty that considers the spatially nonuniform distribution of uncertainty (Ragettli et al., 2016).

### 3.3.2    Glacier area change uncertainty

There are two principal sources of uncertainty in the measurement accuracy of the position of a glacier margin; sensor resolution and the co-registration error between the images acquired at each measurement epoch (Ye et al., 2006; Thakuri et al., 2014). We follow the approach of Ye et al. (2006) to quantify the uncertainty associated with the total area changes documented across our sample of glaciers. We incorporate geolocation accuracy estimates of 10.5 m for Landsat ETM+ imagery and 6.6 m for Landsat OLI imagery (Storey et al., 2014) into the

uncertainty budget and suggest the total measurement uncertainty in glacier area between 2000 and 2015 image sets was ± 0.04 km$^2$ a$^{-1}$. Area weighted, glacier specific uncertainty estimates are given in Supplementary table 3.

### 3.4    Hypsometric analyses and elevation range normalisation

Glacier hypsometry, the distribution of glacier area over altitude, is governed by valley shape, relief and ice

volume distribution (Jiskoot et al., 2009). It is important for long-term glacier response because it defines the distribution of mass with elevation and thus determines how the glacier responds to changes in elevation-dependent temperature (Furbish and Andrews, 1984). To assess glacier hypsometry, we used the aforementioned glacier outlines and the SETSM DEMs, which offer better data coverage than the non-void filled SRTM dataset, to split these glacier extents into segments covering 100 m elevation ranges, and calculated

the area of each segment. We followed the approach of Jiskoot et al. (2009) to categorise each glacier or the population of glaciers in each catchment according to a hypsometric index (*HI*), where:

$$HI = \frac{(H_{max} - H_{med})}{(H_{med} - H_{min})} \qquad (3)$$

and $H_{max}$ and $H_{min}$ are the maximum and minimum elevations of the glacier, and $H_{med}$ the median elevation that divides the glacier area in half (Jiskoot et al., 2009). Glaciers were grouped into five HI categories: 1- HI <

25    −1.5, very top heavy; 2- HI −1.2 to −1.5, top heavy; 3- HI −1.2 to 1.2, equidimensional; 4- HI 1.2 to 1.5, bottom heavy; and 5- HI > 1.5, very bottom heavy. Top heavy glaciers store more ice at higher elevation, for example in broad accumulation zones, whereas bottom heavy glaciers have small accumulation zones and long tongues.

To construct elevation change and glacier hypsometry curves for the 32 glaciers in our sample, we have normalised the elevation range of each glacier following the method of Arendt et al. (2006):

$$H_{norm} = \frac{(H - H_{min})}{(H_{max} - H_{min})}$$ ( 7 )

where $H_{min}$ and $H_{max}$ are the elevations of the glacier terminus and the elevation maximum of each glacier. This normalisation process allows the direct comparison of elevation changes and glacier hypsometry regardless of termini elevation. Surface elevation change and glacier hypsometry curves are presented in Figures 5 and 6.

**3.5 Mass loss calculations**

A conversion factor of 850 kg m$^{-3}$ was used to account for the density of glacier ice for all glaciers in the sample (Huss, 2013). We assigned an additional 7 % to mass loss uncertainty estimates to account for error in the density conversion (Huss, 2013). The mass loss estimates generated for lacustrine terminating glaciers are slight underestimates because, with no information available on bed topography, we cannot account for ice that has been replaced by water during lake expansion. Mass balance estimates for these glaciers therefore only incorporate aerial mass loss from the 2000 calving front, up-glacier. We also acknowledge that our surface lowering estimates incorporate any upward or downward flow of ice resulting from, for example, compressional flow over a zone of transition from active to inactive ice. We do not quantify emergence velocity as the ice thickness and surface velocity data required to do so (Immerzeel et al. 2014) are not available for an adequate number of glaciers in our sample.

**3.6 Estimation of ELAs**

We follow the method of Nuth et al. (2007) to estimate the ELA of glaciers in our sample. We calculate mean surface lowering rates over 100 m elevation bands for the entire altitudinal range of each glacier, and assume that the altitude at which surface lowering curves approach zero is a reliable proxy of the ELA of each glacier over the study period. Although the surface elevation change at a glaciers surface is not a direct measure of its surface mass balance because of ice flux divergence, the ELA is likely to coincide with minimal submergence and emergence (Cherkasov and Ahmetova, 1996; Pope et al., 2016). We therefore use the point of zero elevation change as an approximation of ELA position (see Huss and Farinotti (2012) their Figure 1 and Farinotti et al. (2009) their Figure 11).

To estimate prospective future ELAs in response to climatic warming, we used vertical temperature gradients of –8.5 $^{\circ}$C km$^{-1}$ for the Tibetan Plateau (Kattel et al., 2015) and –5.4 $^{\circ}$C km$^{-1}$ for the Dudh Koshi and Tama Koshi

catchments (Immerzeel et al., 2014) to calculate prospective ELA shifts given different warming scenarios. We calculated ELAs for projected minimum, mean and maximum temperature increases under the 4 main RCP scenarios outlined in the IPCC AR5 working group report (Collins et al., 2013).

## 4 Results

### 4.1 Glacier mass balance

The mean mass balance of all 32 glaciers in our sample was $-0.52 \pm 0.22$ m w.e. a$^{-1}$ between 2000 and 2015. There is considerable variability in the mass balance of glaciers with different terminus type (Figures 3 and 4) and in the rates of surface lowering through the altitudinal range of highlighted glaciers (Figures 5 and 6). The mean mass balance of glaciers in catchments either side of the orographic divide are not markedly different, however.

Mean glacier mass balance (including land and lacustrine terminating glaciers) was $-0.51 \pm 0.22$ m w.e. a$^{-1}$ in the Tama Koshi catchment, $-0.58 \pm 0.19$ m w.e. a$^{-1}$ in the Dudh Koshi catchment, and $-0.61 \pm 0.24$ m w.e. a$^{-1}$ for glaciers flowing onto the Tibetan Plateau over the study period. The mean mass balance of nine lacustrine terminating glaciers was $-0.70 \pm 0.26$ m w.e. a$^{-1}$. This was 32% more negative than land terminating glaciers (mean mass balance of $-0.53 \pm 0.21$ m w.e. a$^{-1}$) we include in our sample. The lowest mass loss rates occurred over debris-free glaciers at high altitude (5600 – 6200 m a.s.l) on the Tibetan plateau. The mean mass balance of these glaciers was $-0.25 \pm 0.22$ m w.e. a$^{-1}$ (Supplementary table 2) over the study period. Individual glacier mass balance estimates can be found in the Supplementary Information.

### 4.2 Glacier surface lowering

The altitude at which maximum surface lowering rates occurred differed depending only on glacier terminus type (Figures 5 and 6). Across all three catchments, substantial surface lowering was pervasive over the middle portions of larger, land terminating glaciers (Figure 2). In the Dudh Koshi, surface lowering rates are at their highest ($-1.06 \pm 0.10$ m a$^{-1}$) around 5200 m a.s.l., although similar surface lowering rates occurred between 5100 and 5300 m a.s.l (Figure 5). In the Tama Koshi the highest rates of surface lowering ($-1.08 \pm 0.12$ m a$^{-1}$) occurred at around 5400 m a.s.l (Figure 5). On the Tibetan Plateau, the highest mean surface lowering rates again occurred between 5300 and 5400 m a.s.l.; the mean surface lowering rate at this altitude was $-1.62 \pm 0.14$ m a$^{-1}$ over the study period. Surface lowering rates over glaciers on the Tibetan Plateau were higher than those in the Tama Koshi and Dudh Koshi catchments (Figure 5) up to 5700 m a.s.l. ($-1.24 \pm 0.21$ m a$^{-1}$ at this altitude). Of note is the surface lowering over clean-ice areas high up on glaciers such as Ngozumpa, Rongbuk,

Gyabrag and Bhote Kosi (Figure 2). Surface lowering extended into tributary branches and the cirques of these largest glaciers. Individual glaciers showed much greater surface lowering, particularly on the Tibetan Plateau. Gyabrag glacier lost an exceptional $-3.33 \pm 0.28$ m a$^{-1}$ between 5300 and 5400 m a.s.l (Figure 5).

The maximum surface lowering rates ($-2.79 \pm 0.29$ m a$^{-1}$) occurred at the lowest elevations (between 4700 and 4900 m a.s.l) of lacustrine terminating glaciers (Figure 6). These nine glaciers all showed a linear surface lowering gradient. We calculate the lowering gradient as surface elevation change per 100 m (m a$^{-1}$/100 m) vertical elevation change below the ELA. Lacustrine terminating glaciers showed a lowering gradient of 0.30 m a$^{-1}$/100 m over the study period. The lowering gradient of land terminating glaciers was non-linear. Surface lowering was negligible around the terminus of most land terminating glaciers, with enhanced ice loss occurring further up-glacier where debris cover may have been thin or patchy. Lowering gradients for the area of land terminating glaciers between the ELA and the altitude of maximum ice loss were 0.59, 0.66 and 0.38 m a$^{-1}$/100 m for glaciers on the Tibetan Plateau, and in the Dudh Koshi and Tama Koshi catchments, respectively. Clean ice glaciers also showed a linear lowering gradient- 0.77 m w.e. a$^{-1}$/100 m.

**4.3 Glacier area changes and hypsometry**

**4.3.1 Total area changes**

Two different patterns of ice area loss occurred over the study area during the last 15 years. Lacustrine terminating glaciers and clean ice glaciers all lost ice around their termini/calving fronts (Figures 3 and 4) as glacial lakes expanded and termini receded. On average, lacustrine terminating glaciers each lost $0.54 \pm 0.07$ km$^2$ of ice (3.58% of their total area) over the 15 year study period. Drogpa Nagtsang reduced in size by 2.37 km$^2$ (9.12 % of its total area: Supplementary table 3) as the associated rapidly-forming lake expanded. Clean ice glaciers lost $0.09 \pm 0.03$ km$^2$ of ice (1.31 % of their total area) on average.

Land terminating glaciers lost little area as their surfaces lowered rather than their termini retreating. In the Tama Koshi and Dudh Koshi catchments, and on the Tibetan Plateau, land terminating glaciers lost a mean of $0.14 \pm 0.12$ km$^2$ (0.50 % of their total area), $0.09 \pm 0.13$ km$^2$ (0.60 % of their total area) and $0.41 \pm 0.12$ (1.77 % of their total area) of ice, respectively. Over these glaciers, any ice area loss was concentrated up-glacier, where their lateral margins dropped down inner moraine slopes and glacier tongues narrowed slightly.

Overall, our sample of glaciers lost $0.12 \pm 0.04$ % of their total area per year over the study period. This figure is identical to that of Bolch et al. (2008) who assessed area change over a smaller number of the same glaciers in our sample between 1962 and 2005. The annual area change rate we calculate is lower than those estimated by

Thakuri et al. (2014) and references within. Thakuri et al. (2014) calculated a median annual surface area change rate of $-0.42 \pm 0.06$ % $a^{-1}$ in the Dudh Koshi catchment between 1962 and 2011. However, Thakuri et al. (2014) document area change over a number of smaller glaciers that are free of debris cover, and therefore readily advance or retreat in response to climatic change, thus our estimates are not directly comparable.

**4.4.2 Glacier hypsometry and approximate ELAs**

The distribution of ice with elevation varies widely among the three studied catchments (Figures 5 and 6). Debris-covered glaciers of the Dudh Koshi catchment and on the Tibetan Plateau are typically very bottom heavy, with average HI scores of 2.63 and 2.34, respectively (Supplementary table 1). Glacier hypsometry is concentrated between 4800 and 5500 m (Figure 5) for the Dudh Koshi catchment, and between 5600 and 6500 m on the Tibetan Plateau. Notable exceptions are Khumbu and Ngozumpa Glaciers which store ice in broad accumulations zones above 7000 m (Supplementary tables 1 and 2). The majority of glaciers in the Tama Koshi have an equi-dimensional hypsometry (mean HI of 1.14), with most ice stored between 5300 and 5800 m. Glaciers in the Tama Koshi have broader accumulation basins than in the Dudh Koshi catchment, and main glacier tongues are formed of multiple, smaller tributaries flowing from higher altitude in a number of cases (Figure 1). The mean hypsometry (Figure 6) of lacustrine terminating glaciers shows no distinctive morphology as the sample is composed of glaciers from all three catchments in the study area. Clean ice glaciers have a mean HI of 1.18 and could therefore be summarised as equidimensional, but the morphology of the 5 glaciers we assess is highly variable (see Supplementary table 3). In complete contrast to debris-covered glaciers, their ice is stored at higher mean altitudes on average; primarily between 6000 and 6500 m (Figure 6).

We estimate the mean ELA of debris-covered glaciers glaciers in the Dudh Koshi and Tama Koshi catchments, and of our selection of glaciers on the Tibetan Plateau to be 5742, 5633, and 6155 m a.s.l., respectively, although, as Figures 5 and 6 show, the altitude of zero surface lowering can differ by hundreds of metres on individual glaciers within each catchment. We estimate the mean ELA of the 5 clean-ice glaciers in our sample to be 6180 m. Using those ELAs the accumulation area ratio (AAR) (Dyurgerov et al., 2009) can be estimated for each glacier and this is a parameter strongly related to long-term mass balance (König et al., 2014). We have calculated mean AARs of 0.37, 0.36 and 0.40 for debris-covered glaciers in the Dudh Koshi and Tama Koshi catchments, and on the Tibetan Plateau, respectively. The mean AAR of clean ice glaciers in our sample is 0.29.

**5    Discussion**

**5.1 Variability in rates of ice loss across the orographic divide**

The mean mass balance estimates we have derived for glaciers situated in catchments North and South of the main orographic divide are not markedly different. However, the contrast in maximum of surface lowering (Figure 5) from glaciers flowing north of the divide and the sustained surface lowering through a broader portion of their elevation range (Figure 5) suggests an additional or amplified process has driven glacier change north of the divide over recent decades. In this section we discuss possible topographic and climatic drivers of the difference in the rates of surface lowering across the range divide.

The Indian summer monsoon delivers a large proportion of total annual precipitation (up to 80% of the total annual amount) to the Everest region of Nepal, resulting in high glacier sensitivity to temperature (Fujita, 2008; Sakai et al., 2015). The extreme topography in this region and the location of the orographic divide perpendicular to the prevailing monsoon result in rainfall peaks that are offset from the maximum elevations, with greatest rainfall occurring to the south of the divide and decreasing to the north across the Everest region (Bookhagen and Burbank, 2010; Wagnon et al. 2013). Around 449 mm a$^{-1}$ of rainfall falls at the Pyramid research station (5000 m a.s.l.) at Khumbu Glacier (Salerno et al., 2015), whereas Dingri on the Tibetan Plateau (4300 m a.s.l.) to the north is much drier with $263 \pm 84.3$ mm a$^{-1}$ of rainfall annually (Yang et al., 2011). Snowfall may follow a similar across-range gradient to rainfall, although falling snow may be carried further into the range by prevailing winds from the south. However, no reliable measurements of snowfall exist in this region with which to compare these trends. The north-south precipitation gradient across the orographic divide promotes differences in the response of these glaciers to climate change, such that those to the north are relatively starved of snow accumulation (Owen et al., 2009) and exposed to greater incoming radiative fluxes under generally clearer skies. Owen et al. (2009) suggest that this precipitation gradient resulted in greater glacier sensitivity to climate change on the northern slopes of the Himalaya during the Late Quaternary, with asymmetric patterns of ELA rise occurring since the Last Glacial Maximum (LGM).

During the period of this study (2000–2015), mean annual air temperatures have increased and rainfall amounts appear to have decreased in the Everest region (Salerno et al., 2015). At the Pyramid Observatory at Khumbu Glacier in the Dudh Koshi catchment, increases in minimum (+0.07 °C a$^{-1}$), maximum (+0.009 ° C a$^{-1}$) and mean (+0.044 ° C a$^{-1}$) annual air temperatures above 5000 m a.s.l. were observed between 1994 and 2013 (Salerno et al., 2015). At Dingri on the Tibetan Plateau 60 km northeast of Mt. Everest, increases in minimum (+0.034 ° C a$^{-1}$), maximum (+0.041 ° C a$^{-1}$) and mean (+0.037 ° C a$^{-1}$) annual air temperatures occurred over the same period (Salerno et al., 2015). Yang et al. (2011) also show a longer-term increase in the mean annual air temperature at Dingri, as do Shrestha et al. (1999) across the southern flank of the greater Himalaya. Between

1959 and 2007, the mean annual air temperature increased by 0.06 ° C a$^{-1}$ at Dingri (Yang et al., 2011). Shestha et al. (1999) calculated an increase in the mean annual air temperature of 0.057 ° C a$^{-1}$ between 1971 and 1994 across a number of sites in the greater Himalaya.

The snowline altitude also appears to have increased recently on the southern flank of the Himalaya; Thakuri et al. (2014) showed a rapid ascent of the snow-line altitude in the Dudh Koshi between 1962 and 2011 (albeit through documenting transient snowlines from single scenes acquired at each epoch), and Khadka et al. (2014) suggest declining snow cover over the winter and spring months in the glacierised altitudinal ranges of the Tama Koshi catchment, between 2000 and 2009; a factor that may influence accumulation rates. Kaspari et al. (2008) showed decreasing accumulation recorded in an ice core collected from East Rongbuk Glacier Col (6518 m a.s.l.) on the northern side of Mount Everest between the 1970s and 2001.

We suggest that the north–south orographic precipitation gradient across the main divide may have caused greater surface lowering rates on glaciers on the Tibetan Plateau than those glaciers to the south over the study period. We also suggest that measured, contemporary increases in air temperature, observations of increasing snowline altitude and declining accumulation are likely to enhance glacier mass loss across the range in future, but considerable unknowns remain in the temporal evolution of debris cover extent and thickness (Thakuri et al., 2014), the strength of the summer monsoon in coming decades (e.g. Boos et al., 2016), and the expansion or shrinkage of glacial lakes (see section 5.3), all of which could additionally influence future glacier mass balance.

### 5.2 Comparison of mass balance estimates with other studies

Several other studies have generated geodetic mass balance estimates for glaciers of the Everest region over several different time periods. Bolch et al. (2011) measured a mass balance of –0.32 ± 0.08 m w.e. a$^{-1}$ for ten glaciers to the south and west of Mt Everest over the period 1970-2007. Nuimura et al. (2012) calculated a regional mass balance of –0.45 ± 0.25 m w.e. a$^{-1}$ for 97 glaciers across the region over the period 1992-2008. Kääb et al. (2012) estimated a mass balance of –0.39 ± 0.11 m w.e. a$^{-1}$ for a 3˚ x 3˚ cell centred on the Everest region between 2003 and 2008. Gardelle et al. (2013) calculated a slightly less negative mass balance of –0.26 ± 0.13 m w.e. a$^{-1}$ between 1999 and 2011, although the SRTM penetration correction applied by Gardelle et al. (2013) may have caused bias towards less negative mass balance (Kääb et al. 2012; Barundun et al., 2015). The regional mass balance of –0.52 ± 0.22 m w.e. a$^{-1}$ that we have calculated suggests that the mass loss rates measured by Bolch et al. (2012), Nuimura et al. (2012) and Kääb et al. (2012) have been sustained and possibly increased in recent years (Table 3).

On the Tibetan Plateau, Neckel et al. (2014) estimated the mass balance of glaciers on the northern side of the orographic divide in the central and eastern Himalaya (their sub-region G) to be $-0.66 \pm 0.36$ m w.e. a$^{-1}$ between 2003 and 2009. The mass balance of glaciers in our sample within the same region was $-0.59 \pm 0.27$ m w.e. a$^{-1}$ between 2000 and 2015. Ye et al. (2015) estimated glacier mass balance to be $-0.40 \pm 0.27$ m w.e. a$^{-1}$ in the Rongbuk catchment between 1974 and 2006, suggesting that glacier ice mass loss rates may have increased over the last decade in this area of the Tibetan Plateau (Table 3).

**5.3 The influence of glacial lakes on glacier mass balance**

Only Nuimura et al. (2012) have directly compared mass loss rates of lacustrine and land terminating glaciers in the study area, showing faster surface lowering rates over Imja and Lumding glaciers in the Dudh Koshi catchment. Our data confirm that lacustrine terminating glaciers can indeed lose ice at a much faster rate than land terminating glaciers. The variability in the mass balance of the nine lacustrine terminating glaciers (Figure 6) we highlight suggests the fastest mass loss rates occur in the later stages of lake development. Glaciers such as the Yanong and Yanong North, in the Tama Koshi catchment, sit behind large proglacial lakes and are in a state of heavily negative mass balance ($-0.76 \pm 0.18$ and $-0.62 \pm 0.25$ m w.e. a$^{-1}$, respectively). Their surfaces lowered by 3 m a$^{-1}$ or more over their lower reaches (Figure 6) over the study period. These glaciers are now relatively small and steep and no longer possess a debris-covered tongue, and so may represent the end-product of debris-covered glacier wastage described by Benn et al. (2012). In contrast, glaciers such as Duiya, on the Tibetan Plateau, currently has only a small lake at its termini, showed moderate area losses (0.5 km$^2$, or 4.28% of its total area) and moderately negative mass balance ($-0.45 \pm 0.13$ m w.e. a$^{-1}$) over the study period. Continued thinning of the terminal regions of glaciers with smaller glacial lakes would lead to a reduction in effective pressure, an increase in longitudinal strain and therefore flow acceleration (Benn et al., 2007). The retreat of the calving front up-valley into deeper bed topography may also increase calving rates (Benn et al., 2007), and a combination of both of these processes would lead to enhanced ice loss. Very little surface velocity data exist for lacustrine terminating debris-covered glaciers. Only Quincey et al. (2009) measured high surface velocities (25 m a$^{-1}$ or more) over Yanong glacier (their Figure 4, panel D), suggesting it is possible for lacustrine terminating glaciers to become more dynamic in the later stages of lake development in the Himalaya. Conversely, Thakuri et al. (2016) have shown flow deceleration of glaciers that coalesce to terminate in Imja Tsho over the period 1992-2014, and suggest that reduced accumulation caused by decreasing precipitation is responsible for diminishing surface flow on this glacier. Clearly, more expansive investigation into the evolving

dynamics of lacustrine terminating glaciers in the Himalaya is required if we are to better understand their potential future mass loss.

**5.4 Glacier stagnation**

A number of studies (Luckman et al. 2007; Scherler et al. 2008, 2011; Quincey et al. 2009) have shown how many glaciers in the Everest region appear to be predominantly stagnant, with large parts of the long, debris-covered glacier tongues in the area showing little to no flow. Watson et al. (2016) have documented an increasing number and total area of supraglacial melt ponds over a number of the same glaciers studied by Quincey et al. (2009) in the Dudh Koshi catchment (Khumbu, Ngozumpa, Lhotse, Imja and Ama Dablam), since the early 2000's. Over these glaciers, our data show a very distinctive surface lowering pattern (Figure 2), with localised, heterogenous surface lowering appearing to mirror the distribution of large supraglacial ponds and ponds networks. This ice loss pattern is prevalent on the Erbu, Gyachung, Jiuda, Shalong, and G1 glaciers (Figure 2), and high resolution imagery available on Google Earth shows that these glaciers also have well developed networks of supraglacial ponds. We would therefore suggest that large parts of the biggest glaciers in the Tama Koshi catchment and on the Tibetan Plateau are also stagnant, and may see increasing supraglacial meltwater storage in the future, similar to that documented by Watson et al. (2016).

**5.5 Susceptibility of glaciers to future mass loss**

**5.5.1 ELA ascent in response to warming**

The coincidence of maximum surface lowering rates with the altitude of maximum hypsometry in the Dudh Koshi catchment (Figure 5) suggests large glacier mass losses in this catchment. Sustained and prolonged mass loss may lead to a bi-modal hypsometry here, with the physical detachment of debris-covered glacier tongues and their high-elevation accumulation zones a possibility (Rowan et al., 2015; Shea et al., 2015). Surface lowering maxima in the Tama Koshi catchment presently occur at a slightly lower elevation range than the main hypsometric concentration, and across lower reaches of glacier tongues on the Tibetan Plateau.

Our ELAs are above those estimated by Asahi (2001) for an earlier epoch (see section 2) and similar to those estimated by Gardelle et al. (2013) over a similar study period to ours, suggesting that ELAs have indeed risen over recent decades. Figure 7 shows projected AARs, averaged across each catchment, in response to different levels of temperature rise. These predictions are based on published lapse rates (Immerzeel et al., 2014; Kattel et

al., 2015) that may be spatially variable and assume no changes in precipitation type or amount, or any variability in the contribution of avalanches to accumulation.

To allow the comparison of our results with similar estimates of other studies (Shea et al., 2015; Rowan et al., 2015), we focus specifically on ELA rise resulting from RCP 4.5 minimum and maximum projected warming of annual air temperatures (+0.9 $^{\circ}$C to +2.3 $^{\circ}$C by 2100). Such temperature increases would cause a rise in ELA of between 165 and 425 m in the Dudh and Tama Koshi catchments, and between 107 and 270 m of ELA ascent over glaciers on the Tibetan Plateau. A rise in ELAs would most significantly affect the Tama Koshi catchment glaciers, with AARs potentially decreasing to 0.25 and 0.03, respectively. The greater altitudinal range and higher accumulation zones of glaciers in the Dudh Koshi catchment and on the Tibetan Plateau would dampen the effects of a rise in ELA on glacier mass balance. AARs could decrease to 0.26 or 0.18 in the Dudh Koshi and to 0.30 or 0.17 on the Tibetan Plateau. ELA rise in response to this particular warming scenario would mean a 17–51% increase in the total glacierised area below the ELA on the Tibetan Plateau, a 17–30 % increase in the Tama Koshi catchment, and a 14-37% increase in the Dudh Koshi catchment.

Should greater temperature increases occur, for example high-end RCP 6.0 warming (+1.3 $^{\circ}$C to +2.7 $^{\circ}$C by 2100), AARs could reduce to zero in the Tama Koshi catchment as ELAs rise above glacierised altitudes. Clean ice glacier AAR adjustment could be rapid given more than 1 $^{\circ}$C of warming, with AARs again approaching zero should high-end RCP 8.5 warming occur in the region. The AAR of glaciers in the Dudh Koshi catchment could reduce quickly under RCP 2.6 warming, but their AAR reduction may be less rapid given greater temperature increases, presumably because of the extreme relief of the catchment. The AAR of glaciers on the Tibetan Plateau could become lower than glaciers of the Dudh Koshi catchment once warming approaches 2.5 $^{\circ}$C.

## 6 Conclusions

DEM differencing has revealed substantial mass loss from many large, debris-covered glaciers in the central Himalaya over the last 15 years. Geodetic mass balance estimates have been calculated for 32 glaciers across three different catchments around the Everest region. We found similarly negative mass budgets for glaciers flowing onto the southern flank of the Himalaya, in the Tama Koshi (–0.51 ± 0.22 m w.e. a$^{-1}$) and Dudh Koshi (–0.58 ± 0.19 m w.e. a$^{-1}$) catchments, and onto the Tibetan Plateau (–0.61 ± 0.24 m w.e. a$^{-1}$).

The division of our sample of glaciers depending on their terminus type shows contrasting mass loss rates between land and lacustrine terminating glaciers. The mean mass balance of nine lacustrine terminating glaciers

we assessed was $-0.70 \pm 0.26$ m w.e. a$^{-1}$, 32% more negative than land terminating glaciers (mean mass balance of $-0.53 \pm 0.21$ m w.e. a$^{-1}$). The mass balance of nine lacustrine terminating glaciers ranged from $-0.91 \pm 0.22$ m w.e. a$^{-1}$ to $-0.45 \pm 0.13$ m w.e. a$^{-1}$ and we would suggest that glacial lakes in the region are at different stages of expansion. Accelerating mass loss is likely from several of these lacustrine terminating glaciers whose

termini will retreat into deeper lake water.

Surface lowering curves show that the maximum lowering rate ($-1.62 \pm 0.14$ m a$^{-1}$ between 5300 and 5400 m.a.s.l.) of glaciers flowing onto the Tibetan Plateau was well above the maximum lowering rate of glaciers flowing south of the orographic divide ($-1.06 \pm 0.10$ m a$^{-1}$ between 5200 and 5300 m a.s.l. in the Dudh Koshi catchment, $-1.08 \pm 0.12$ m a$^{-1}$ between 5200 and 5300 m a.s.l. in the Tama Koshi catchment), and that glaciers

flowing onto the Tibetan Plateau are losing ice over a much broader altitudinal range than their south-flowing counterparts. We suggest that the across-range contrast in annual precipitation amount, combined with rising mean air temperatures over recent decades may have caused greater ice loss rates from the north flowing glaciers.

Predicted warming in the Everest region will lead to increased ELAs and, depending on glacier hypsometry,

substantial increases in the size of ablation areas. We show that glaciers of the Tama Koshi catchment will see the greatest reduction in glacier AAR due to their equidimensional hypsometry and more limited elevation range in comparison to glaciers of the Dudh Koshi or Tibetan Plateau. Warming of +0.9 $^{o}$C to +2.3 $^{o}$C by 2100 (IPCC RCP 4.5) would decrease glacier AAR to 0.25 or 0.03 in the Tama Koshi catchment, 0.26 or 0.18 in the Dudh Koshi catchment, and 0.30 or 0.17 on the Tibetan Plateau, respectively.

Our findings are important for two reasons. First, they suggest that glacial lake growth and current glacial lake expansion that has been documented across the Himalaya could be accompanied by amplified glacier mass loss in the near future. Second, they show that glacier AAR adjustment in response to predicted warming across the Himalaya could be spatially very variable, complicating the prediction of future glacier meltwater runoff contribution from river catchments across the region.

Author contribution

OK, DQ and JC designed the study. OK carried out all data processing and analysis. OK, DQ, JC and AR wrote the paper.

Acknowledgements

SETSM DEMs are available for download from http://www.pgc.umn.edu/elevation. The SRTM dataset is available from https://lta.cr.usgs.gov/SRTM1Arc. EGM2008 gridded data is available from http://earth-info.nga.mil/GandG/wgs84/gravitymod/egm2008/egm08_gis.html. OK is a recipient of a NERC DTP PhD studentship. We are grateful for the comments of Benjamin Robson for his comments on an early version of the paper, and for guidance on the use of SETSM data from Ian Howat. We finally thank Tobias Bolch, Joseph Shea and an anonymous reviewer for their thorough and constructive assessments of the manuscript.

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

Table 1. Scenes used in glacier outline delineation, ASTER DEM generation, SRTM ice facies mask generation and by the Polar Geospatial Centre in the generation of SETSM DEMs.

| Sensor | Scene ID | Date of acquisition | Purpose |
|---|---|---|---|
| Landsat OLI | LC81400412014334LGN00 | 30/11/2014 | Glacier outlines |
| Landsat ETM+ | LE71390412000302SGS00 | 29/10/2000 | Glacier outlines |
| Landsat ETM+ | LE71400402002005SGS00 | 05/01/2002 | Ice facies mask |
| Landsat ETM+ | LE71400412002005SGS00 | 05/01/2002 | Ice facies mask |
| ASTER | L1A.003:2014050545 | 29/11/2014 | ASTER DEM |
| Worldview 3 | WV03_20150121_10400100076C0700 | 21/01/2015 | SETSM DEM |
| Worldview 1 | WV01_20150504_102001003C5FB900 | 04/05/2015 | SETSM DEM |
| Worldview 1 | WV01_20140115_102001002A289F00 | 15/01/2014 | SETSM DEM |
| Worldview 1 | WV01_20140324_102001002D263400 | 24/03/2014 | SETSM DEM |
| Worldview 1 | WV01_20150204_102001003A5B7900 | 04/02/2015 | SETSM DEM |
| Worldview 2 | WV02_20150202_103001003D4C7900 | 02/02/2015 | SETSM DEM |
| Worldview 1 | WV01_20140218_102001002C5FA100 | 18/02/2014 | SETSM DEM |
| Worldview 1 | WV01_20141022_102001003525D400 | 22/10/2014 | SETSM DEM |
| Worldview 2 | WV02_20141110_1030010039013C00 | 10/11/2014 | SETSM DEM |
| Worldview 1 | WV01_20141129_102001002776B500 | 29/11/2014 | SETSM DEM |
| Worldview 1 | WV01_20140514_102001003001E400 | 14/05/2014 | SETSM DEM |

Table 2. Mean differences and the standard deviation associated with off-glacier elevation difference data between ASTER, SETSM and SRTM DEMs before and after the DEM correction process. The uncertainty associated with DEM difference data (sum of standard error estimates for each 100 m elevation bin of difference data) is also listed for each SETSM and ASTER DEM.

| Sensor | ASTER scene ID | Pre correction mean & StDev stable ground differences Vs SRTM (m) | | Post correction mean & StDev stable ground differences Vs SRTM (m) | | dh/dt uncertainty ($\pm$ m a$^{-1}$) |
|--------|----------------|--------|--------|--------|--------|--------|
| ASTER | L1A.003:2014050545 | −64.12 | 25.99 | 0.43 | 11.30 | 0.47 |
| | SETSM tile | | | | | |
| WV 3 | WV03_20150121_10400100076C0700 | −6.07 | 11.54 | 0.53 | 6.43 | 0.25 |
| WV 1 | WV01_20150504_102001003C5FB900 | −5.68 | 15.76 | −0.43 | 5.89 | 0.40 |
| WV 1 | WV01_20140115_102001002A289F00 | −3.56 | 9.50 | 0.50 | 6.64 | 0.27 |
| WV 1 | WV01_20140324_102001002D263400 | −2.21 | 8.92 | 0.07 | 5.90 | 0.33 |
| WV 1 | WV01_20150204_102001003A5B7900 | −1.26 | 17.50 | −0.36 | 5.65 | 0.31 |
| WV 2 | WV02_20150202_103001003D4C7900 | −3.80 | 12.34 | −0.03 | 6.56 | 0.29 |
| WV 1 | WV01_20140218_102001002C5FA100 | −2.00 | 9.80 | −0.23 | 6.71 | 0.28 |
| WV 1 | WV01_20141022_102001003525D400 | −9.54 | 16.50 | 0.36 | 6.89 | 0.35 |
| WV 2 | WV02_20141110_103001003E9013C00 | −2.89 | 9.83 | 0.07 | 5.87 | 0.15 |
| WV 1 | WV01_20141129_102001002776B500 | −5.72 | 8.31 | 0.16 | 4.76 | 0.18 |
| WV 1 | WV01_20140514_102001003001E400 | −3.51 | 10.12 | −0.26 | 5.91 | 0.26 |

Table 3. Mass balance estimates (from geodetic and altimetric studies) for the broader Everest region and comparable sub-regions/ catchments.

| Time period and area | Mass balance estimate (m w.e. a$^{-1}$) | Study |
|---|---|---|
| **Dudh Koshi** | | |
| 1970-2008 | −0.32 ± 0.08 | Bolch et al. (2011) |
| 1992-2008 | −0.45 ± 0.25 | Nuimura et al. (2012) |
| 2000-2015 | −0.58 ± 0.19 | This study |
| **Pumqu (Tibetan Plateau)** | | |
| 1974-2006 | −0.40 ± 0.27 | Ye et al. (2015) |
| 2003-2009 | −0.66 ± 0.32 | Neckel et al. (2014) |
| 2000-2015 | −0.61 ± 0.24 | This study |
| **Tama Koshi** | | |
| 2000-2015 | −0.51 ± 0.22 | This study |
| **Everest region** | | |
| 1999-2011 | −0.26 ± 0.13 | Gardelle et al. (2013) |
| 2003-2008 | −0.39 ± 0.11 | Kääb et al. (2012) |
| 2000-2015 | −0.52 ± 0.22 | This study |

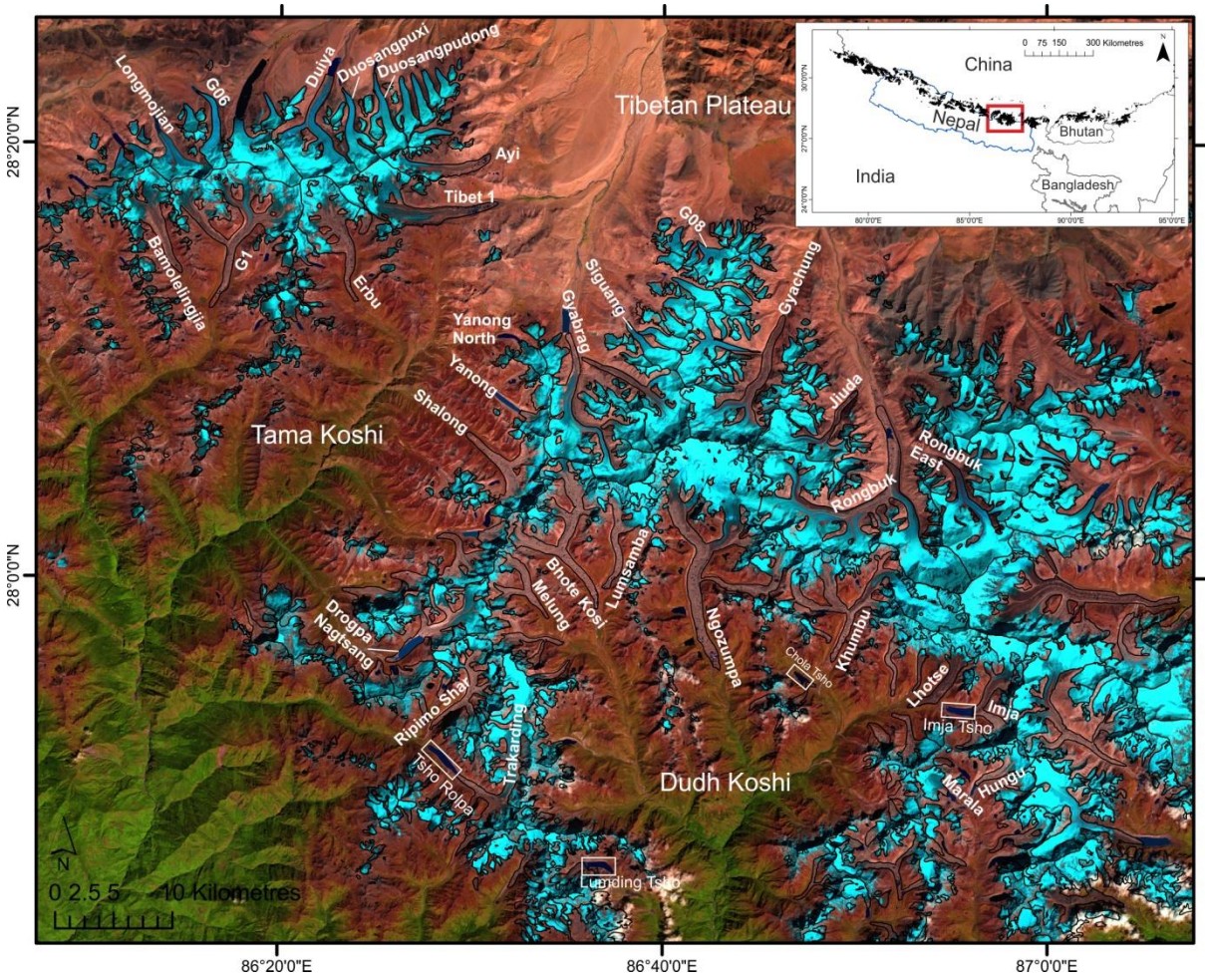

Figure 1. The glaciers of the Everest region. Named glaciers are the glaciers we highlight in this study. Major catchments include the Tama Koshi and Dudh Koshi on the southern flank of the Himalaya and the Pumqu river catchment on the northern side of the divide, with glaciers flowing onto the Tibetan Plateau (China). Named glacial lakes are highlighted, although many remain unnamed. Background imagery is a Landsat OLI image from 2014 available from http://earthexplorer.usgs.gov/.

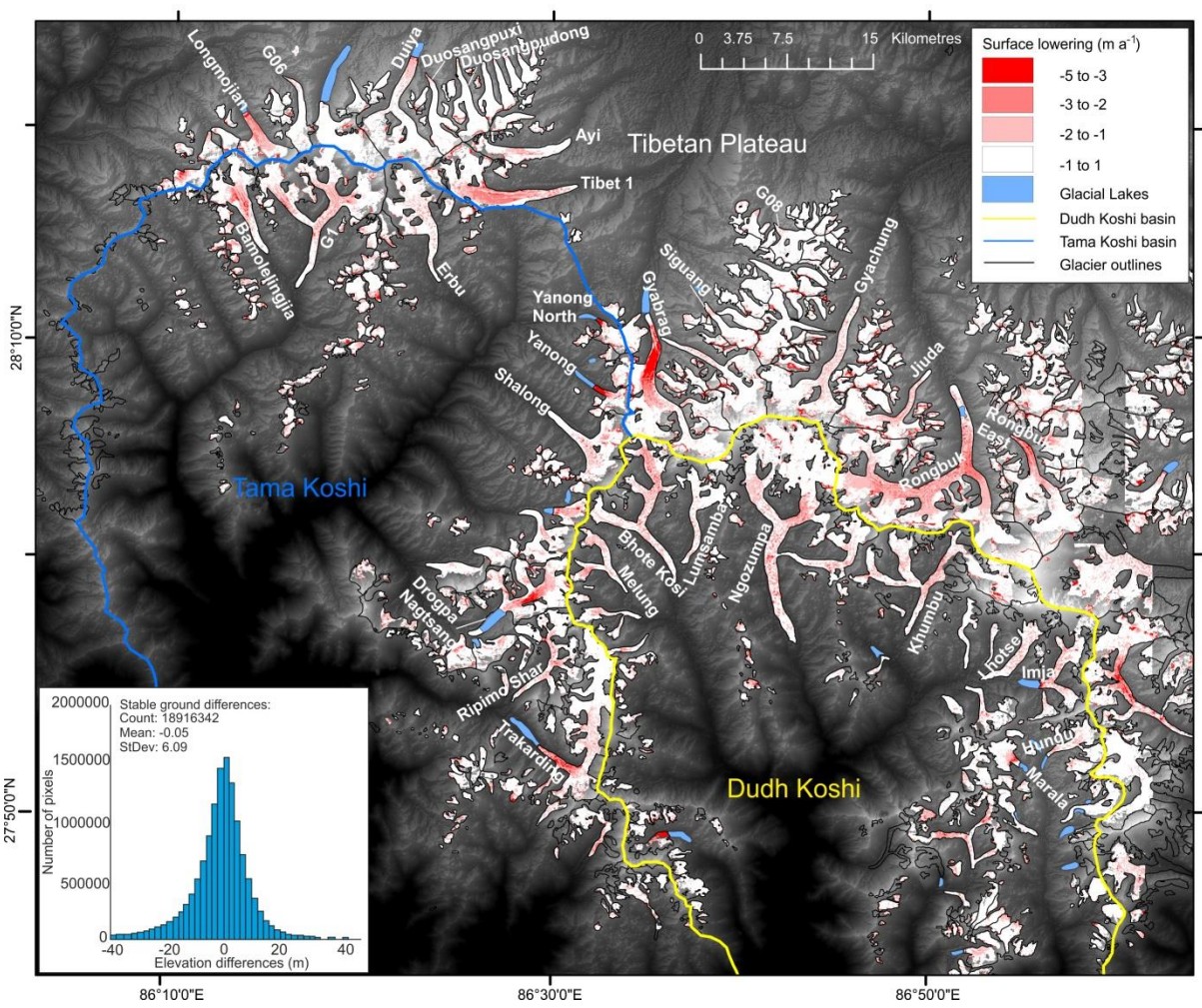

Figure 2. Glacier surface elevation change over the study area between 2000 and 2014/15. Also shown is a summary of off-glacier terrain differences. Areas of no data show the ASTER GDEM underlay.

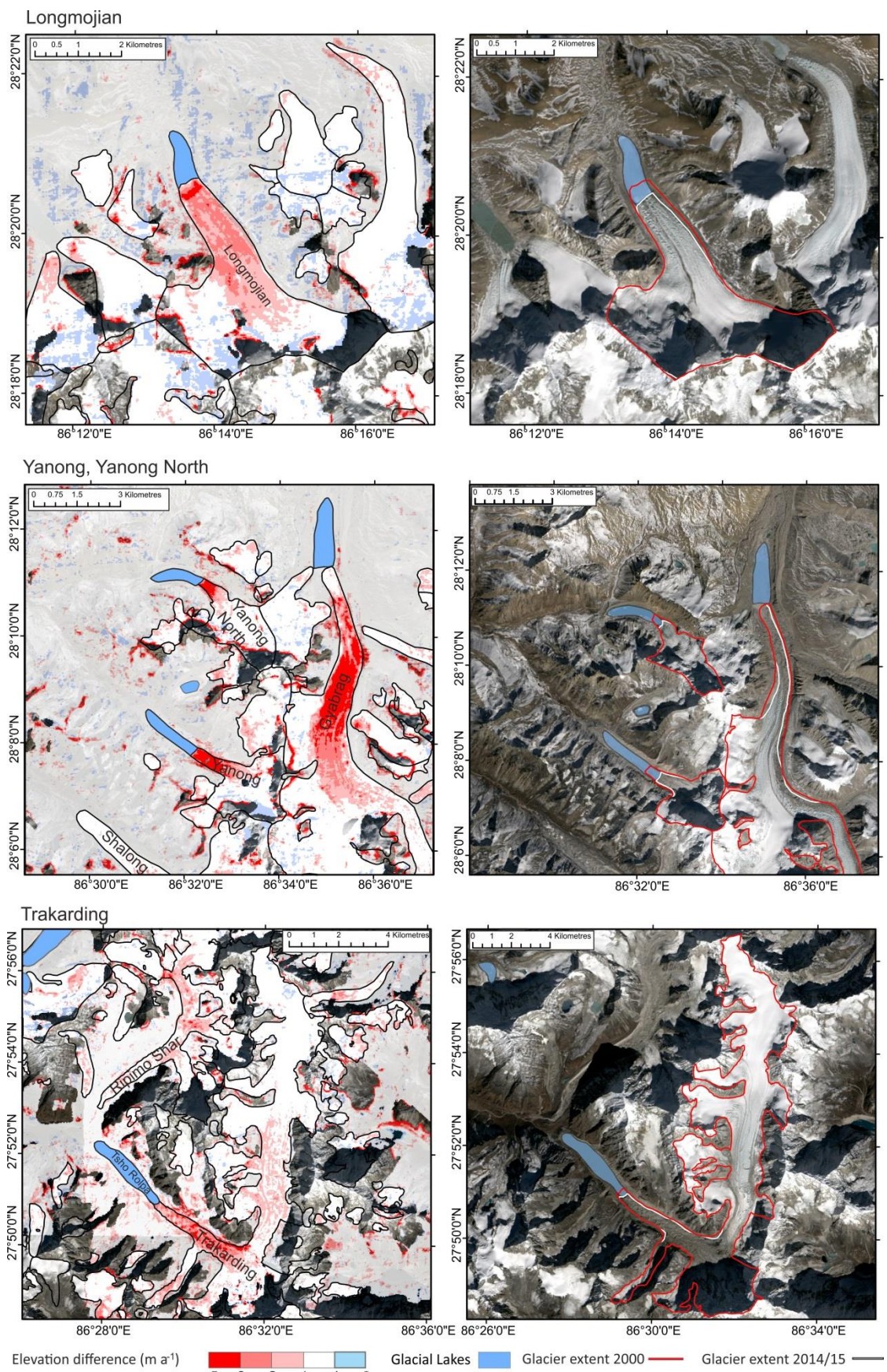

Figure 3. Examples of surface elevation change and total area change over the study period on lacustrine terminating glaciers. Semi-transparent, off-glacier differences are also shown.

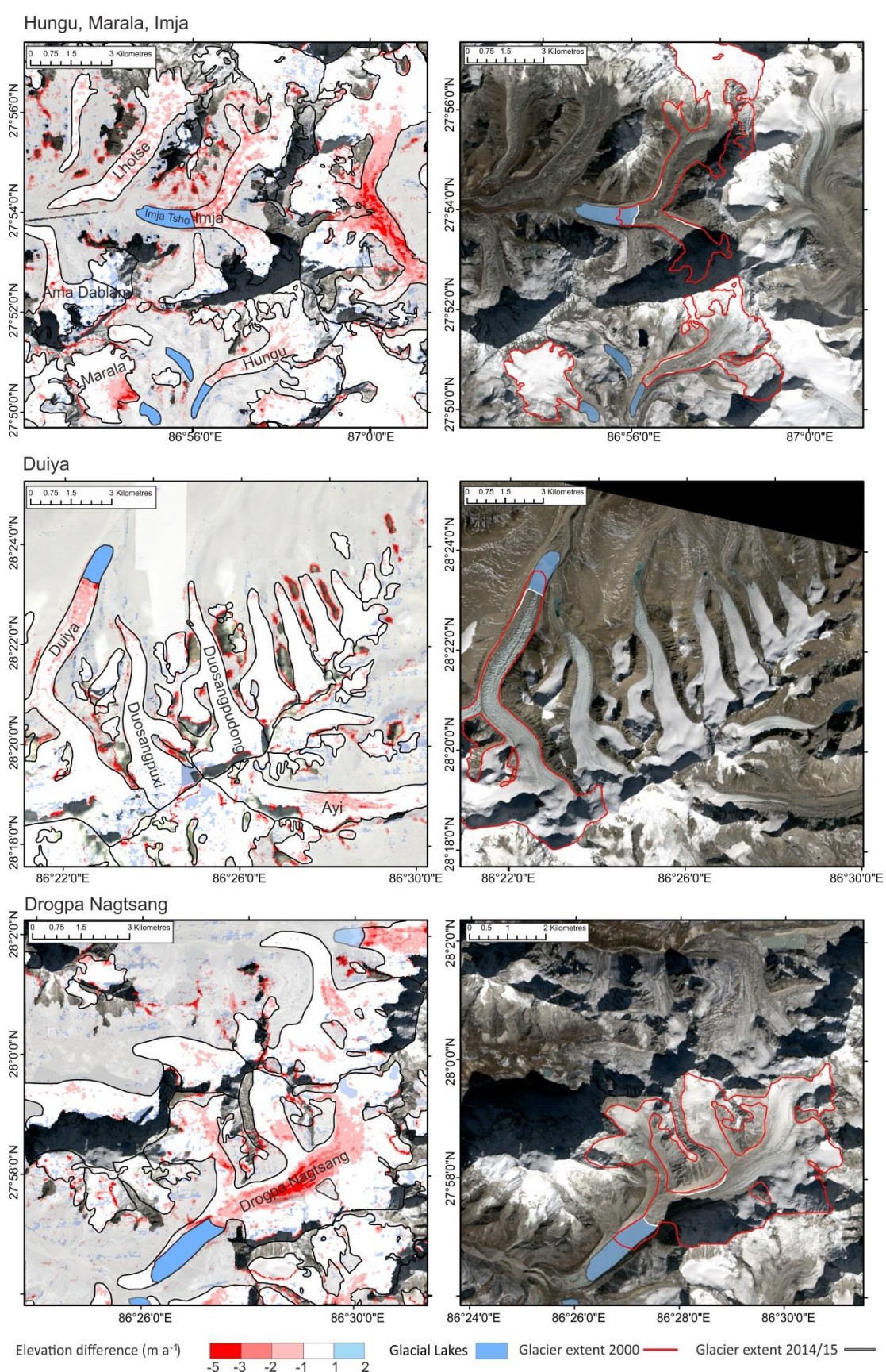

Figure 4. Further examples of glacier surface elevation change and total area change over the study period on lacustrine terminating glaciers. Semi-transparent, off-glacier differences are also shown.

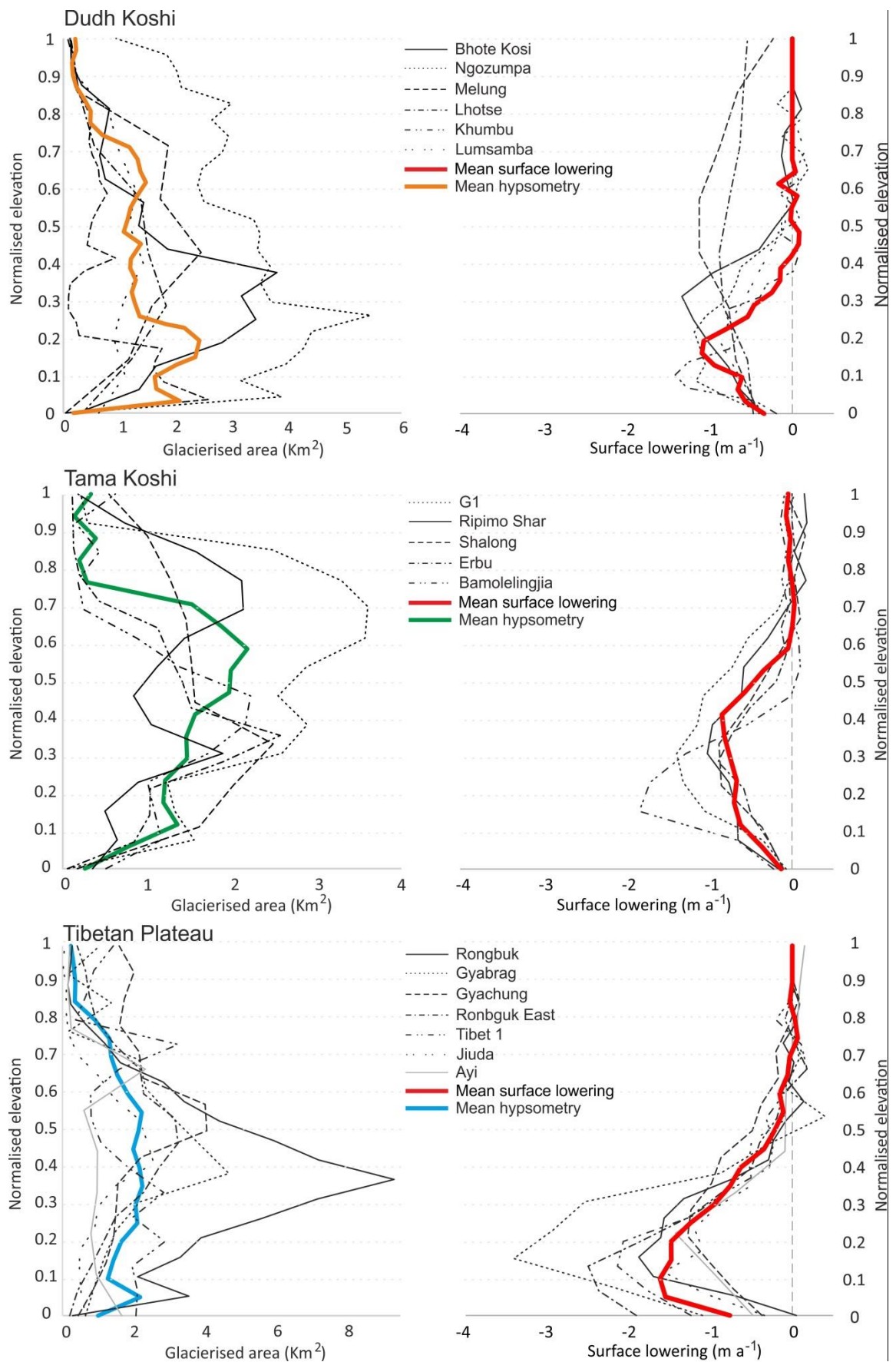

Figure 5. Surface elevation change and glacier hypsometry curves for all land terminating glaciers in the three different catchments of the study area.

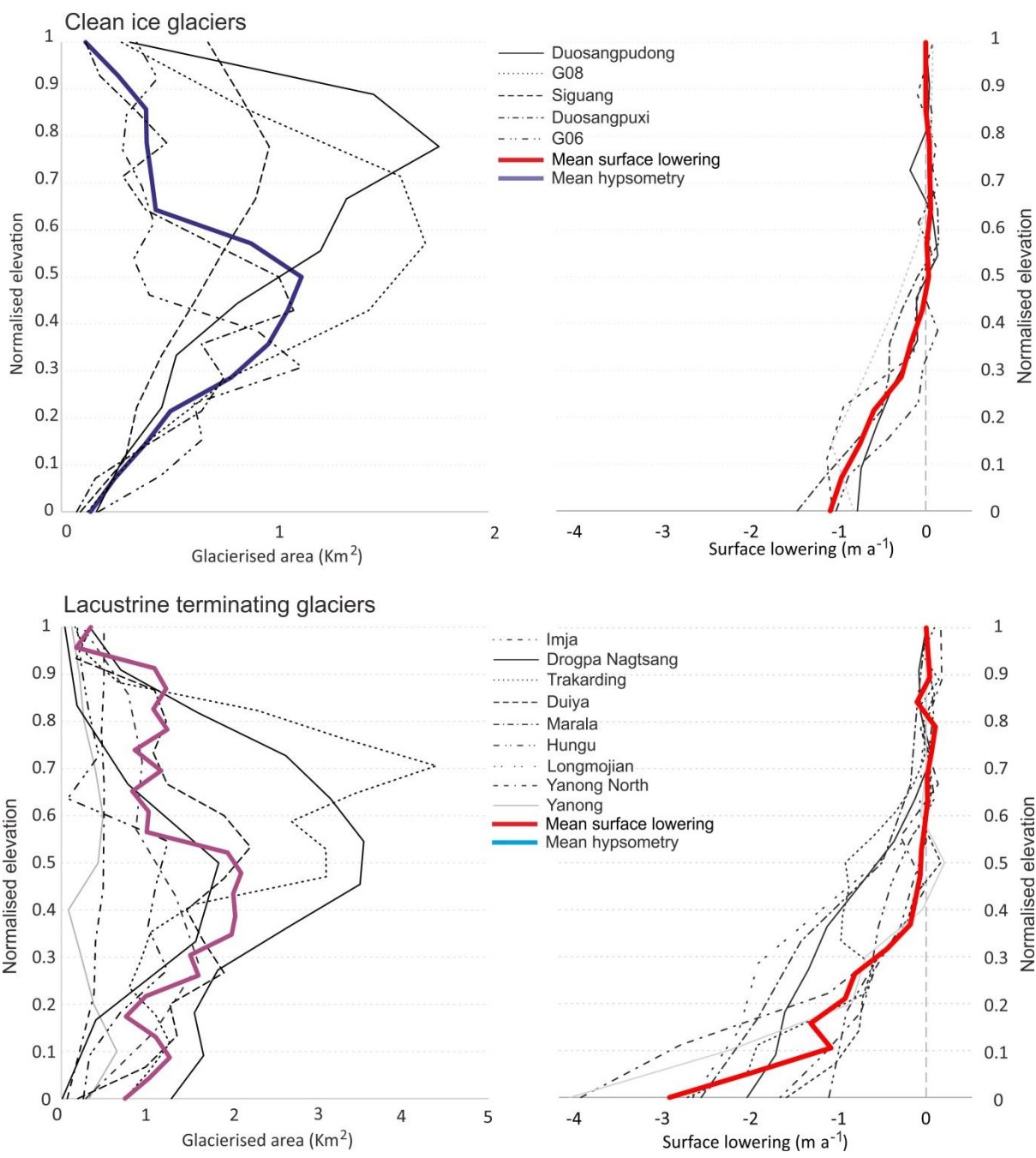

Figure 6. Surface lowering and glacier hypsometry curves for clean ice and lacustrine terminating glaciers in the study area.

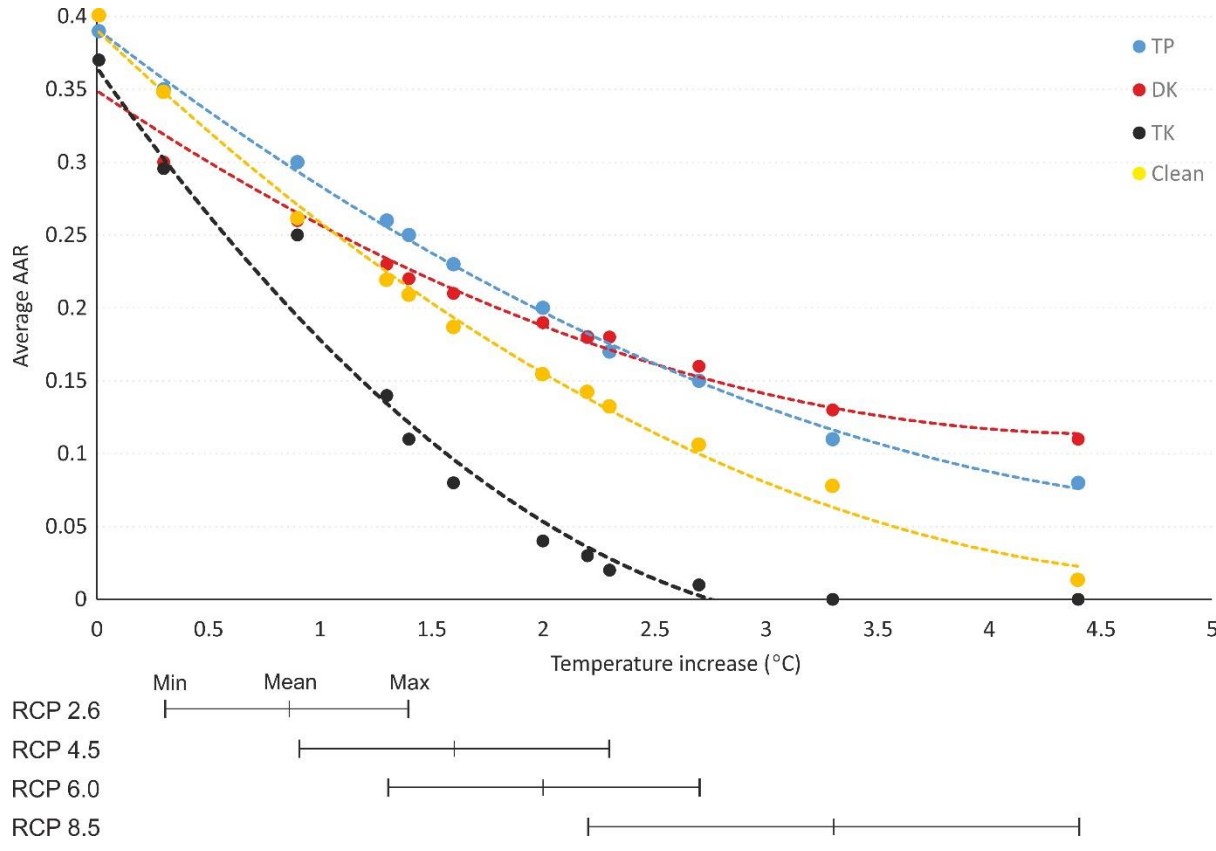

Figure 7. Projected AARs (averaged across each catchment) based on different scenarios of temperature increase relative to the present day and accompanying ELA rise. Temperature rise scenarios have been used from the IPCC AR5 Working Group report. TP- Tibetan Plateau; DK- Dudh Koshi; TK- Tama Koshi; Clean-Clean ice glaciers. Each point represents a projected AAR given minimum, mean or maximum temperature rise under each RCP scenario.