# Peer review of "Spatial variability in mass loss of glaciers in the Everest region, central Himalaya, between 2000 and 2015"

_The Cryosphere, 2016_

## Referee Comment (RC1) · J.M. Shea (Referee) · 23 Jun 2016

Review - King et al., TC-2016-99

King et al. present an analysis of glacier surface lowering data for 31 large glaciers in three catchments in the Everest region of the Himalaya. They examine the role of lakes in the observed spatial variabilty, compare surface elevation change profiles with the conceptual model given by Benn et al. (2012), suggest potential future changes based on equilibrium line altitude and climate change assumptions, and conduct an analysis of glacier hypsometry.

The topic is relevant and interesting, and the paper is generally well-written, but there

are a number of substantial issues that must be addressed. I provide general and specific comments below.

GENERAL COMMENTS

1) The title of the paper and section 3.6 suggest that glacier "mass change" is examined. However mass change estimates cannot be calculated for only part of a glacier (e.g. the ablation zone) using the geodetic approach! Emergence velocity somehow needs to be taken into account. Fortunately, the authors have also neglected to show or discuss mass changes, so I would suggest that section 3.6 be removed and the title changed.

2) As a short follow-up to point 1, the authors should provide the caveat that emergence velocities will affect the observed surface lowering rates, though this cannot be quantified (or can it?).

3) The authors focus on the largest glaciers within the respective catchments, and justify this decision by saying that these glaciers 'provide the greatest volume of meltwater to downstream areas'. I see two problems with this: first, this justification is unreferenced, and possibly incorrect as the largest glaciers will likely also be debris-covered and have the lowest melt rates. Second, the melt rates of smaller glaciers are also of significant interest, and excluding them may result in biased average lowering rates.

4) Equilibrium line altitudes are not discussed until the Results section (4.2.3) but this is a very important part of the overall approach used (i.e. surface lowering is only considered for areas below the ELA). A section needs to be added to the methods describing how the ELA is determined, and how are future ELAs calculated. Section 5.4.1 is rather slim on details.

5) The range of temperature projections (+0.9 to +2.3C) appears to be a global mean, though this is not defined or justified, and higher emission scenarios show higher increases (+3.7C by 2100 for RCP8.5; Collins et al., 2013). Also, temperatures in the

Tibetan Plateau and Himalayas are expected to increase at a higher rate (e.g. Rangwala et al., 2013). Potential increases in freezing level have been examined by other authors (Shea et al., 2015; Viste and Sorteberg, 2015)

5) I am confused by the 'highlighted glaciers' and the presentation of the results, because the classes can be mixed. For example: Figure 5 shows all land terminating glaciers. Figure 6 shows clean Tibetan Plateau glaciers (top) and lacustrine terminating glaciers (bottom). Are some clean TP glaciers land-terminating? Are they counted in both graphs? Perhaps a more rigourous classification by basin would be useful to highlight the differences by glacier type and by region (e.g. Dudh Koshi -> land-terminating -> clean ice).

6) Formulas in the error analysis need to be presented correctly, and suitable symbols applied, see specific comments below.

SPECIFIC COMMENTS

Abstract: somewhere in here the time period for the analysis should be defined.

P1L13: what kind of glacial lakes? Be specific.

P1L18: 'Average surface lowering rates...'

P1L21: what is deep water calving?

P1L23: 'area', not volume...

P1L26: 'respectively' is missing somewhere

P2L2: The area and number of glaciers refers to the Himalayas, the Hindu Kush, and the Karakoram (not just the Himalayas).

P2L17: 'more stable' in the eastern Himalayas is incorrect. The greatest rates of surface lowering are observed in eastern Himalayas (Kaab et al., 2015).

P3L2-3: 'and it thus remains untested' is superfluous.

P3L20: clarify are these 40 largest glaciers \*partially\* debris-covered?

P4L4-7: There are observed and modelled ELA data from Wagnon et al., (2013) and Shea et al., (2015), respectively.

P4L19:what is 'non-void filled'?

P5L10-15: do you do any comparison between the ASTER and SETSM DEMs on stable ground?

P5L20-25 : Just to clarify, you take the GLIMS glacier extents, and modify them for 2000 and and 2014 extents based on Landsat imagery. And it should be mentioned here that you use the 2000 and 2014 extents to calculate area changes.

P7L9: Gardelle et al (2013) add a 50

P7L14: The graph shown in the Supplementary Information could be placed in the main text, but the caption needs to be improved as it is not clear what is being presented.

Eq.1 and below: I'd suggest using $\sigma_{stable}$ for standard deviation. And the root symbol should be over the whole term: $\sqrt{n_{diff}}$.

P8L1-10: Fix terms in the text: subscripts and italics are missing or inconsistent Eq. 2: italize $n_{diff}$, $n_{tot}$, $PS$

P8L7-8: this sentence is unclear. What value of d was used in this study?

Eq.3: root symbol needs to be over the whole expression: $\sqrt{SE^2 + MED^2}$, and I would suggest using $\overline{dZ_{stable}}$ for mean elevation differences (MED). MED looks a lot like median...

P9L3: "...the MEDIAN elevation...'

Section 3.6: suggest removing completely.

P9L25: report lowering rates with negative sign (e.g. -0.80 +/- 0.35) to be consistent with Table 4.

P10L1-3: 'Mass loss' should be 'surface lowering' here, and don't rates increase downglacier in all cases (not only lake-terminating glaciers)?

P10L2: surface lowering rates right at glacier termini are always negligible because there is limited ice available for melt.

P10L24: Refer to Figure 5 here.

P11L13: 'patterns', not 'scenarios'

P11L13-25: provide rates of area change per year for comparison with other studies?

P11L19: This is where the more rigorous classification would be useful. E.g.: Land-terminating clean vs. land-terminating debris-covered?

P12L13-20: Suggest moving this section to methods and adding more details on how current and future ELAs are determined.

P12L16-17: This phrasing is a bit awkward. It seems like you are trying to say that the approximated ELAs give an AAR of 0.37 in the Dudh Kosi catchment. (AAR = Accumulation Area/Total Area)

P13L3: Though around 80

P13L18-25: Some skepticism might be warranted when referencing the snow line altitude shifts given by Thakuri et al., (2014): these are based on single-image delineations of transient snowlines, and in the Himalayas these do not remain constant at the end of the summer season.

P14 Sec5.2: "surface lowering", not "mass loss"

P14L26-27: For lake terminating glaciers its complicated, but for land-terminating glaciers thinning should reduce the driving stresses and lead to decreased glacier velocities (e.g. Berthier and Vincent, 2012; Haritashya et al., 2015)

P15L25: the sensitivity of Dudh Kosi glaciers to future ELA changes based on its

hyspometry was noted previously by Shea et al. (2015)

Table 3: separate columns for means and standard deviations

Figure 1: Add the imagery extents here?

Figure 2: text labels with glacier names are impossible to read. Also, is it possible to show the data voids in the DEM differencing?

Figures 3 and 4: Why are glacier extents shown in 2014 (left panel) not also present in 2000 extents?

Figure 5 and 6: Larger fonts required! Caption should point out that surface lowering curves are on the right and hypsometry on the left. Maybe show hypsometry as relative (% of total area) as opposed to absolute? and show surface lowering rates as boxplots by elevation band?

Figure 6: Why is approximate ELA only shown on top panel? What about project future ELAs?

REFERENCES

Berthier, E. and Vincent, C., 2012. Relative contribution of surface mass-balance and ice-flux changes to the accelerated thinning of Mer de Glace, French Alps, over 1979–2008. Journal of Glaciology, 58(209), pp.501-512.

Haritashya, U.K., Pleasants, M.S. and Copland, L., 2015. Assessment of the Evolution in Velocity of Two Debris‐Covered Valley Glaciers in Nepal and New Zealand. Geografiska Annaler: Series A, Physical Geography, 97(4), pp.737-751.

Kääb, A., Treichler, D., Nuth, C. and Berthier, E., 2015. Brief Communication: Contending estimates of 2003–2008 glacier mass balance over the Pamir–Karakoram–Himalaya. The Cryosphere, 9(2), pp.557-564.

Rangwala, I., Sinsky, E. and Miller, J.R., 2013. Amplified warming projections for high

altitude regions of the northern hemisphere mid-latitudes from CMIP5 models. Environmental Research Letters, 8(2), p.024040.

Shea, J.M., Immerzeel, W.W., Wagnon, P., Vincent, C. and Bajracharya, S., 2015. Modelling glacier change in the Everest region, Nepal Himalaya. The Cryosphere, 9(3), pp.1105-1128.

Viste, E. and Sorteberg, A., 2015. Snowfall in the Himalayas: an uncertain future from a little-known past. The Cryosphere, 9(3), pp.1147-1167.

Wagnon, P., Vincent, C., Arnaud, Y., Berthier, E., Vuillermoz, E., Gruber, S., Ménégoz, M., Gilbert, A., Dumont, M., Shea, J.M. and Stumm, D., 2013. Seasonal and annual mass balances of Mera and Pokalde glaciers (Nepal Himalaya) since 2007. Cryosphere, 7(6), pp.1769-1786.

---

## Referee Comment (RC2) · Anonymous Referee #2 · 8 Jul 2016

Review of King et al., The Cryosphere, July 2016

In their paper, King and co-authors measured glacier surface elevation changes in the Everest area between Feb 2000 and 2014/2015 using remotely-sensed DEMs and studied the spatial pattern of elevation change in the ablation area of glaciers. Rate of surface elevation changes are compared between three different basins and also interpreted considering the glacier type. A special focus is drawn on the influence of proglacial lakes on glacier wastage. Sensitivity of these glaciers to the future projected warming is discussed by examining their hypsometry.

This study is not ready for publication. At several places in the manuscript (MS), there are some misconceptions, especially a problematic confusion between rate of elevation changes (dh/dt, what the authors measured) and ablation rates (i.e. surface mass balance). The two variables are different and cannot be compared as the authors do (e.g., in their comparison of their data to Benn's model). Some of the conclusions are not really supported by the data themselves (e.g., statistically significant difference between the 3 main basins? Attribution of the thinning to climate drivers). In the end, the author is also left without a real take-home message. The limited implications of the present study are partly due to the fact that the authors decided not to compute glacier-wide mass balances. This is probably a reasonable choice given the lack of knowledge of SRTM penetration depth in the upper reaches of the Everest area glacier but still it makes the interpretation of the observations very difficult because rate of elevation changes for a portion of the glacier are not equivalent to surface mass balance, they also depend on ice dynamics. In the end, the reader is left with the question: "what did we learn in this study that we did not before?"

**General comments**

One major issue is that authors draw some conclusions between glaciers in three different basins or with different terminus type from **dh/dt measured in the ablation area only**. Such comparisons carry little significance because these generally small differences in dh/dt the ablation areas could easily be compensated by differences of opposite signed in the accumulation areas. Hence one cannot conclude unambiguously that the mass loss is larger for such basin compared to such basin or for this type of glacier terminus. Although the differences are often not statistically strongly different. A comparison of the different rate of elevation changes with altitude (Figure 7) is also partly misleading because the elevation range of the compared glaciers is really different (due to different climate setting). A solution could be for example to normalize the elevation range has was done in (Arendt et al., 2006), among others.

All along the text and in the tables, the authors provide many details about individual glaciers such that it is **difficult to extract the big picture**, the take-home message. A table summarizing mean dh/dt in the ablation area average by large basin and glacier type (area loss / mean dh/dt for the ablation area) should be added. See also the specific comment below were I suggest moving Table 4 and 5 in to the supplement and replace them with synthetic figures.

**Errors on dh/dt**. One problem with the metric which is used currently is that it does not take into account the size of the averaging area, i.e., the error on the rate of elevation change is the same for a 0.1 km² and a 80 km² ablation area. This is obviously not realistic.

The discussion of the **climate drivers of this glacier thinning in the ablation area is currently very weak**. For example (13.18), the authors make a weak statement about climate trend during 2000-2015, also the period of the dh/dt measurements. Even if T,P were stable (no trend) during the study period, a strong thinning rate could still be observed between 2000-2015 if, for example, a step-like warming (or change in precipitation) occurred in the years preceding the study period. In other words, the glacier disequilibrium to the climate depend a lot on what happened before the study period and not only on the climate trend during the study period.

Figure 7 and the related text. **It is not acceptable to compare dh/dt and mass balance**. They are simply not glaciologically comparable. The statement 17.7 that " The ablation gradients shown by lacustrine terminating glaciers are also very similar to regime 3 of Benn et al. (2012)" is a clear illustration of this confusion. Authors seem to believe that they measure ablation gradient when they measured gradient in dh/dt in the ablation area. They entirely neglect the role of emergence velocity which is not physically realistic.

**More specific comments (some still substantial)**

Title needs to include "ablation areas"

1.17. not all these glaciers are flowing southward (the basins are located southward of the main ridge)

1.18. a negative lowering rate suggest a thickening of the glacier (double negative). Either authors should change the sign or used "rate of surface elevation changes".

1.19. "small lakes". Are these supraglacial? Proglacial?

1.24. Providing the present AAR and how it will potentially change in the future due to the rise of the ELA is probably a more useful and conventional metric to illustrate this hypsometric sensitivity of the different basins.

1.28. I miss a sentence at the end of the abstract indicating the implications of this study. A sort of take-home message for the readers. To answer this question: What did we learn here that we did not before? A statement well-supported by the data that will make other researchers cite the present paper.

2.13. "ice melt from the region may contribute 8.7–17.6 mm of sea level rise". Glaciers melt seasonally even if they are in balance and even if they do not contribute to sea level rise.... Replace by "glacier imbalance". Melt is not synonym of mass loss.

2.14. Authors need to stress that these estimates are for the first decade of the 21st century only.

2.16. The study by (Kääb et al., 2015) suggest strongly negative mass balance in the southeast Tibetan plateau. Update.

2.18. Kapnick et al. 2015 was a welcome modelling effort to understand the cause of the anomaly, but this is not among the studies that documented the Karakoram anomaly. See rather (Bolch et al., 2012; Gardelle et al., 2012; Hewitt, 2005; Rankl and Braun, 2016)

2.18. Future hydrology. Is the debate relate settled? This need explanation or should be deleted. Because at least in the next decades, more negative glacier mass balance means more water in the rivers...

2.26. Description of Benn et al. 2012 conceptual model. Why is this included in the paragraph about measuring glacier mass loss. Separate paragraph needed.

3.6. Already here the reader starts to wonder why only mass loss in the ablation area is observed. This should be better explained/justified right away.

3.14. is it really the majority? I guess in term of area yes but in terms of numbers I am not so sure (there are many small glaciers...)

3.18. do the authors mean "beneath steep cliffs"? Improve terminology. Khumbu glacier also sit beneath the Everest "massif" and has a wide and flat accumulation area of several km²....

3.23 there are not so many studies measuring acceleration in the rate of surface lowering so authors could probably list them. Nuimura et al. 2012 is the other one I can think of.

4.4 Table 2 in Gardelle et al., 2013 list some ELA values from three different studies. So there is more information about ELA than what the present text suggests.

4.21 if the authors mention the two SAR systems, then they need to tell which one of the two was used to generate the version 3.0 DEM they are using. Readers are confused otherwise.

4.27. images are listed in Table 1, not Table 2. Further, these images are acquired at very different time of the year which raise the issue of how seasonal variation in height have been accounted for in the study. If not correction was applied, this needs to be well-justified and the uncertainties quantified.

6.1. Can the authors better justified the need to work on a selection of glaciers and not work on each individual glacier? Rational for that?

6.26. Although the spatial variability of the geoid height must be rather small at the scale of the DEMs processed here, it is not acceptable to compare DEMs defined above different datum. There are gridded versions of the EGM96 geoid that can easily be used to correct for the elevation difference. Conversion from geoid to ellipsoid (and vice versa) is also a built-in tool in many GIS software (including in the open source gdal libraries).

6.28. "first order trends". More details needed. Are these corrections estimated using all ice free pixels? How do the authors take into account large outliers that always occur in DEMs from satellite stereo imagery and that may contaminate their corrections?

7.7. "penetration corrections are rarely applied". Is this a good justification? Not really. Strongly biased estimates of geodetic mass balances have been published in the past due to the lack of correction of this systematic effect. See for example (Fischer et al., 2015) that demonstrated that the geodetic mass balances from (Paul and Haeberli, 2008) were strongly biased negatively and (Kääb et al., 2015) & (Barundun et al., 2015) that have shown that (Gardelle et al., 2013) Pamir mass balance estimates are likely biased toward positive values for the same reasons. This is a systematic source of errors and as such it cannot be treated by simply adding it to the error bars. The poor knowledge of the SRTM penetration depth is maybe the reason why the authors have limited their analysis to the ablation area. If this is the case, this needs to be explained/justified. But as said in my general comments, this is really limit the implications of the study.

7.15. Such an elevation dependent correction cannot be applied to one DEM alone but to the elevation difference between two DEMs.

7.20. unclear what the authors mean by "real topographic change on the stable terrain".

8.7. what matters is not the spatial autocorrelation in each DEM but the autocorrelation in the map of elevation difference. So only one auto-correlation distance should be reported.

8.9. can the authors explain why a MED remain after all the adjustments? I would have expect the mean difference to be 0 "by construction". Did the authors examined the overlapping areas of the WV DEMs as a verification of the DEM adjustment?

8.11. "independent" of what?

8.17. Can the authors confirm that in table 3, the standard error (and not "e", the elevation change uncertainty) is listed. I find it extremely strange that the last column of Table 3 (labelled "st error") is always so close to the value of the remaining mean elevation difference as listed in the "post correction" column of the same table (Table 3). The similarity is unexpected because one column is in m and the other in m/yr. I think authors need to double check this and clarify their terminology.

8.22. The Landsat images are used to refine the outlines not to extract the hypsometry, as the authors explained earlier in the MS. Be brief here and just tell that the 100-m hypsometry was extracted from the SRTM (?) DEM and the glacier outlines. Void filled DEM or not?

8.26. "glacier area change". Not relevant in the hypsometry section.

9.9. "we did not generate mass balance estimates". Do the authors mean glacier-wide mass balance estimates? The lack of knowledge of the SRTM penetration depth is another good reason to avoid this. Still I find this disappointing, It would have allowed a direct comparison to other studies and better comparison of individual glaciers/basins.

9.18. Here I am not sure I understood what the authors exactly did. Do they mean that they only summed mass loss occurring upstream of the 2014/2015 calving front? Why not taking into account at least aerial mass loss (i.e. above the lake level) for the area between the 2000 and 2015 calving front?

10.1. to draw such a conclusion "The presence of a glacial lake altered the gradient of surface lowering over glacier surfaces" authors need to compute the dh/dt gradient and compare them to support their statement. Is it the gradient with altitude? With distance to the terminus? Statement not demonstrated in the paper.

10.23. "mean" over what? A 100-m altitude band centred around 5300 m asl? Clarify.

10.6. The mean value of 2.04 m/yr is for which catchment? All merged?

11.13. What are these two scenarios? Unclear. Also what is the meaning of "scenario" in this context?

11.17. Why not providing the same % for lake-terminating glacier.

12.1. the basin-wide hypsometries should be added to Figure 7 to be compared easily to dh/dt also averaged by basin. And figure 5-6 would keep only individual glaciers (no basin-wide average).

12.13. "The altitude at which surface lowering curves approach zero is a good indicator of the ELA of glaciers". This statement is surprising. I checked the Nuth et al., 2007 reference and indeed found the following sentence : "The hypsometric (area–altitude) distribution for Brøggerhalvøya/Oscar II Land is greatest between 250 and 550 m a.s.l., with the 54 year average ELA (position where the elevation change curve approaches zero) at 350 m (Fig. 5a)." So there is no reference or data to support this statement in Nuth et al. This is a strong approximation that suggest similarities between null dh/dt and null mass balance. Rate of elevation change and mass balance are not the same quantities, I do not see how you can do such an hypothesis.

12.16-20. Complicate wording! Do they authors mean that the AAR is 37%, 36% and 40% in the different catchments?

12.23. The sensitivity of these results to the uncertainties in the ELA need to be quantified.

13.4. Regarding sensitivity to temperature (and contrast between different regions), the studies by Fujita and Sakai (Fujita, 2008; Sakai et al., 2015) are better references. (Rupper et al., 2012) is based on very thin data and only examined Bhutanese glaciers so it is not the right reference to claim that the sensitivity is high in Nepal; By the way, high compared to what/where?

13.9. In addition to the quoted studies, (Wagnon et al., 2013) have described in detail the precipitation gradient with the Khumbu basin, from Lukla to the Pyramid station.

13.16. This statement is in contradiction to the general belief that glaciers in maritime climate (more humid) are more sensitive to temperature change than glaciers in a more continental climate. See for example (Hock et al., 2009). Without a full sensitivity analysis and without some glacier-wide mass balance measurements, I do not see how the authors can conclude to such statement. Unsupported by the data.

13.28. Again (like in 13.18.) a weak reasoning. Why would the rise in the snowline altitude be a proof of accumulation decrease? How can the authors separate this way the respective role of temperature and precipitation trends? (this is even more complex in Nepal than in other mountain ranges because accumulation and ablation season are simultaneous)

13.29. "since the 1970s". Authors need to give the exact time period over which the decrease has been observed (i.e. provide the end year).

14.3. Again a poor reasoning. A rise in temperature is sufficient to explain a decline is snow cover (and the time period of 9 years is really short to draw conclusions). How can the authors draw conclusions about accumulation rates just based on this proxy?

14.10 Authors quote a lengthy time series of DEMs but provide the result for only a five time period... no need for "lengthy" or then authors should provide the results over the long time spam.

14.11. "0.79 m/yr and 0.84 m/yr" can only be compared if error bars are provided. I doubt the authors can conclude here to a significant difference between these two highly similar values.

14.12. Comparison to the thinning rate of (Gardelle et al., 2013). Does this bring something to the discussion? Is it for exactly the same area and the same altitude range?

14.18. "given" missing I think. The entire sentence needs improvement in fact.

15.9-11. Understatement. I do not understand how these statements are related to the rest of the paragraph. What do the authors want to conclude here? Do they want to explain why the dh/dt is not as negative for Imja? Make the logics easier to capture by the reader.

15.18. Can the authors explains what is this "similar surface lowering pattern". It has not been presented in the result section. How can they be certain that this is due to enhanced ablation at cliffs/ponds rather than advection by ice flow of an heterogeneous surface topography?

16.4. "earlier epochs". Provide year of estimates.

16.15. Unclear wording. Why not simply mentioning the reduction in the AAR due to the ELA rise (this would be the theoretical reduction of course because this would be based on the present-day hypsometry not considering the future area loss, mainly at low elevations)

16.24. Again a very strange structure for this sentence: change are described for Dudh Koshi and TP glaciers and the sentence finishes with a conclusion for ... Tama Koshi basin. Improve logics.

17.21. The statement in the conclusion that there is decreased ice influx from accumulation zone comes from nowhere. Was never discussed earlier in the MS, never shown by the data.

17.24. Here and before. How do the authors calculate the uncertainty for their basin-wide average? Must not be simply the mean of the individual glacier uncertainties.

17.27. "We suggest that the across-range contrast in annual precipitation total may have caused greater ice loss on the north flowing glaciers ". Are they different enough statistically (compare 0.80 and 0.95 m/yr) to deserve an explanation? See also my general comment about the weak attribution to climate drivers.

18.1. Add "in their ablation area"

18.13. Again, same as above (see general comments). Authors did not measure ablation gradients!!! They maybe measure dh/dt gradient (with altitude? distance?). But no plot show these dh/dt gradient data.

Table 2. Authors could draw an horizontal bar to clearly separate the different catchment.

Table 4 (like Table 2 and Table 5) are not a really useful way to present the data. If the authors think that the list of glaciers is really important (I am not sure it is) then these tables should me moved as appendix or supplement. A much more concise way to present these numbers (in a figure rather than a table) should be preferred. For example a whisker plot showing the mean/median, range of values etc... for each catchment and each glacier type would condense the info and then, the corresponding text could be shorten also.

Figure 1: Authors needs to indicate in the caption what is the background image and the source of the inventory.

Figure 2: it would be good to show the off glacier dh/dt at least in a figure in the Supplement.

References cited in my review

Arendt, A., Echelmeyer, K., Harrison, W., Lingle, C., Zirnheld, S., Valentine, V., Ritchie, B. and Druckenmiller, M.: Updated estimates of glacier volume changes in the western Chugach Mountains, Alaska, and a comparison of regional extrapolation methods, J Geophys Res-Earth, 111(F3), 2006.

Barundun, M., Huss, M., Sold, L., Farinotti, D., Azisov, E., Salzmann, N., Usubaliev, R., Merkushkin, A. and Hoelzle, M.: Re-analysis of seasonal mass balance at Abramov glacier 1968–2014, Journal of Glaciology, 61(230), 1103–1117, doi:10.3189/2015JoG14J239, 2015.

Bolch, T., Kulkarni, A., Kääb, A., Huggel, C., Paul, F., Cogley, J. G., Frey, H., Kargel, J. S., Fujita, K., Scheel, M., Bajracharya, S. and Stoffel, M.: The State and Fate of Himalayan Glaciers, Science, 336(6079), 310–314, 2012.

Fischer, M., Huss, M. and Hoelzle, M.: Surface elevation and mass changes of all Swiss glaciers 1980–2010, The Cryosphere, 9(2), 525–540, doi:10.5194/tc-9-525-2015, 2015.

Fujita, K.: Effect of precipitation seasonality on climatic sensitivity of glacier mass balance, Earth and Planetary Science Letters, 276, 14–19, doi:10.1016/j.epsl.2008.08.028, 2008.

Gardelle, J., Berthier, E. and Arnaud, Y.: Slight mass gain of Karakorum glaciers in the early 21st century, Nat Geosci, 5(5), 322–325, doi:10.1038/ngeo1450, 2012.

Gardelle, J., Berthier, E., Arnaud, Y. and Kääb, A.: Region-wide glacier mass balances over the Pamir-Karakoram-Himalaya during 1999–2011, The Cryosphere, 7, 1263–1286, doi:10.5194/tc-7-1263-2013, 2013.

Hewitt, K.: The Karakoram anomaly? Glacier expansion and the "elevation effect," Karakoram Himalaya, Mountain Research and Development, 25(4), 332–340, 2005.

Hock, R., de Woul, M., Radic, V. and Dyurgerov, M.: Mountain glaciers and ice caps around Antarctica make a large sea-level rise contribution, Geophys Res Lett, 36, 2009.

Kääb, A., Treichler, D., Nuth, C. and Berthier, E.: Brief Communication: Contending estimates of 2003–2008 glacier mass balance over the Pamir–Karakoram–Himalaya, The Cryosphere, 9(2), 557–564, doi:10.5194/tc-9-557-2015, 2015.

Paul, F. and Haeberli, W.: Spatial variability of glacier elevation changes in the Swiss Alps obtained from two digital elevation models, Geophysical Research Letters, 35, L21502, 2008.

Rankl, M. and Braun, M.: Glacier elevation and mass changes over the central Karakoram region estimated from TanDEM-X and SRTM/X-SAR digital elevation models, Annals of Glaciology, 51(71), 273–280, doi:10.3189/2016AoG71A024, 2016.

Rupper, S., Schaefer, J. M., Burgener, L. K., Koenig, L. S., Tsering, K. and Cook, E. R.: Sensitivity and response of Bhutanese glaciers to atmospheric warming, Geophysical Research Letters, 39(19), n/a-n/a, doi:10.1029/2012GL053010, 2012.

Sakai, A., Nuimura, T., Fujita, K., Takenaka, S., Nagai, H. and Lamsal, D.: Climate regime of Asian glaciers revealed by GAMDAM glacier inventory, The Cryosphere, 9(3), 865–880, doi:10.5194/tc-9-865-2015, 2015.

Wagnon, P., Vincent, C., Arnaud, Y., Berthier, E., Vuillermoz, E., Gruber, S., Menegoz, M., Gilbert, A., Dumont, M., Shea, J. M., Stumm, D. and Pokhrel, B. K.: Seasonal and annual mass balances of Mera and Pokalde glaciers (Nepal Himalaya) since 2007, Cryosphere, 7(6), 1769–1786, doi:10.5194/tc-7-1769-2013, 2013.

---

## Author Comment (AC1) · 9 Aug 2016

We are grateful for the thorough and constructive comments of Joseph Shea and an anonymous reviewer regarding our recent submission to The Cryosphere Discussions. The reviewers have highlighted a number of issues with the manuscript in its current form, as well as with our approach to data analysis. We agree with many of their points, and are pleased to be able to incorporate them into a revised version of the manuscript. Notably, with some relatively minor additional data processing, we have now calculated glacier mass balance estimates; a change that addresses many of the points that the reviewers raised.

Here, we outline our approach to amending the manuscript in line with the comments of the reviewers. Figures that have changed or new figures that have been added can be found at the end of this document.

Reviewer 1- Joseph Shea

GENERAL COMMENTS

**1)      The title of the paper and section 3.6 suggest that glacier "mass change" is examined. However mass change estimates cannot be calculated for only part of a glacier (e.g. the ablation zone) using the geodetic approach! Fortunately, the authors have also neglected to show or discuss mass changes, so I would suggest that section 3.6 be removed and the title changed.**

The title of the work has been changed slightly to 'Spatial variability in mass loss of glaciers in the Everest region, central Himalaya, between 2000 and 2015' because we do now calculate mass balance estimates for our sample of glaciers. We have kept section 3.6 as a result because details on mass loss calculations and associated uncertainties are required.

**2)      As a short follow-up to point 1, the authors should provide the caveat that emergence velocities will affect the observed surface lowering rates, though this cannot be quantified (or can it?).**

We have acknowledged (P9, L15 in the new manuscript) that without up-to-date glacier surface velocity data and ice thickness measurements we cannot specifically quantify emergence and its contribution to our surface lowering data. Previous work (e.g. Quincey et al., 2009) has identified active vs inactive ice boundaries for a number of the glaciers we include in our analyses, thus compressive flow and emergence is likely to occur, but we see no obvious evidence in our surface lowering data; unlike Immerzeel et al. (2015), who use DEMs of much higher spatial and temporal resolution.

**3)      The authors focus on the largest glaciers within the respective catchments, and justify this decision by saying that these glaciers 'provide the greatest volume of meltwater to downstream areas'. I see two problems with this: first, this justification is unreferenced, and possibly incorrect as the largest glaciers will likely also be debris-covered and have the lowest melt rates. Second, the melt rates of smaller glaciers are also of significant interest, and excluding them may result in biased average lowering rates.**

We have amended the text to say that these glaciers 'provide the greatest *potential* volume of meltwater to downstream areas' (P3, L28). Another justification for focusing on the large glaciers is that these are likely to be more negatively out of balance with climate, particularly the debris-covered tongues and lake-terminal glaciers, whereas small glaciers at high altitude would need a greater rise in air T (and ELA) before they lose mass.

Unfortunately, as the smallest glaciers in the area are typically found at higher altitudes and in complex topography, DEM coverage, particularly in the SRTM dataset, is lacking and we are not able to provide data for these areas.

**4)      Equilibrium line altitudes are not discussed until the Results section (4.2.3) but this is a very important part of the overall approach used (i.e. surface lowering is only considered for areas below the ELA). A section needs to be added to the methods describing how the ELA is determined, and how are future ELAs calculated. Section 5.4.1 is rather slim on details.**

Agreed. We have added a portion of text (P9, L18-25) to the methods section that specifically addresses how we estimate ELAs from surface lowering data, and how we calculate future prospective ELAs.

**5)      The range of temperature projections (+0.9 to +2.3C) appears to be a global mean, though this is not defined or justified, and higher emission scenarios show higher increases (+3.7C by 2100 for RCP8.5;**

**Collins et al., 2013). Also, temperatures in the Tibetan Plateau and Himalayas are expected to increase at a higher rate (e.g. Rangwala et al., 2013). Potential increases in freezing level have been examined by other authors (Shea et al., 2015; Viste and Sorteberg, 2015).**

The range of temperature projections are taken from Collins et al. (2013) for the tropics (including the monsoon influenced Himalaya) for CMPI5 RCP 4.5. Collins et al. (2013) estimate a minimum, mean and maximum dT of 0.9, 1.6 and 2.3 ºC by 2100 for this region under this scenario. We have calculated ELA rise and associated AARs for all RCP scenarios in the latest IPCC Working Group report (AR5), but only showed RCP 4.5 to allow for comparison with other studies that have used the same scenario (e.g. Shea et al., 2015; Rowan et al., 2015). In light of the reviewer's comment we have now included an additional figure showing estimates of catchment averaged AARs for all temperature rise scenarios (RCP 2.6, 4.5, 6.0 & 8.5; see Figure 7 at end of this document).

**6)    I am confused by the 'highlighted glaciers' and the presentation of the results, because the classes can be mixed. For example: Figure 5 shows all land terminating glaciers. Figure 6 shows clean Tibetan Plateau glaciers (top) and lacustrine terminating glaciers (bottom). Are some clean TP glaciers land-terminating? Are they counted in both graphs? Perhaps a more rigorous classification by basin would be useful to highlight the differences by glacier type and by region (e.g. Dudh Koshi -> land-terminating -> clean ice).**

Agreed in regard to the 'Highlighted glacier' section. This is slightly repetitive and confusing. We have removed this section and added a small amount of text to the study area section to point out the glaciers we treat as 'lacustrine terminating'.

Figure 5 shows surface lowering curves for land terminating, debris covered glaciers exclusively. The top panel of Figure 6 shows surface lowering curves for only land terminating, clean ice glaciers on the Tibetan Plateau. Likewise, the lower panel of Figure 6 is reserved for only lacustrine terminating glaciers (but from all three catchments, regardless of debris cover). Glaciers do not appear in multiple groups. We have amended figure captions to make sure that this is clear. We have not divided groups further as they can't be separated based on debris cover or terminus type.

**7)    Formulas in the error analysis need to be presented correctly, and suitable symbols applied, see specific comments below.**

Formulae have been formatted according to your suggestions. Thanks!

SPECIFIC COMMENTS

**JS comment: Abstract: somewhere in here the time period for the analysis should be defined.**

Agreed. Sentence in the abstract modified to 'We quantify mass loss rates over the period 2000-2015 for 32 glaciers…'

**JS comment: P1 L13: what kind of glacial lakes? Be specific.**

We have updated the text to describe the types of glacial lakes we cover with our analysis:

'…and specifically examine the role of 7 proglacial, and 2 supraglacial lakes in glacier mass change.'

**JS comment: P1 L18: 'Average surface lowering rates...'**

This part of the abstract has been largely re-written and the original text has been removed.

**JS comment: P1 L21: what is deep water calving?**

We did not intend to infer that 'deep water calving' is a specific phenomenon. Rather, we were discussing that as lake depth increases, calving rates are likely to increase, as shown by Benn et al. (2007- Earth-Science Reviews). We have amended the text to make this clearer:

'...and that rates of mass loss are likely to increase as glacial lakes expand and calving can occur in deeper water.'

**JS comment: P1 L23: 'area', not volume...**

Agreed. Text amended.

**JS comment: P1 L26: 'respectively' is missing somewhere**

Agreed. Text amended.

**JS comment: P2L2: The area and number of glaciers refers to the Himalayas, the Hindu Kush, and the Karakoram (not just the Himalayas).**

Agreed. Text amended to:

'Estimates of ice volume range from 2,300 km$^3$ to 7,200 km$^3$ (Frey et al., 2014 and references within) distributed amongst more than 54,000 glaciers across the Hindu Kush Himalaya (HKH) and the Karakoram.'

**JS comment: P2L17: 'more stable' in the eastern Himalayas is incorrect. The greatest rates of surface lowering are observed in eastern Himalayas (Kääb et al., 2015).**

Agreed. Text amended to:

'Recent studies have identified spatial heterogeneity in mass loss across the Himalaya (Kääb et al., 2012; Gardelle et al., 2013; Kääb et al., 2015). Glaciers in the Eastern Nyainqêntanglha, in the eastern Himalaya, are losing mass most quickly (Kääb et al., 2015), as are glaciers in the Spiti Lahaul and Hindu Kush (Kääb et al., 2015). Glaciers in the central Himalaya appear to be more stable (Gardelle et al., 2013). The anomalous balanced, or even slightly positive, glacier mass budget in the Karakoram is well documented (Bolch et al., 2012; Gardelle et al., 2012).'

**JS comment: P3L2-3: 'and thus remains to be tested' is superfluous.**

Agreed. Text removed.

**JS comment: P3L20: clarify are these 40 largest glaciers \*partially\* debris-covered?**

Agreed. Text amended.

**JS comment: P4L4-7. There are observed and modelled ELA data from Wagnon et al., (2013) and Shea et al., (2015), respectively.**

The manuscript has been amended to include the ELA estimates of Wagnon et al. (2013) and Shea et al. (2015) (P4, L7-9).

**JS comment: P4L19: what is 'non-void filled'**

The non-void filled SRTM dataset contains 'no data' in gaps rather than being filled with data from other global elevation datasets, such as the ASTER GDEM. A void filled version (known as the SRTM Plus or SRTM NASA V3) is available, but is filled with ASTER GDEM data- this dataset is multi-temporal and thus cannot be used in DEM differencing. We have not amended the manuscript in response to this question.

**JS comment: P5L10-15: do you do any comparison between ASTER and SETSM DEMs on stable ground.**

Yes: off-glacier differences between SETSM and ASTER DEMs are low (mean -0.16, StDev of 10.42). We therefore consider the ASTER DEMs to be a robust replacement for the missing SETSM data despite their coarser resolution.

**JS comment: P5L10-15. Just to clarify, you take the GLIMS glacier extents, and modify them for 2000 and 2014 extents based on Landsat imagery. And it should be mentioned here that you use the 2000 and 2014 extents to calculate area changes.**

We have amended the text here to clarify our approach for documenting glacier area change, using the suggestion above:

'Glacier outlines were downloaded from the Global Land Ice Measurement form Space (GLIMS) Randolph Glacier Inventory (RGI) Version 5.0 (Liu and Guo, 2014; Bajracharya et al., 2014; Racoviteanu and Bajracharya, 2008) and modified for 2000 and 2014 glacier extents based on Landsat scenes closely coinciding in acquisition with the DEM data. Glacier extents from these two epochs were used to calculate area changes. The 2000 Landsat…'

**JS comment: P7L14. The graph shown in the Supplementary Information could be places in the main text, but the caption needs to be improved as it is not clear what is being presented.**

We do not include this graph in the main text as it would mean a number of additional graphs would need to be produced and included in the manuscript to illustrate the effect of the DEM correction process on the stable ground difference statistics. We consider this to be unnecessary as the products of the correction process are much better demonstrated in other work (Nuth and Kääb, 2011). Table 3 gives a summary of the effects of the correction process which, in our opinion, is adequate evidence of its success. The caption for this supplementary figure has been rewritten to better explain what the graph shows.

**JS comment: P8L1-10: Fix terms in the text: subscripts and italics are missing or inconsistent.**

Done.

**JS comment: Eq.2: italize $n_{diff}$, $n_{tot}$, PS**

Done.

**JS comment: P8L7-8: this sentence is unclear. What value of d was used in this study?**

We now take an alternative approach to quantifying the uncertainty associated with DEM difference data, so the comment above no longer applies.

**JS comment: Eq.3 root symbol needs to be over the whole expression: p SE2 +MED2, and I would suggest using dZstable for mean elevation differences (MED). MED looks a lot like median...**

This comment is no longer applicable, but the reviewer's comment on the presentation of formulae has been considered in the amended manuscript.

**JS comment: Section 3.6: suggest removing completely.**

See response to general comment 1 regarding section 3.6.

**JS comment: P9L25: report lowering rates with negative sign (e.g. -0.80 +/- 0.35) to be consistent with Table 4.**

We now report mass balance estimates and they all have the appropriate sign in text.

**JS comment: P10L1-3: 'Mass loss' should be 'surface lowering' here, and don't rates increase downglacier in all cases (not only lake-terminating glaciers)?**

We are now able to show ablation gradients for lacustrine terminating glaciers which clearly show a linear trend with elevation. Debris covered glaciers have distinctly non-linear ablation gradients.

**JS comment: P10L24: Refer to Figure 5 here.**

Text amended to refer to figure 5:

'Mass loss rates over glaciers on the Tibetan Plateau were higher than those in the Tama and Dudh Koshi catchments (Figure 5) up to 5800 m a.s.l.'

**JS comment: P11L13: 'patterns', not 'scenarios'**

Agreed. Text amended.

**JS comment: P11L13-25: provide rates of area change per year for comparison with other studies?**

We have calculated annual area change rates and made a brief comparison with values given in previous work (P11, L17). Our annual change rate, albeit not for exactly the same group of glaciers in the study area, is very similar to that of Bolch et al. (2008) (0.12% a⁻¹). Our annual area loss values are lower than those of Thakuri et al. (2014) (and references within); this can probably be explained by the type of glaciers each set of work includes in their analysis. Thakuri et al. (2014) document area change over a number of glaciers that are free of debris cover, and therefore readily shrink in response to climatic change, whereas our sample of glaciers is made up of the largest, most debris mantled glaciers in the region (that do not lose glacier area as rapidly).

We refer to other studies of a greater temporal and spatial resolution in the manuscript, to provide more information on glacier area change in the region.

**JS comment: P11L19: This is where the more rigorous classification would be useful. E.g.: Land-terminating clean vs. land-terminating debris-covered?**

Fortunately, all of the land-terminating glaciers in our three main groups (Tama Koshi, Dudh Koshi and Tibetan Plateau) are covered by a substantial amount of debris, so cannot be sub-divided based on debris-cover. The glaciers we highlight as 'TP clean' are all land-terminating and debris free, so provide a direct comparison to the glaciers in the three main groups.

**JS comment: P12L13-20: Suggest moving this section to methods and adding more details on how current and future ELAs are determined.**

Agreed. We have added a short section to the methods to describe how we estimate ELAs from our DEM differencing and how we estimate future ELAs using lapse rates for different catchments (P9, L17). We have kept section 4.2.3 as a short summary of ELA estimates based on surface lowering curves.

**JS comment: P12L16-17: This phrasing is a bit awkward. It seems like you are trying to say that the approximated ELAs give an AAR of 0.37 in the Dudh Kosi catchment. (AAR = Accumulation Area/Total Area).**

We have modified the text to describe AARs in a more conventional way, as suggested.

**JS comment: P13L3: Though around 80**

Text amended to give estimate of the percentage of total annual rainfall delivered by the monsoon in the study area.

**JS comment: P13L18-25: Some skepticism might be warranted when referencing the snow line altitude shifts given by Thakuri et al., (2014): these are based on single-image delineations of transient snowlines, and in the Himalayas these do not remain constant at the end of the summer season.**

Agreed. We have amended the text as follows:

'Thakuri et al. (2014) showed a rapid ascent of the snow-line altitude in the Dudh Koshi between 1962 and 2011 (albeit through documenting transient snowlines from single scenes acquired at each epoch), and Kaspari et al. (2008)…'

**JS comment: P14 Section 5.2 title. 'surface lowering' not 'mass loss'**

As we are now able to provide mass balance estimates we have kept the title of this section the same.

**JS comment: P14L26-27: For lake terminating glaciers its complicated, but for land-terminating glaciers thinning should reduce the driving stresses and lead to decreased glacier velocities (e.g. Berthier and Vincent, 2012; Haritashya et al., 2015)**

As this section of the manuscript focuses on the potential future evolution of lacustrine terminating glaciers we have chosen not to modify it to discuss the dynamic of land-terminating glaciers too.

**JS comment: P15L25: the sensitivity of Dudh Kosi glaciers to future ELA changes based on its hypsometry was noted previously by Shea et al. (2015).**

We have altered the text to include acknowledgement of the hypsometric analysis conducted by Shea et al. (2015):

'The coincidence of maximum surface lowering rates with the altitude of maximum hypsometry in the Dudh Koshi catchment (Figure 5) means a large amount of ice is readily available to sustain mass loss rates here. Sustained and prolonged mass loss may lead to a bi-modal hypsometry here, with the separation of debris covered glacier tongues and their high-elevation accumulation zones a possibility (Rowan et al., 2015; Shea et al., 2015).

**JS comment: Table 3: separate columns for means and standard deviations**

Table modified according to the above suggestion.

**JS comment: Figure 1: Add the imagery extents here.**

The large footprint of the Landsat scenes used in this figure means that only one image is needed to cover all of the glaciers in our sample. The Figure has therefore not been modified.

**JS comment: Figure 2: text labels with glacier names are impossible to read. Also, is it possible to show the data voids in DEM differencing?**

The size of the text labels for the glaciers we highlight has been increased and the labels moved where possible to make them more obvious. Data voids have been set to transparent to maintain the clarity of the figure and to avoid distraction from the surface lowering data. We haven't changed the figure in this regard.

**JS comment: Figures 3 and 4: Why are glacier extents shown in 2014 (left panel) not also present in 2000 extents?**

2000 and 2014/15 extents are both shown on the right hand panel. Only the 2000 extents are shown on the left panel to show the DEM differencing data as clearly as possible.

**JS comment: Figure 5 and 6: Larger fonts required! Caption should point out that surface lowering curves are on the right and hypsometry on the left. Maybe show hypsometry as relative (% of total area) as opposed to absolute? and show surface lowering rates as boxplots by elevation band?**

Font size has been increased and the caption amended as suggested. We have kept the mass balance curves the same to allow for easy comparison with the conceptual mass balance curves of Benn et al. (2012).

**JS comment: Figure 6: Why is approximate ELA only shown on top panel? What about projected future ELAs?**

We have calculated and added future prospective AARs for the clean glacier sample and included them in Figure 7 (see below). As we have normalised the elevation range of glaciers in Figures 5 and 6 the ELAs cannot now be plotted here. We have marked the approximate ELA (where the mean mass balance curve of all the glaciers in the sample first approaches 0 on the x-axis) for all glacier types in our sample to figure 8. We have not calculated projected AARs for lacustrine terminating glaciers as this group contains glaciers from either side of the orographic divide (thus different lapse rates must be considered) and glaciers of contrasting hypsometry.

Review of King et al., The Cryosphere, July 2016

**In their paper, King and co-authors measured glacier surface elevation changes in the Everest area between Feb 2000 and 2014/2015 using remotely-sensed DEMs and studied the spatial pattern of elevation change in the ablation area of glaciers. Rate of surface elevation changes are compared between three different basins and also interpreted considering the glacier type. A special focus is drawn on the influence of proglacial lakes on glacier wastage. Sensitivity of these glaciers to the future projected warming is discussed by examining their hypsometry.**

**This study is not ready for publication. At several places in the manuscript (MS), there are some misconceptions, especially a problematic confusion between rate of elevation changes (dh/dt, what the authors measured) and ablation rates (i.e. surface mass balance). The two variables are different and cannot be compared as the authors do (e.g., in their comparison of their data to Benn's model). Some of the conclusions are not really supported by the data themselves (e.g., statistically significant difference between the 3 main basins? Attribution of the thinning to climate drivers). In the end, the author is also left without a real take-home message. The limited implications of the present study are partly due to the fact that the authors decided not to compute glacier-wide mass balances. This is probably a reasonable choice given the lack of knowledge of SRTM penetration depth in the upper reaches of the Everest area glacier but still it makes the interpretation of the observations very difficult because rate of elevation changes for a portion of the glacier are not equivalent to surface mass balance, they also depend on ice dynamics. In the end, the reader is left with the question: "what did we learn in this study that we did not before?"**

**General comments**
**One major issue is that authors draw some conclusions between glaciers in three different basins or with different terminus type from dh/dt measured in the ablation area only. Such comparisons carry little significance because these generally small differences in dh/dt the ablation areas could easily be compensated by differences of opposite signed in the accumulation areas. Hence one cannot conclude unambiguously that the mass loss is larger for such basin compared to such basin or for this type of glacier terminus. Although the differences are often not statistically strongly different. A comparison of the different rate of elevation changes with altitude (Figure 7) is also partly misleading because the elevation range of the compared glaciers is really different (due to different climate setting). A solution could be for example to normalize the elevation range has was done in (Arendt et al., 2006), among others. All along the text and in the tables, the authors provide many details about individual glaciers such that it is difficult to extract the big picture, the take-home message. A table summarizing mean dh/dt in the ablation area average by large basin and glacier type (area loss / mean dh/dt for the ablation area) should be added. See also the specific comment below where I suggest moving Table 4 and 5 in to the supplement and replace them with synthetic figures.**

Several of the reviewer's comments relate to the fact that we present surface lowering data rather than estimates of glacier mass balance. The reason that we did not initially calculate mass balance is partly because of the lack of a thorough understanding of C-band radar penetration depth into snow, firn and clean ice in the Himalaya and its influence on the quality of the SRTM dataset over glacier accumulation areas. However, the recent work by Kääb et al. (2012, 2015) has considerably refined our knowledge of this problem, and they demonstrate how their corrections reconcile previously divergent estimates of glacier mass balance in the Eastern Himalaya (Gardelle et al., 2013 Vs Kääb et al., 2012). With a small amount of additional processing we have thus corrected our data to account for C-band radar penetration following the method and values presented in Kääb et al. (2015). As our baseline dataset (the SRTM DEM) is the same as Gardelle et al. (2013), this yields the same spatial coverage and we are thus able to directly compare our results.

To allow for direct inter and intra catchment comparison of surface lowering curves (i.e. what has become mass balance data in the revised manuscript), we have followed the approach of Arendt et al. (2006) and normalize these data by the each glaciers elevation range. We thank the reviewer for this suggestion.

We have simplified our discussion and presentation of results to avoid unnecessary mention of individual glaciers, and focused solely on the behavior of glaciers depending on terminus type and the variability of mass loss between catchments. We have amended text in the conclusion to give the key findings greater emphasis. Additionally, as suggested by the reviewer, Tables 4 and 5 are now presented as supplementary information should a reader require information on individual glaciers.

**Errors on dh/dt. One problem with the metric which is used currently is that it does not take into account the size of the averaging area, i.e., the error on the rate of elevation change is the same for a 0.1 km² and a 80 km² ablation area. This is obviously not realistic.**

Thanks for pointing this out. Choosing the most appropriate error metric was something that we deliberated over for some time. In light of the reviewer's comment we have reassessed the uncertainty estimates associated with our elevation change data using a root of sum of squares approach, similar to that of Wang and Kääb (2015) & Melkonian et al. (2013, 2014). We acknowledge that the total error budget should contain assessments of not only the standard error associated with pixel scale differences, but also uncertainty estimates of elevation differences/ volume change averaged over a larger area. We have taken an area-weighted approach to calculate catchment wide error budgets in the updated version of the manuscript.

We also acknowledge a later comment by the reviewer that the contrasting acquisition dates of the WorldView imagery used in SETSM DEM extraction may introduce a seasonal variation in glacier surface heights. See our response to comment 4.27 in regard to this problem.

**The discussion of the climate drivers of this glacier thinning in the ablation area is currently very weak. For example (13.18), the authors make a weak statement about climate trend during 2000-2015, also the period of the dh/dt measurements. Even if T,P were stable (no trend) during the study period, a strong thinning rate could still be observed between 2000- 2015 if, for example, a step-like warming (or change in precipitation) occurred in the years preceding the study period. In other words, the glacier disequilibrium to the climate depend a lot on what happened before the study period and not only on the climate trend during the study period.**

We agree that the response time of these glaciers to climatic change is likely to be greater than the length of our study period. Here we were simply trying to suggest that the contemporary climate records taken at the Pyramid research station and at Dingri on the Northern side of Everest are evidence that glacier mass loss will continue into the future. We have clarified this in the revised manuscript.

We have also ensured that studies such as Yang et al. (2011), who present temperature data for the period 1959-2007 at Dingri, are clearly acknowledged in the revised manuscript, to give the importance of long-term records more prominence.

**Figure 7 and the related text. It is not acceptable to compare dh/dt and mass balance. They are simply not glaciologically comparable. The statement 17.7 that "The ablation gradients shown by lacustrine terminating glaciers are also very similar to regime 3 of Benn et al. (2012)" is a clear illustration of this confusion. Authors seem to believe that they measure ablation gradient when they measured gradient in dh/dt in the ablation area. They entirely neglect the role of emergence velocity which is not physically realistic.**

As detailed above we have now estimated mass balance from our data. These are now directly comparable with the conceptual mass balance curves of Benn et al. (2012).

We have acknowledged (P9, L15) that, without up-to-date glacier surface velocity data and ice thickness measurements, we cannot specifically quantify emergence and its contribution to our surface lowering data. Previous work (Quincey et al., 2009) has identified active vs inactive ice boundaries for a number of the glaciers we include in our analyses so emergence is likely to occur, but we see no obvious evidence in our surface lowering data; unlike Immerzeel et al. (2015), who use DEMs of much higher spatial and temporal resolution. A clear explanation of this caveat will be included in the amended manuscript (also suggested by reviewer Shea).

**More specific comments (some still substantial)**

**Title needs to include "ablation areas"**

Now that we have generated mass balance estimates we have kept the title the same.

**1.17. not all these glaciers are flowing southward (the basins are located southward of the main ridge)**

Text amended.

**1.18. a negative lowering rate suggest a thickening of the glacier (double negative). Either authors should change the sign or used "rate of surface elevation changes".**

We now give glacier mass balance estimates throughout so the comment no longer applies.

**1.19. "small lakes". Are these supraglacial? Proglacial?**

The text has been amended to give more detail on lake type.

**1.24. Providing the present AAR and how it will potentially change in the future due to the rise of the ELA is probably a more useful and conventional metric to illustrate this hypsometric sensitivity of the different basins.**

Text amended here and throughout the manuscript to report AAR in a more conventional manner.

**1.28. I miss a sentence at the end of the abstract indicating the implications of this study. A sort of take-home message for the readers. To answer this question: What did we learn here that we did not before? A statement well-supported by the data that will make other researchers cite the present paper.**

We have added a couple of sentences at the end of the abstract to emphasise the importance of our results:

'Our results are significant because they suggest that documented glacial lake growth and/or expansion across the Himalaya is likely to be accompanied by increased ice mass loss in the near future. Further, the influence of temperature increases may be highly variable across different catchments, complicating the prediction of the future contribution of glacial meltwater to river flow.'

**2.13. "ice melt from the region may contribute 8.7–17.6 mm of sea level rise". Glaciers melt seasonally even if they are in balance and even if they do not contribute to sea level rise.... Replace by "glacier imbalance". Melt is not synonym of mass loss.**

Text amended according to reviewer suggestion.

**2.14. Authors need to stress that these estimates are for the first decade of the 21st century only.**

Text amended.

**2.16. The study by (Kääb et al., 2015) suggest strongly negative mass balance in the southeast Tibetan plateau. Update.**

Text amended to give more detail on the results of Kääb et al. (2015).

**2.18. Kapnick et al. 2015 was a welcome modelling effort to understand the cause of the anomaly, but this is not among the studies that documented the Karakoram anomaly. See rather (Bolch et al., 2012; Gardelle et al., 2012; Hewitt, 2005; Rankl and Braun, 2016).**

Text amended and we now cite several of the studies suggested by the reviewer.

**2.18. Future hydrology. Is the debate relate settled? This need explanation or should be deleted. Because at least in the next decades, more negative glacier mass balance means more water in the rivers...**

We agree that the long-term impact of negative glacier mass balance in the region is uncertain, but it is an implication that should be acknowledged. We have slightly reformatted this part of the introduction so that this matter is not mentioned in the middle of the discussion of mass loss heterogeneity.

**2.26. Description of Benn et al. 2012 conceptual model. Why is this included in the paragraph about measuring glacier mass loss. Separate paragraph needed.**

Now in a paragraph on its own.

**3.6. Already here the reader starts to wonder why only mass loss in the ablation area is observed. This should be better explained/justified right away.**

We now quantify glacier mass balance, thus this sentence has been removed and the comment no longer applies.

**3.14. is it really the majority? I guess in term of area yes but in terms of numbers I am not so sure (there are many small glaciers...)**

Text amended slightly later in the paragraph to emphasise that most glacier area is debris covered.

**3.18. do the authors mean "beneath steep cliffs"? Improve terminology. Khumbu glacier also sit beneath the Everest "massif" and has a wide and flat accumulation area of several km².....**

We are happy with our current description of glacier types in the study area.

**3.23 there are not so many studies measuring acceleration in the rate of surface lowering so authors could probably list them. Nuimura et al. 2012 is the other one I can think of.**

Nuimura et al. (2012) now cited in text.

**4.4 Table 2 in Gardelle et al., 2013 list some ELA values from three different studies. So there is more information about ELA than what the present text suggests.**

Text updated to include ELA estimates of Gardelle et al. (2013), among others suggested by Joseph Shea (other reviewer).

**4.21 if the authors mention the two SAR systems, then they need to tell which one of the two was used to generate the version 3.0 DEM they are using. Readers are confused otherwise.**

Text amended to give more information on the C-band SAR system used by the SRTM.

**4.27. images are listed in Table 1, not Table 2. Further, these images are acquired at very different time of the year which raise the issue of how seasonal variation in height have been accounted for in the study. If not correction was applied, this needs to be well-justified and the uncertainties quantified.**

Text amended to refer to the correct table. Two overlapping SETSM DEMs (ending FA100 and 3C00 in Table 1) have been generated from Worldview imagery acquired before and after the summer monsoon (when glaciers receive most accumulation) of 2014, thus any spatially consistent off-glacier differences may show a remnant snow pack that would cause an elevation bias. The difference between these two SETSM DEMs is slight (mean -0.17 m, $\sigma$ 2.84 m), but we cannot be sure that these differences represent a region-wide average. We incorporate an assessment of the standard error ($\sigma_{season}$) of these seasonal differences into our overall uncertainty budget.

**6.1. Can the authors better justified the need to work on a selection of glaciers and not work on each individual glacier? Rational for that?**

We have selected the glaciers containing the largest volumes of ice and therefore the greatest potential contribution of meltwater. Our ability to include many other smaller glaciers is hindered by the lack of suitable data coverage over steeper, higher topography where many smaller glaciers in the study region are located. We have clarified our rationale in the updated manuscript.

**6.26. Although the spatial variability of the geoid height must be rather small at the scale of the DEMs processed here, it is not acceptable to compare DEMs defined above different datum. There are gridded versions of the EGM96 geoid that can easily be used to correct for the elevation difference. Conversion from geoid to ellipsoid (and vice versa) is also a built-in tool in many GIS software (including in the open source gdal libraries).**

We have now incorporated a geoid correction into our data processing and incorporated the details in the methods section of the paper.

**6.28. "first order trends". More details needed. Are these corrections estimated using all ice free pixels? How do the authors take into account large outliers that always occur in DEMs from satellite stereo imagery and that may contaminate their corrections?**

We mention later in the manuscript that outliers of $>\pm$ 60 m over stable, off-glacier terrain were filtered from the difference data. We have now included this statement in this earlier section. Only ice-free pixels were used to inform on the shifts applied to DEMs. 'First order trends' refer to linear trends fitted through difference data showing clear along or cross track biases. We have amended the text to make this clearer.

**7.7. "penetration corrections are rarely applied". Is this a good justification? Not really. Strongly biased estimates of geodetic mass balances have been published in the past due to the lack of correction of this systematic effect. See for example (Fischer et al., 2015) that demonstrated that the geodetic mass balances from (Paul and Haeberli, 2008) were strongly biased negatively and (Kääb et al., 2015) & (Barundun et al., 2015) that have shown that (Gardelle et al., 2013) Pamir mass balance estimates are likely biased toward positive values for the same reasons. This is a systematic source of errors and as such it cannot be treated by simply adding it to the error bars. The poor knowledge of the SRTM penetration depth is maybe the reason why the authors have limited their analysis to the ablation area. If this is the case, this needs to be explained/justified. But as said in my general comments, this is really limit the implications of the study.**

We have now corrected for SRTM radar penetration following the approach of Kääb et al. (2015). We were reluctant to attempt to correct the SRTM for C-band radar penetration using the estimates of penetration depth given in studies such as Gardelle et al. (2013) as no thorough comparison of the contrast in X Vs C band radar penetration had been carried out. The success of Kääb et al. (2015) in reconciling previously divergent mass balance estimates using a different C-band penetration correction approach means we can now be confident in the correction we have applied.

**7.15. Such an elevation dependent correction cannot be applied to one DEM alone but to the elevation difference between two DEMs.**

Agreed, text amended.

**7.20. unclear what the authors mean by "real topographic change on the stable terrain".**

The section of text describing the calculation of uncertainties associated with mass loss data has now been re-written in the updated manuscript (P7, L16 onwards), therefore this comment no longer applies.

**8.7. what matters is not the spatial autocorrelation in each DEM but the autocorrelation in the map of elevation difference. So only one auto-correlation distance should be reported.**

As the DEM difference grid has a pixel size of 30 m, the autocorrelation distance would be 600 m following Bolch et al. (2011). This has been specified in the updated manuscript.

**8.9. can the authors explain why a MED remain after all the adjustments? I would have expect the mean difference to be 0 "by construction". Did the authors examined the overlapping areas of the WV DEMs as a verification of the DEM adjustment?**

The success of the co-registration process is limited by pixel size of the DEMs involved. For example, Nuth and Kääb (2011) suggest that the co-registration solution has an internal horizontal accuracy of 1/3 of a 30 m ASTER DEM (although often 1/10 of a pixel) so there will be a residual difference that could only be eliminated if the DEMs being corrected were of a finer resolution. Our residual mean differences are all below 1/10 of the pixel size in our DEMs, thus we are confident that our co-registration is optimal.

See our response to comment 4.27 about the comparison of overlapping SETSM DEMs.

**8.11. "independent" of what?**

Comment no longer applies as we now take a different approach to uncertainty estimation.

**8.17. Can the authors confirm that in table 3, the standard error (and not "e", the elevation change uncertainty) is listed. I find it extremely strange that the last column of Table 3 (labelled "st error") is**

**always so close to the value of the remaining mean elevation difference as listed in the "post correction" column of the same table (Table 3). The similarity is unexpected because one column is in m and the other in m/yr. I think authors need to double check this and clarify their terminology.**

As above, we now use an alternative method to calculate uncertainty associated with our difference data. But to clarify the point raised by the reviewer, SE is always similar to the MED when there are a large number of elevation difference measurements and our values were correct.

**8.22. The Landsat images are used to refine the outlines not to extract the hypsometry, as the authors explained earlier in the MS. Be brief here and just tell that the 100-m hypsometry was extracted from the SRTM (?) DEM and the glacier outlines. Void filled DEM or not?**

Text amended according to the above suggestion.

**8.26. "glacier area change". Not relevant in the hypsometry section.**

Agreed. Text deleted.

**9.9. "we did not generate mass balance estimates". Do the authors mean glacier-wide mass balance estimates? The lack of knowledge of the SRTM penetration depth is another good reason to avoid this. Still I find this disappointing, It would have allowed a direct comparison to other studies and better comparison of individual glaciers/basins.**

We now give mass balance estimates following the correction of SRTM data for radar penetration. In doing so, we achieve the same data coverage as Gardelle et al. (2013) and can directly compare our results to this study, and others such as Bolch et al. (2011) and Nuimura et al. (2012).

**9.18. Here I am not sure I understood what the authors exactly did. Do they mean that they only summed mass loss occurring upstream of the 2014/2015 calving front? Why not taking into account at least aerial mass loss (i.e. above the lake level) for the area between the 2000 and 2015 calving front?**

We have modified our approach to incorporate the areal mass loss from between the 2000 and 2014 calving fronts and explain that below water level ice loss cannot been included in the revised manuscript.

**10.1. to draw such a conclusion "The presence of a glacial lake altered the gradient of surface lowering over glacier surfaces" authors need to compute the dh/dt gradient and compare them to support their statement. Is it the gradient with altitude? With distance to the terminus? Statement not demonstrated in the paper.**

We have now calculated ablation gradients (from the ELA to the termini of lake terminating and clean ice glaciers, and from the ELA to the altitude of maximum mass loss for debris covered glaciers) and compare them in the revised manuscript (P10, L29). The ablation gradient of lacustrine terminating and clean ice glaciers is linear from ELA to terminus, whereas the ablation gradient is clearly non-linear for debris covered glaciers, something which has also been identified in previous studies.

**10.23. "mean" over what? A 100-m altitude band centred around 5300 m asl? Clarify.**

Text removed so comment now no longer applies.

**10.6. The mean value of 2.04 m/yr is for which catchment? All merged?**

Comment no longer applies as this part of the text has been re-written.

**11.13. What are these two scenarios? Unclear. Also what is the meaning of "scenario" in this context?**

Text amended and we now describe 'patterns' of ice loss that occurred. Specifically, there we refer to the loss of glacier area around the termini of lake terminating glaciers and clean ice glaciers, and the loss of glacier area as glacier surfaces lowered and narrowed, mostly in the middle portions of debris covered glaciers.

**11.17. Why not providing the same % for lake-terminating glacier.**

This section has been mostly rewritten to give ice area loss totals and as percentages of total glacier area for all groups of glaciers in our study.

**12.1. the basin-wide hypsometries should be added to Figure 7 to be compared easily to dh/dt also averaged by basin. And figure 5-6 would keep only individual glaciers (no basin wide average).**

We have not changed the format of Figures 5, 6 and 8 as the mass balance curves and glacier hysometry curves are easily compared in their current format, especially now that glacier elevation ranges have been normalised.

**12.13. "The altitude at which surface lowering curves approach zero is a good indicator of the ELA of glaciers". This statement is surprising. I checked the Nuth et al., 2007 reference and indeed found the following sentence : "The hypsometric (area–altitude) distribution for Brøggerhalvøya/Oscar II Land is greatest between 250 and 550 m a.s.l., with the 54 year average ELA (position where the elevation change curve approaches zero) at 350 m (Fig. 5a)." So there is no reference or data to support this statement in Nuth et al. This is a strong approximation that suggest similarities between null dh/dt and null mass balance. Rate of elevation change and mass balance are not the same quantities, I do not see how you can do such an hypothesis.**

Now that we are able to show mass balance curves we can use the approach of Nuth et al. (2007) with more confidence, as the point at which the mass balance curves approaches or crosses zero is the point of null mass balance over the study period. We prefer this method of ELA calculation to, for example, the mapping of the maximum snowline altitude at the end of the ablation season as the snowline is transient, and thus a long time series of snow and cloud free imagery would be needed to delineate an accurate, average snowline altitude. The mass balance data also incorporate the mass contribution of avalanches to the glaciers, which is an important influence on glacier mass balance in the study area (Benn and Lehmkuhl, 2000).

**12.16-20. Complicate wording! Do they authors mean that the AAR is 37%, 36% and 40% in the different catchments?**

We have amended the text to summarise AARs in a more conventional manner.

**12.23. The sensitivity of these results to the uncertainties in the ELA need to be quantified.**

Again, we now generate glacier wide mass balance estimates rather than limiting our data to below the ELA, so this comment no longer applies.

**13.4. Regarding sensitivity to temperature (and contrast between different regions), the studies by Fujita and Sakai (Fujita, 2008; Sakai et al., 2015) are better references. (Rupper et al., 2012) is based on very thin data and only examined Bhutanese glaciers so it is not the right reference to claim that the sensitivity is high in Nepal; By the way, high compared to what/where?**

We thank the reviewer for pointing these studies out. We have amended the text to cite these two. Sakai et al. (2015) give a thorough description of the sensitivity of summer accumulation type glaciers to temperature and also give a comparison with the sensitivity of winter accumulation type glaciers to temperature. They conclude that summer accumulation type glaciers are more sensitive to temperature variations than winter accumulation types.

**13.9. In addition to the quoted studies, (Wagnon et al., 2013) have described in detail the precipitation gradient with the Khumbu basin, from Lukla to the Pyramid station.**

We have now cited Wagnon et al. (2013) at this point in the manuscript.

**13.16. This statement is in contradiction to the general belief that glaciers in maritime climate (more humid) are more sensitive to temperature change than glaciers in a more continental climate. See for example (Hock et al., 2009). Without a full sensitivity analysis and without some glacier-wide mass balance measurements, I do not see how the authors can conclude to such statement. Unsupported by the data.**

We agree with the reviewer that glaciers in wetter climates are more sensitive to temperature change, and we were not trying to state the contrary at the point in the manuscript to which the reviewer refers. To avoid such confusion,

we have altered the structure of this section (P13, L14) slightly to separate the description of published temperature and precipitation data and our inferences about their effect on glacier mass loss from the study area.

**13.28. Again (like in 13.18.) a weak reasoning. Why would the rise in the snowline altitude be a proof of accumulation decrease? How can the authors separate this way the respective role of temperature and precipitation trends? (this is even more complex in Nepal than in other mountain ranges because accumulation and ablation season are simultaneous)**

We have amended the text in the updated manuscript (P13, L28) to avoid the direct inference of decreasing accumulation on the southern flank of the Himalaya caused by a rising snowline altitude. The data of Kaspari et al. (2008) allow the more confident suggestion that accumulation has been decreasing on the northern flank of the mountain range.

**13.29. "since the 1970s". Authors need to give the exact time period over which the decrease has been observed (i.e. provide the end year).**

We have added this detail to the text.

**14.3. Again a poor reasoning. A rise in temperature is sufficient to explain a decline is snow cover (and the time period of 9 years is really short to draw conclusions). How can the authors draw conclusions about accumulation rates just based on this proxy?**

This section compiles evidence that temperature is rising and solid precipitation is decreasing in the region; trends that are likely to adversely impact accumulation rates if they continue. We feel this link between climate and accumulation is reasonable to make, and have therefore not altered the text.

**14.10 Authors quote a lengthy time series of DEMs but provide the result for only a five time period... no need for "lengthy" or then authors should provide the results over the long time spam.**

Text amended- deleted the word 'lengthy'.

**14.11. "0.79 m/yr and 0.84 m/yr" can only be compared if error bars are provided. I doubt the authors can conclude here to a significant difference between these two highly similar values.**

Now that we give estimates of mass balance the previous comparison has been removed from the manuscript.

**14.12. Comparison to the thinning rate of (Gardelle et al., 2013). Does this bring something to the discussion? Is it for exactly the same area and the same altitude range?**

We are now able to make a comparison of our mass balance estimates to those of a number of other studies (Bolch et al., 2011; Nuimura et al., 2012; Gardelle et al., 2013; Kääb et al., 2012) who generated mass balance data for different time periods over a similar selection of glaciers and over a time period stretching back to 1970. This new section starts at line 9 on page 14 of the updated manuscript. Due to differences in identifying the sources of data used in each study we are not able to compare the same area and altitudinal range of the same glaciers. Apart from the data published in Gardelle et al. (2013), which the reviewer suggests is biased towards the positive because of their SRTM correction, there appears to have been a steady increase in mass loss rates in the study area since the 1970s. We include this discussion in the updated version of the manuscript.

**14.18. "given" missing I think. The entire sentence needs improvement in fact.**

We have changed the wording of this sentence slightly to improve its clarity.

**15.9-11. Understatement. I do not understand how these statements are related to the rest of the paragraph. What do the authors want to conclude here? Do they want to explain why the dh/dt is not as negative for Imja? Make the logics easier to capture by the reader.**

Now that we have calculated mass balance estimates for Imja and other nearby, land-terminating glaciers, we see that Imja has lost much more ice over the study period. As a result, we do not have sufficient evidence to suggest that ice loss from this glacier is being slowed by the presence of an ice foot in the lake. This section of text has been removed.

**15.18. Can the authors explains what is this "similar surface lowering pattern". It has not been presented in the result section. How can they be certain that this is due to enhanced ablation at cliffs/ponds rather than advection by ice flow of an heterogeneous surface topography?**

This section has been re-written (see P15, L10) to more clearly explain the surface lowering pattern common in areas of stagnant ice (*cf.* Quincey et al. 2009) with well-developed supraglacial pond networks (e.g. Watson et al. 2016). A similar pattern of surface lowering is evident over the long, debris covered tongues of the larger glaciers in the Tama Koshi catchment and on the Tibetan Plateau, and it is not unreasonable to suggest that large parts of these glaciers are now stagnant (indeed we have unpublished data that confirm this).

**16.4. "earlier epochs". Provide year of estimates.**

We provide the years associated with ELA estimates in section 2, so now refer back to this section at this point in the text.

**16.15. Unclear wording. Why not simply mentioning the reduction in the AAR due to the ELA rise (this would be the theoretical reduction of course because this would be based on the present-day hypsometry not considering the future area loss, mainly at low elevations.)**

Again, we have amended the text to refer to ELA rise induced AAR change in a more conventional way.

**16.24. Again a very strange structure for this sentence: change are described for Dudh Koshi and TP glaciers and the sentence finishes with a conclusion for ... Tama Koshi basin. Improve logics.**

We have altered the structure of this part of the manuscript slightly.

**17.21. The statement in the conclusion that there is decreased ice influx from accumulation zone comes from nowhere. Was never discussed earlier in the MS, never shown by the data.**

With hindsight we agree that this statement cannot be shown to be true from our results alone and we thank the reviewer for picking this up. We have removed it from the manuscript.

**17.24. Here and before. How do the authors calculate the uncertainty for their basin-wide average? Must not be simply the mean of the individual glacier uncertainties.**

We have now calculated an area weighted average uncertainty for each catchment and these are quoted in the manuscript where average mass balances for each catchment area described.

**17.27. "We suggest that the across-range contrast in annual precipitation total may have caused greater ice loss on the north flowing glaciers ". Are they different enough statistically (compare 0.80 and 0.95 m/yr) to deserve an explanation? See also my general comment about the weak attribution to climate drivers.**

Given that the mean mass balance estimates that we have now produced for each catchment are not markedly different across the orographic divide, we have toned down what was previously written about potential drivers of the differences in mass loss between Tibetan and Nepalese glaciers.

**18.1. Add "in their ablation area"**

Comment no longer applies now that we have calculated mass balance estimates.

**18.13. Again, same as above (see general comments). Authors did not measure ablation gradients!!! They maybe measure dh/dt gradient (with altitude? distance?). But no plot show these dh/dt gradient data.**

We now do. See comment 10.1.

**Table 2. Authors could draw an horizontal bar to clearly separate the different catchment.**

Table amended according to suggestion.

**Table 4 (like Table 2 and Table 5) are not a really useful way to present the data. If the authors think that the list of glaciers is really important (I am not sure it is) then these tables should me moved as appendix or supplement. A much more concise way to present these numbers (in a figure rather than a table) should be preferred. For example a whisker plot showing the mean/median, range of values etc... for each catchment and each glacier type would condense the info and then, the corresponding text could be shorten also.**

We have moved the three large tables containing information on each individual glacier to the supplementary information. The results section of the manuscript has also been largely rewritten to give a more concise summary of the key statistics associated with glacier groups of different location and terminus type. We have not added any additional figures to summarise these statistics.

**Figure 1: Authors needs to indicate in the caption what is the background image and the source of the inventory.**

Figure caption amended according to suggestion.

**Figure 2: it would be good to show the off glacier dh/dt at least in a figure in the Supplement.**

An additional figure has been added to the supplementary information to show off-glacier difference data. See Figure S2 below.

References cited:

[revised manuscript text omitted]

---

## Referee Report (RR1)

The revised MS is improved in several aspects but unfortunately one of the major issues that I raised in my first review has NOT been corrected. Authors are still believing/pretending that they measured the pattern of mass balance (for example with altitude, see the legend and axis title in Figure 5, 6 and 8) although they only measured the rate of elevation change (dh/dt, see comment 9.17 below). I think many of their conclusions (e.g., role played by glacial lakes in controlling glacier mass loss) would still hold without this basic error but the paper cannot be published in its current form. If the paper is publish like this, there would be a risk that many glaciologists would confuse dh/dt (readily available from DEM comparison) and mass balance in the future and this would be dramatic because these are two very different quantities with profoundly different meaning/interpretation. Only the glacier-wide averages of these two quantities are equal (with a proper density assumption).

Another major issue is the error estimate that is different from the one presented in the submitted MS. Authors have followed a non standard approach (not a problem per se) but not very clearly described/justified. In my view some errors are double-counted which lead to overestimated uncertainties.

My line by line comments are provided below (Page.Line).

Abstract and elsewhere in the paper. Space sometime missing between "w.e." and " a-1"

3.1-5 paragraph not well linked with the rest of the introduction

5.4 Two versions (X and C-Band) of the SRTM DEMs are presented. Authors need to clarify which one of the two was used. See comment from my initial review on this. By the way, why presenting the SRTM X-Band DEM if it is not used at all?

6.26 Authors used EGM2008 for the WV DEM geoid correction. But SRTM C-Band uses a different geoid, i.e. EGM96. It would have been best to use the same geoid. Further, if the DEMs are both registered to the geoid I do not understand why a 30 m systematic elevation difference (see Table 2) remain between them. I would have expected a few meters of bias, no more (orbital WV errors + differences between EGM96 and EGM2008). These unexpected systematic elevation differences make the whole DEM processing suspicious.

7.8 can the authors confirms (and write in the MS) that no penetration correction was applied for debris-covered areas? Was still unclear to me. Authors use an average value for the ablation/accumulation area because these are the one available from Kaab et al. However, they need to state that there is a potentially strong spatial/altitudinal variability in the SRTM penetration depth (depending on firn temperature and water content) and thus that this is source of error when examining the spatial pattern of elevation change.

7.23 Error estimate. Equation (1): First I do not understand why authors do this calculation, taken from Wang & Kaab. They are not really interested in the error of individual DEMs but rather in the error on elevation changes which is directly provided by Sigma_i. Or did I miss something?

Further, following standard error propagation, I would have expected that
$Sigma\_dh^2 = Error\_dem1^2 + Error\_dem2^2$
so the sign does not appear to be OK here. This error was also present in the Wang & Kaab paper unfortunately and authors should avoid propagating it in the literature. See http://ipl.physics.harvard.edu/wp-uploads/2013/03/PS3_Error_Propagation_sp13.pdf for example.

8.1. Write maybe "standard elevation differences over stable terrain ($\sigma_{stable}$)".

8.15 in their $\sigma_{season}$, the authors include the $\sigma_{stable}$ that is the std deviation of the DEM differencing on the stable terrain. But this source of error has already been accounted for earlier in Eq 2? Rather the mean bias on glaciers between two WV2 DEMs acquired a few weeks/months apart could maybe be a better estimate for the (systematic) seasonal error.

9.17-19 "We do not quantify emergence velocity".  Of course, it is very difficult to infer mass balance (MB) from elevation change (velocity, ice thickness data are needed and not available, true). Then, locally, the values that the authors show are not mass balances but elevation changes. Despite this statement, in the rest of the MS, authors ignored totally that they did not observe mass balance. Maybe the authors believe that Himalayan glaciers are special and that in their case MB and dh/dt are equal? If this is so, they are wrong. It has been demonstrated for one of their study glaciers (Khumbu).
Authors can refer to Nuimura et al. 2011 to check that the magnitude of the emergence velocity (even in Himalaya) can match or be even much larger than the value of the surface mass balance. In Nuimura's Table 4, the dh/dt are 0.7 m/a and the emergence velocity needed to retrieve the surface mass balance for some portions of the ablation area of Khumbu Glacier is... 5 to 6 m/yr, i.e. one order of magnitude larger!!! Clearly it illustrates how dh/dt and MB cannot be mixed/confused.
Nuimura, T., Fujita, K., Fukui, K., Asahi, K., Aryal, R. and Ageta, Y.: Temporal changes in elevation of the debris-covered ablation area of Khumbu Glacier in the Nepal Himalaya since 1978, Arctic, Antarctic, and Alpine Research, 43(2), 246–255, 2011.

9.21 Authors did not calculate mass balance but elevation changes! See comment (9.17). Consequently, the altitude where dh/dt approaches 0 is NOT the ELA. If this was true then the ELA would be above most Alaskan glaciers for example (See dh/dt vs altitude curve in Arendt et al., 2006 approaching 0 close to the glacier head, just one example among many studies showing that thinning can indeed affect the entire accumulation zone of some glaciers) but we all know that the AAR of most of these Alaskan glaciers is not 0! They still have an accumulation area. The same would hold in the Alps and many other mountain ranges. Altitude of zero elevation change and ELA have nothing in common.

10.4 "in the" repeated

10.5 Here I want to recall my above comment (9.17) to the authors that their dh/dt can only be interpreted as (glacier-wide) mass balance after averaging over the whole glacier area. This is not true for point, individual altitude band and considering ablation/accumulation areas separately because of the divergence of the ice fluxes. See text books.

10.29 authors did not measure the ablation gradient. They measured the gradient of dh/dt with altitude. See comment 9.17.

11.26 what do the authors mean by "here"?

13.19 I would expect the rise in mean temperature to be between the minimum and the maximum. But maybe I wrong? Maybe authors can double check the reference cited?

14.5 This is a conclusion inherited from the previous version of the paper that does not hold anymore. Or is it for a specific altitude range?

14.17 omit parenthesis

15.7 "m a-1" or "m/a", author need to be consistent in their notations

15.27 what do the authors mean by "here" in this context?

26.5-7 This is already stated in the Method, no need to repeat I believe.

16.28 again, authors did not measure mass balance curves. This whole comparison is then problematic. See comment 9.17

17.33 does it mean that water is stored in the englacial hydrological network? Or in ponds? Maybe authors could clarify what is the "distributed water storage".

17.7 no "s" for glacier margin

Figure2. Authors do not only show "surface lowering" as written in the legend and caption. There are some areas of thickening (!) in their map. Rather they show the "rate of elevation change". Same for Figure 3.

Figure 5. How normalization was done should be described in the method section of the text (maybe with a reference to justify/explain it?). This normalization is really useful to compare the different basins and glacier type. Thanks for following my advice.

Figure 5-6 and 8. Caption and axis title are wrong. These are NOT mass balance curves. But curve of dh/dt

Supplementary. Figure 2. With this color scale one does not see much. Supplementary figure 1 (which by the way is very convincing) suggests that a color scale between -15 to +15 (with a step of 5 m) would be more appropriate.

---

## Referee Report (RR2)

Authors have now well-taken into account most comments from my first and second reviews.

I still fully disagree with the statement that the ELA corresponds to 0 elevation change. This is not a key point/finding of this paper but it is not glaciologically acceptable. The publications by Huss & Farinotti (cited by the authors to back up their assumption) do not agree with this statement. I reproduce below the Figure 1 of their 2012 JGR paper. Dh/dt does not cross the axis! I recommend that the authors use the much more conventional hypothesis that the ELA = median altitude of the glacier (see for example http://www.the-cryosphere.net/9/2135/2015/ or Figure 3 in https://www-cambridge-org.biblioplanets.gate.inist.fr/core/services/aop-cambridge-core/content/view/7A6999E78A3C73FDF340119462BFF313/S0260305500000239a.pdf/div-class-title-the-high-mountain-asia-glacier-contribution-to-sea-level-rise-from-2000-to-2050-div.pdf ).

[Figure]

**Figure 1.** Schematic representation of the altitudinal distribution of surface mass balance $\dot{b}$, hypothetical elevation change $\rho \partial h/\partial t$, and apparent mass balance $\tilde{b}$ over the glacier using linear elevation gradients (approximating $\dot{b} - \rho \cdot \partial h/\partial t$, see equation (1)).

In the event where the authors would decide to keep this wrong statement in their paper (I really hope not and do not see what they would stick to it), authors need to explain how they tackle the glaciers for which the dh/dt curves do not cross the 0 elevation change line (this is the case for some of the glaciers in Figure 5 and will be observed for many glaciers in a retreat state due to simple principles of glacier dynamics). More specifically, how do the authors quantitatively define the "altitude where the curve closes 0"? It is a bit too vague for a scientific paper and lack reproducibility. Do they use a specific minimum absolute value for dh/dt to define where dh/dt become undistinguishable from 0 based on the uncertainty of their measurements?

Further, is this definition of the ELA used to apply different SRTM penetration bias correction for clean ice and snow/firn? This is currently not clearly specified.

P13.25, authors state "Using those ELAs the accumulation area ratio (AAR) (Dyurgerov et al., 2009) can be estimated for each glacier and this is a parameter strongly related to long-term mass balance (König et al., 2014)". Significance of the ELA/AAR for debris covered glaciers is different than for clean ice glaciers. When the proportion of debris cover is large and the accumulation is partly made by avalanches the significance of the AAR and its relationship to mass balance / climate is dramatically

change. See for example discussion this in (Iturrizaga, 2011), P211) for Karakoram glaciers. Authors need to acknowledge this additional complexity.

I will now trust the editor in its ability to check carefully that this last round of comments is well-taken into account by the authors.

Good finalization of the paper.

P8 L19. Eastern

P3 L28. New paragraph before "There"

Some readers may wonder whether the WV DEM are available to others. Were they not restricted to 2015 post-earthquake studies as originally announced on the cryolist if I recall correctly (maybe to mention P5.10 or in the acknwoledgments) ?

P12 L13 fix spacing between words

---

## Author Response (AR2)

We thank both reviewers and the editor for their additional detailed comments on our resubmission to The Cryosphere. We have considered them carefully, and where necessary re-computed some of our data to ensure they are adequately addressed. We have also revised the manuscript to provide further clarification or information where requested.

Both reviewers made similar points about our treatment of surface elevation change data as an indicator of mass balance. We acknowledge that this may be flawed and consequently no longer use surface elevation change profiles to infer mass balance versus altitude relationships. We have been particularly careful to distinguish between the two different datasets throughout the updated version of the manuscript. As we are able to present mass balance estimates for each of the glaciers in our sample (following previous additional data processing and analyses) we believe that the main findings of the paper remain robust.

Here we outline our approach to amending the manuscript in response to other more general comments made by both reviewers and the editor, along with smaller changes recommended.

**Reviewer 1- Joseph Shea**

General comments:

**1. Mass loss calculations: how are missing areas in the SRTM dealt with, particularly in accumulation areas? And why not use the slightly lower and generally more accepted density of 850 +/- 60 kg/m3 (Huss, 2013)?**

We followed a similar approach to that of Ragettli et al. (2016) to fill data gaps in DEM difference data. We have added a much more thorough description of this process to the manuscript (the new section 3.3.3). In summary, median values of surface elevation change data from 100 m elevation bands were used to fill data gaps.

As we adopt the approach of Huss (2013) in assigning an additional 7% to the uncertainty estimate of mass balance loss data, we have adjusted the conversion factor used from 900 kg m$^{-3}$ to 850 kg m$^{-3}$ as the reviewer suggests. Updated mass balance estimates have been substituted into the manuscript and supplementary information, although the differences from this adjustment are small.

**Also: a mass balance can be calculated for the entire glacier/region, but for vertical profiles (e.g. Figure 5 in the revised manuscript) it must be kept as surface elevation differences - the conversion to mass change can only be done if dynamics are taken into account (or they cancel each other out, as in the case for the whole glacier). As Gardelle et al. note in their 2013 paper: "Note, elevation changes over separate sections of a glacier cannot be treated as mass changes due to the disregard of glacier dynamics."**

**As a result, mass loss rates cannot be compared for different elevations (P10L15-27). This will need to be treated as elevation differences. And ablation gradients should be 'surface lowering' or 'elevation change' gradients (P11L1-10).**

We acknowledge that we cannot treat surface elevation change as a measure of surface mass balance throughout the entire elevation range of each glacier and thank both reviewers for their clear explanation of this point. We have followed the above recommendation and now present surface elevation change curves; we have also been careful to make the distinction between these dh/dt curves and our separate mass balance estimates made for whole glaciers. As the contrast in mass balance estimates remains depending on glacier terminus type (lake Vs land terminating), one of the main findings of the work still stands.

**2. Derivation of ELA from mass balance (Sec. 3.7). In the previous version of the mansucript, both reviewers pointed to this as a potentially weak point in the methods. The authors have not provided any additional support for the assumption that the ELA can be determined from geodetic mass balance observations (other than Nuth et al., 2007). I think the method of estimating future ELAs/AARs is useful**

**but the method for current ELA calculation needs to be further justified. Also, as the authors calculate mean geodetic mass balance for 100 m elevation bands to estimate ELA the submergence/emergence velocities again become an issue that needs to be considered.**

Thanks for this assessment – we are happy to provide further justification both here and in the revised submission.

A similar assumption about the similarity of the zero point of dh/dt and the ELA has been made by a number of other studies (e.g. Huss et al., 2008; Farinotti et al., 2009; Huss and Farinotti, 2012). These studies show how, at the ELA, there is little difference between observed (measured in the field) surface mass balance and the change in surface elevation of the same point in the glacier system. We acknowledge that this may not hold with increasing distance from the ELA (see Huss and Farinotti, 2012 their figure 1) due to the influence of ice dynamics, as both reviewers have clearly stated. But we consider the point of zero elevation change to be a reasonable estimate of ELA because here submergence and emergence should be at a minimum. We have reinforced section 3.7 with reference to the studies mentioned above and believe that the method we follow to estimate ELA is robust.

Comparison of the ELAs we have derived using this method with those of other studies (that have used alternative techniques) also reinforces our approach. Our ELA estimate for the Khumbu glacier (6000 m) is identical to that of Rowan et al. (2015) who used a dynamic glacier model to reconstruct glacier evolution since the Little Ice Age. Our ELAs for Imja glacier, Bhote Kosi Glacier and Ngozumpa Glacier (approximately 5600, 5700 and 5600 m, respectively) are also very similar to those of Thakuri et al. (2014) who mapped snow line altitudes (SLA) over these and many other glaciers in our study area for six different time periods between 1962 and 2011. Thakuri et al. (2014) use SLA as an estimate of ELA, and suggest ELAs of 5691, 5632 and 5570 m, respectively for Imja, Bhote Kosi and Ngozumpa glaciers.

**Specific comments:**

**Abstract: just a suggestion to shorten to the recommended 100 - 200 words**

We have shortened the abstract to 263 words.

**P2L23: Central Himalayan glaciers have less negative mass balances (as opposed to more stable).**

Agreed. Text amended.

**P3L24: clarify - do the 40 largest glaciers comprise 70% of the total glacierized area?**

Yes, precisely that (Bajracharya and Mool, 2009). We have changed the text slightly to make this point clearer.

**P3L29: New paragraph for Tama Koshi.**

Agreed. Text amended.

**P4L1: "The Tama Koshi is a poorly studied catchment..."**

Agreed. Text amended.

**P4L3: New paragraph for Pumqu catchment.**

Agreed. Text amended.

**P9L25: By definition, lapse rates are positive. I'd personally keep it negative and use 'vertical temperature gradient'.**

Agreed. Text amended.

**P11L28: Thakuri et al (2014) also examine smaller glaciers, which might help explain the greater rates of area change.**

We have added 'smaller' to this sentence to make this point clearer.

**P12L24: 'elevation change' instead of 'mass loss' (and elsewhere)**

We have changed this incorrect wording throughout the manuscript.

**P12L26: 'here' - be specific again, e.g. 'north of the divide'**

Agreed. Text amended.

**P13L1: Measured precipitation is actually low - suggest 'delivers a large proportion of total annual precipitation'**

Agreed. Text amended.

**P13L25: Shrestha et al., (1999)**

Agreed. Text amended.

**P14L17: (Barundun et al., 2015)**

Agreed. Text amended.

**P14L17: 'Our results'...re-state the regional mass loss estimates here for comparison. A table with the current results and those from previous studies would also be helpful.**

We have altered the text as requested and added a table to compare the regional mass balance estimates in the literature:

Table 3. Mass balance estimates (from geodetic and altimetric studies) for the broader Everest region and comparable sub-regions/ catchments.

| Time period and area | Mass balance estimate (m w.e. a$^{-1}$) | Study |
| --- | --- | --- |
| Dudh Koshi | | |
| 1970-2008 | $-0.32 \pm 0.08$ | Bolch et al. (2011) |
| 1992-2008 | $-0.45 \pm 0.25$ | Nuimura et al. (2012) |
| 2000-2015 | $-0.50 \pm 0.28$ | This study |
| Pumqu (Tibetan Plateau) | | |
| 1974-2006 | $-0.40 \pm 0.27$ | Ye et al. (2015) |
| 2003-2009 | $-0.66 \pm 0.32$ | Neckel et al. (2014) |
| 2000-2015 | $-0.59 \pm 0.27$ | This study |
| Tama Koshi | | |
| 2000-2015 | $-0.40 \pm 0.21$ | This study |
| Everest region | | |
| 1999-2011 | $-0.26 \pm 0.13$ | Gardelle et al. (2013) |

| 2003-2008 | −0.39 ± 0.11 | Kääb et al. (2012) |
|-----------|--------------|---------------------|
| 2000-2015 | −0.52 ± 0.22 | This study |

**P14L27: 'surface lowering' instead of 'mass loss'**

We have changed this incorrect wording throughout the manuscript.

**P15L26: awkward phrasing. Perhaps: '...suggests large glacier mass losses.''**

Agreed. Text amended.

**P16L5-120: Expand on Figure 7! E.g. at high-end temperature projections, AARs go to zero and ELAs are above the head of glaciers in the Tama Kosi. Initial response to temperature change is greater for Dudh Kosi...**

We have added text to explain figure 7 in more detail:

'Should greater temperature increases occur, for example high-end RCP 6.0 warming, AARs could reduce to zero in the Tama Koshi catchment as ELAs rise above glacierised altitudes. Clean ice glacier AAR adjustment could be rapid given more than 1 $^{\circ}$C of warming, with AARs again approaching zero should high-end RCP 8.5 warming occur in the region. The AAR of glaciers in the Dudh Koshi catchment could reduce quickly under RCP 2.6 warming, but their AAR reduction may be less rapid given greater temperature increases, presumably because of the extreme relief of the catchment. The AAR of glaciers on the Tibetan Plateau could become lower than glaciers of the Dudh Koshi catchment once warming approaches 2.5 $^{\circ}$C.'

**P18L3. "Projected warming in the Everest region will lead to increased ELAs and, depending on glacier hyspometry, substantial increases in ablation areas.'**

Agreed. Text amended.

**Figure 1: Glacier names still too small to read.**

We have increased the font size of glacier names again.

**Figure 2: Glacier names too small, and are the grey regions on the glaciers areas with no data?**

We have increased the font size of glacier names again. Yes, grey areas (actually the ASTER GDEM underlay) are areas of no data. We have added to the figure caption to explain this.

**Figure 7: What does each point represent?**

Each point represents a projected AAR given minimum, mean or maximum temperature rise under each RCP scenario. We have added this to the figure caption.

**Reviewer 2**

**The revised MS is improved in several aspects but unfortunately one of the major issues that I raised in my first review has NOT been corrected. Authors are still believing/pretending that they measured the pattern of mass balance (for example with altitude, see the legend and axis title in Figure 5, 6 and 8) although they only measured the rate of elevation change (dh/dt, see comment 9.17 below). I think many of their conclusions (e.g., role played by glacial lakes in controlling glacier mass loss) would still hold without this basic error but the paper cannot be published in its current form. If the paper is publish like this, there would be a risk that many glaciologists would confuse dh/dt (readily available from DEM comparison) and mass balance in the future and this would be dramatic because these are two very different quantities with profoundly different meaning/interpretation. Only the glacier-wide averages of these two quantities are equal (with a proper density assumption).**

**Another major issue is the error estimate that is different from the one presented in the submitted MS. Authors have followed a non standard approach (not a problem per se) but not very clearly described/justified. In my view some errors are double-counted which lead to overestimated uncertainties.**

We respond to line by line comments relating to these two issues below.

**My line by line comments are provided below (Page.Line).**

**Abstract and elsewhere in the paper. Space sometime missing between "w.e." and " a-1"**

This mistake has been corrected throughout the manuscript.

**3.1-5 paragraph not well linked with the rest of the introduction**

We have added a sentence to try and link the description of the conceptual model of Benn et al. (2012) and the previous statement about the findings in the study by Gardelle et al. (2013).

**5.4 Two versions (X and C-Band) of the SRTM DEMs are presented. Authors need to clarify which one of the two was used. See comment from my initial review on this. By the way, why presenting the SRTM X-Band DEM if it is not used at all?**

We have removed any mention of the X-band SRTM data as we did not use it in this study. We have explained as clearly as possible that we use only SRTM C-band data.

**6.26 Authors used EGM2008 for the WV DEM geoid correction. But SRTM C-Band uses a different geoid, i.e. EGM96. It would have been best to use the same geoid. Further, if the DEMs are both registered to the geoid I do not understand why a 30 m systematic elevation difference (see Table 2) remain between them. I would have expected a few meters of bias, no more (orbital WV errors + differences between EGM96 and EGM2008). These unexpected systematic elevation differences make the whole DEM processing suspicious.**

The figures presented in the most recent version of the manuscript were incorrect and included by mistake. The correct, pre-registration DEM difference statistics have now been included in Table 2. We apologise for this oversight.

**7.8 can the authors confirms (and write in the MS) that no penetration correction was applied for debris-covered areas? Was still unclear to me. Authors use an average value for the ablation/accumulation area because these are the one available from Kaab et al. However, they need to state that there is a potentially**

**strong spatial/altitudinal variability in the SRTM penetration depth (depending on firn temperature and water content) and thus that this is source of error when examining the spatial pattern of elevation change.**

We did not apply any penetration correction to debris covered areas given the uncertainty expressed by Kääb et al. (2012) about the influence of greater than average snowpack depth at the point of ICESat acquisition and the properties of the snowpack at the point of SRTM data acquisition on their penetration estimate. We have updated section 3.3.2 to emphasise that the SRTM penetration depth is likely to be spatially variable and clearly state that we have not applied a penetration correction over debris covered glacier areas as suggested by the reviewer.

**7.23 Error estimate. Equation (1): First I do not understand why authors do this calculation, taken from Wang & Kaab. They are not really interested in the error of individual DEMs but rather in the error on elevation changes which is directly provided by Sigma_i. Or did I miss something?**

**Further, following standard error propagation, I would have expected that Sigma_dh²=Error_dem1²+Error_dem2² so the sign does not appear to be OK here. This error was also present in the Wang & Kaab paper unfortunately and authors should avoid propagating it in the literature. See [http://ipl.physics.harvard.edu/wp-uploads/2013/03/PS3_Error_Propagation_sp13.pdf](http://ipl.physics.harvard.edu/wp-uploads/2013/03/PS3_Error_Propagation_sp13.pdf) for example.**

We have reverted back to a slightly modified version of our original method of calculating uncertainty associated with elevation changes. We now follow the approach originally outlined by Gardelle et al. (2013) that calculates the standard error of elevation changes over different elevation bands. This is a more standardised approach that has been recommended by the editor. In regard to the reviewer comment above, we have checked the sign on our RSS calculations (explained in more detail below) to ensure error propagation is accounted for correctly. We thank the reviewer for guidance on this issue. Our uncertainty estimates are lower than those produced by the method adopted in our resubmission – which was a concern raised by reviewer 2.

**8.1. Write maybe "standard elevation differences over stable terrain ($\Box$stable)".**

We have amended the text following the advice above.

**8.15 in their $\Box$season, the authors include the $\Box$stable that is the std deviation of the DEM differencing on the stable terrain. But this source of error has already been accounted for earlier in Eq 2? Rather the mean bias on glaciers between two WV2 DEMs acquired a few weeks/months apart could maybe be a better estimate for the (systematic) seasonal error.**

We have incorporated the mean elevation difference (+0.69 m) over glaciers (G1 and Bamolelingjia) covered in the overlapping segments of SETSM DEMs from different time periods into our uncertainty budget to try and quantify any seasonal variability in elevation, in accordance with the reviewers suggestion. This can only be considered a rough approximation of this source of error as the depth of any remnant snowpack in the later DEMs is likely to have been highly variable across the study area.

**9.17-19 "We do not quantify emergence velocity". Of course, it is very difficult to infer mass balance (MB) from elevation change (velocity, ice thickness data are needed and not available, true). Then, locally, the values that the authors show are not mass balances but elevation changes. Despite this statement, in the rest of the MS, authors ignored totally that they did not observe mass balance. Maybe the authors believe that Himalayan glaciers are special and that in their case MB and dh/dt are equal? If this is so, they are wrong. It has been demonstrated for one of their study glaciers (Khumbu).**

**Authors can refer to Nuimura et al. 2011 to check that the magnitude of the emergence velocity (even in Himalaya) can match or be even much larger than the value of the surface mass balance. In Nuimura's Table 4, the dh/dt are 0.7 m/a and the emergence velocity needed to retrieve the surface mass balance for some portions of the ablation area of Khumbu Glacier is... 5 to 6 m/yr, i.e. one order of magnitude larger!!! Clearly it illustrates how dh/dt and MB cannot be mixed/confused.**

**Nuimura, T., Fujita, K., Fukui, K., Asahi, K., Aryal, R. and Ageta, Y.: Temporal changes in elevation of the debris-covered ablation area of Khumbu Glacier in the Nepal Himalaya since 1978, Arctic, Antarctic, and Alpine Research, 43(2), 246–255, 2011.**

Thanks for the clear description of the issue with our confusion of elevation changes and mass balance. It is now clear that we cannot present one as the other, and we have adopted our previous approach of showing surface elevation change curves in the manuscript. As we have carried out additional data processing (SRTM correction, gap filling) and are able to generate mass balance estimates for all of the glaciers in our sample, we can still make a comparison between the mass balances of glaciers of different terminus type, thus one of the main messages of the paper remains robust.

We have also chosen to remove the section of the manuscript where we made a comparison between surface lowering curves and the conceptual mass balance curves presented in Benn et al. (2012). The reviewer has very clearly explained how these two datasets differ.

**9.21 Authors did not calculate mass balance but elevation changes! See comment (9.17). Consequently, the altitude where dh/dt approaches 0 is NOT the ELA. If this was true then the ELA would be above most Alaskan glaciers for example (See dh/dt vs altitude curve in Arendt et al., 2006 approaching 0 close to the glacier head, just one example among many studies showing that thinning can indeed affect the entire accumulation zone of some glaciers) but we all know that the AAR of most of these Alaskan glaciers is not 0! They still have an accumulation area. The same would hold in the Alps and many other mountain ranges. Altitude of zero elevation change and ELA have nothing in common.**

We disagree with the reviewer on this point. Huss and Farinotti (2012) have shown (following Huss et al. 2008 and Farinotti et al. 2009) that for a number of glaciers in the European Alps the altitude of zero elevation change (derived through DEM differencing) may be a useful indicator of the point of neutral measured surface mass balance. Around the ELA, ice flux divergence, or what Farinotti et al. (2009) calculate as 'apparent mass balance', should be minimal (as has been shown in field measurements – Cherkasov and Ahmetova, 1996) and have little influence on surface elevation change. Therefore, we would suggest that the identification of an area of zero elevation change at a glaciers surface (obviously disregarding the area around a debris covered glaciers terminus) is a reliable proxy for the position of the ELA. The impact of irregular avalanche input is hard to quantify and could influence surface elevation changes, but over our 15 year study period it may be considered negligible.

Comparison of the ELAs we have derived using this method with those of other studies (that have used alternative techniques) also reinforces our approach. Our ELA estimate for the Khumbu glacier (6000 m) is identical to that of Rowan et al. (2015) who used a dynamic glacier model to reconstruct glacier evolution since the Little Ice Age. Our ELAs for Imja glacier, Bhote Kosi Glacier and Ngozumpa Glacier (approximately 5600, 5700 and 5600 m, respectively) are also very similar to those of Thakuri et al. (2014) who mapped snow line altitudes (SLA) over these and many other glaciers in our study area for six different time periods between 1962 and 2011. Thakuri et al. (2014) use SLA as an estimate of ELA, and suggest ELAs of 5691, 5632 and 5570 m, respectively for Imja, Bhote Kosi and Ngozumpa glaciers.

**10.4 "in the" repeated**

Text deleted.

**10.5 Here I want to recall my above comment (9.17) to the authors that their dh/dt can only be interpreted as (glacier-wide) mass balance after averaging over the whole glacier area. This is not true for point, individual altitude band and considering ablation/accumulation areas separately because of the divergence of the ice fluxes. See text books.**

We have updated this section of the manuscript so that we now refer to the mass balance estimates (calculated for whole glaciers, not with elevation) that we have generated and the surface lowering curves in a much more distinct manner, so as to avoid the incorrect inference that the two are similar as the reviewer has pointed out.

**10.29 authors did not measure the ablation gradient. They measured the gradient of dh/dt with altitude. See comment 9.17.**

We have updated our measurements of elevation change gradients and again reworded this portion of text to emphasise that we show the gradient of dh/dt and not mass balance. We still include these gradients because they show clearly the contrast in the pattern of surface lowering that has occurred on land Vs lacustrine terminating glaciers.

**12.26 what do the authors mean by "here"?**

To the north of the orographic divide. We have altered the text slightly to make this clearer.

**13.19 I would expect the rise in mean temperature to be between the minimum and the maximum. But maybe I wrong? Maybe authors can double check the reference cited?**

After checking Salerno et al. (2015) we have amended the text and updated the minimum temperature increase figure taken from this study.

**14.5 This is a conclusion inherited from the previous version of the paper that does not hold anymore. Or is it for a specific altitude range?**

We believe that our suggestion of enhanced ice loss from glaciers on the northern slope of the Himalayas is still correct. These glaciers show elevated maximum surface lowering rates and surface lowering through a much broader elevation range when compared to glaciers flowing south. The available meteorological data from both sides of the main orographic divide also clearly shows a north-south contrast in annual precipitation amount, and Owen et al. (2009) have suggested that such a contrast may have substantially influenced fluctuations in glacier extent during the Late Quaternary. We would suggest that this contrast exerts a similar control on present day glacier behaviour.

As requested by the editor, we have added to this section of the manuscript to point out that there are other factors (debris cover extent and thickness evolution, monsoon strength) that could have influenced recent glacier mass loss and also mass loss into the near future.

**14.17 omit parenthesis**

Agreed. Text amended.

**15.7 "m a-1" or "m/a", author need to be consistent in their notations**

We have checked carefully to make sure m a$^{-1}$ has been used throughout the manuscript.

**15.27 what do the authors mean by "here" in this context?**

We are referring to the Dudh Koshi catchment. We have clarified that in the manuscript.

**26.5-7 This is already stated in the Method, no need to repeat I believe.**

Agreed. We have removed this unnecessary portion of text from the manuscript.

**16.28 again, authors did not measure mass balance curves. This whole comparison is then problematic. See comment 9.17**

As we have explained in response to earlier comments, we have now removed the comparison with the conceptual model of Benn et al. (2012) so this point no longer applies.

**17.33 does it mean that water is stored in the englacial hydrological network? Or in ponds? Maybe authors could clarify what is the "distributed water storage".**

As we no longer compare our surface lowering data with the conceptual model of Benn et al. (2012) this portion of text has been removed and the comment no longer applies.

**17.7 no "s" for glacier margin**

Agreed. Text amended.

**Figure2. Authors do not only show "surface lowering" as written in the legend and caption. There are some areas of thickening (!) in their map. Rather they show the "rate of elevation change". Same for Figure 3.**

We have altered figure captions in accordance with this suggestion. Thanks for pointing this out.

**Figure 5. How normalization was done should be described in the method section of the text (maybe with a reference to justify/explain it?). This normalization is really useful to compare the different basins and glacier type. Thanks for following my advice.**

Agreed. We have updates the methods section to include a description of the normalisation process of Arendt et al. (2006). This amendment was an excellent suggestion.

**Figure 5-6 and 8. Caption and axis title are wrong. These are NOT mass balance curves. But curve of dh/dt**

We have updated the figure captions now that we show surface lowering curves again.

**Supplementary. Figure 2. With this color scale one does not see much. Supplementary figure 1 (which by the way is very convincing) suggests that a color scale between -15 to +15 (with a step of 5 m) would be more appropriate.**

We have updated the figure as requested.

**Editor- Tobias Bolch**

**I have received the reviews and studied your revision by myself carefully. Although both reviewers differ with the overall recommendation they both point out one major shortcoming which was not addressed: Using a geodetic approach, glacier mass balance can only be calculated for an entire glacier or region. However you used the SRTM DEM which has several data voids. Hence, it needs to be clearly written how you dealt with this problem. You need also provide more information about how you tackled outliers. If these issues are not treated adequately I cannot accept this paper.**

**In addition, the other reviewers comments I ask you to consider the following:**

**- Shorten the Abstract: A good word count is 250 words.**

We have shortened the abstract to 263 words.

**- P2, L5: Wrong statement. What is about Alaska? Lease also consider that the % of the volume of the glaciers is much lower than the area.**

We have removed the incorrect statement from the manuscript. The introduction now begins with a slightly altered version of what was the second sentence.

**- L. 22 Eastern Nyainqentanglha is not considred as being part of the Himalaya**

We have removed mention of Eastern Nyainqêntanglha from the manuscript.

**- L. 30: Although the sample is quite limited you may think about considering Basnet et al. 2013.**

We have now cited the work of Basnet et al. (2013) in the introduction of the manuscript.

**- P. 3, L 17: Hambrey et al. 2008 does only study Khumbu Glacier to my knowledge. Please check.**

Hambrey et al. (2008) studied the Khumbu, Imja, Lhotse and Chukhung glaciers. These glaciers are good representations of land terminating and lacustrine terminating glaciers in the region, and also of glaciers with and without decoupled margins. There are few other studies that have assessed the glacial geomorphology of the region in such detail and we believe it is the best reference to cite here.

**- P. 4, L. 5: Is it really only terminus recession? Please check.**

Che et al. (2014) do not describe any particular pattern or style of glacier area loss so we have changed the wording of this sentence.

**- L. 13: This number is difficult to interpret without knowing the extent of "Everest region". Please also consider other publications on glacial lakes in this region, e.g. those by Franco Salerno.**

We have amended the text to better describe the spatial extent of studies that have mapped glacial water body extent in the area. We have also made mention of the studies of Salerno et al. (2012) and Watson et al. (2016).

**- L. 21/23: supraglacial ≠proglacial…**

We have rewritten this short section as the distinction between glaciers with proglacial and supraglacial lakes was previously quite poor. Thanks for pointing this out.

**- P. 8, L. 5: Should be 2011.**

Text amended.

**- Section 4.2: What is about the uncertainty in area and area changes?**

We have now followed the approach of Ye et al. (2006) to quantify the uncertainty associated with our total area change estimates. We have added area-weighted uncertainty estimates to the manuscript where necessary and updated Table 3 in the supplementary information.

**- Sections 4.2.2 and 2.3 are rather short. I suggest to combine. A very interesting addition would be to add the hypsometry of the debris-covered vs. the debris-free glaciers.**

Agreed and we have combined these two sections. We have also added more detail on the hypsometry of clean ice glaciers and made direct comparison with debris covered glaciers in our sample, although we haven't changed any of the figures including hypsometric curves (figures 5 and 6). We have also updated the manuscript to include ELA and AAR estimates for these clean glaciers.

**- P. 14 L. 5ff: Rather speculative. What is about the influence of the overall topography and debris-cover?**

We have added text to this section to point out that there are a number of other factors that could substantially influence the amount of ice mass loss that occurs in this region in the future, including the effect of evolving debris cover extent and thickness and the strength of the summer monsoon. Similarly, we have added text to state that the influence of ice flux divergence has not been considered in this study, to avoid convincing the reader that submergence or emergence velocities could be disregarded.

**- Section 5.2: What is about Gardelle et al. 2013, Ye et al. 2015, Bolch et al. 2011 (latest period in this study)?**

We already provided a summary and comparison between our results and those of Bolch et al. (2011) and Gardelle et al. (2013), but we have added a comparison with the results of Ye et al. (2015) and also Nickel et al. (2014) to give a more thorough discussion of the rate of glacier ice mass loss on the part of the Tibetan Plateau that our data covers. We have also added a Table (3) to summarise geodetic mass balance estimates for our study area, as requested by reviewer 1 (Joe Shea).

**- Section 5.3: You may consider Thakuri et al. 2016, Ann. Glaciol.**

Thanks for pointing this recently published paper out. We have incorporated the findings of Thakuri et al. (2016) into this section.

**- Section 5.4: L. 11 There are also other studies who measured the velocity. Include at least one more (not necessarily mine) which would make the statement stronger. You may also be interested to read Ragettli et al. 2016 TCD. It is now accepted for TC.**

We have added citations of the work by Scherler et al. (2008, 2011) and Luckman et al. (2007) to reinforce the statement about glacier stagnation in the Everest region.

**- P. 16, ~L. 20. A word about the estimated distribution of the volume possible?**

The paucity of ice thickness data available for glaciers across the region (other than the Khumbu) could make any estimate of the ablation zone' volume' expansion quite inaccurate, so we haven't attempted to do so.

**References cited:**

[revised manuscript text omitted]

---

## Author Response (AR3)

We would like to thank both reviewers and the editor for their reviews of our resubmission to the Cryosphere. We are glad that our amendments to the manuscript have been acceptable and the paper is now undoubtedly a stronger piece of work as a result of attentive reviews.

5 To remedy the remaining issue noted by both reviewers, we have followed the recommended modification of our approach to estimate ELA using the median altitude of each glacier in our sample. Recalculated glacier AARs and prospective future glacier AARs have been incorporated into our results, and the manuscript altered accordingly.

10 The few other minor changes have been made and are described below, and a tracked changes version of the manuscript can be found at the end of this document.

**Reviewer 1- Joseph Shea**

General comments:

15 This is a review of the revised manuscript submitted by King et al (version 5). Below I provide some general and specific comments on both the revised manuscript and the response to reviewers.

General comments:

1) The authors have responded to most of the comments from reviewers 1 and 2. The biggest reviewer concerns about the confusion between mass balance and surface lowering (dH/dT) appears to have 20 been resolved. Figures 5 and 6 show surface lowering rates versus elevation, while overall basin and glacier changes are given as mass change (m w.e./yr). Uncertainty in dH/dT and area change estimates, a large concern from Reviewer 2, have been addressed in section 3.3.

2) However, the authors have disagreed with both reviewers and continue to assert that the elevation where dH/dT = 0 can be used to approximate the equilibrium line altitude (ELA). In their response the 25 authors argue that the ELA should coincide more or less with the elevation where emergence and submergence are equal - but this is not equivalent to the elevation where surface elevation changes are zero! As this disagreement cannot (and should not) be resolved within the current manuscript, perhaps the authors could use the median glacier elevation of the individual glaciers as a proxy of the ELA (e.g. Braithwaite and Raper, 2010), and recalculate their mass changes. If they wanted to compare 30 the median elevation with the elevation where dH/dT = 0 that could be a very useful additional analysis.

**We have followed the suggestion of reviewer 1 and recalculated glacier ELAs, AARs and potential future AARs using the approach of Braithwate and Raper (2010) and later Zhao et al. (2016) where median glacier altitude is used as a proxy for ELA. We have included a table later in this response 35 letter to compare median altitude ELAs with those we previously estimated from surface lowering curves, but have not included this comparison in the manuscript.**

Specific comments:

Response-P2: Snowlines reported in Thakuri et al. (2014) should be treated cautiously as they are snapshot only. Transient snowline elevations are (a) difficult to determine due to cloud cover and (b) 40 can vary dramatically at the end of the monsoon season. They do not likely reflect the true glaciological ELA (as noted in Sec. 5.1), and should probably not be used to justify your ELA proxy.

**We have removed this section of the manuscript as we no longer need to justify our use of zero surface lowering as the glacier ELA. We have also removed a later mention of the ELA results of Gardelle et al. (2013) as they used a similar technique as Thakuri et al. (2014) but for just one epoch, thus their ELA estimates may be unreliable.**

5    Manuscript-P1L21. "Our results suggest that glacial lake expansion..."

**Text altered according to the above suggestion.**

Figure 1: Font sizes are generally still too small to read clearly...

**We have not altered this figure further as increased glacier label sizes would block surface lowering illustrations.**

10   References: Cherkasov et al. reference should include the third author (Hastenrath)

**Reference updated.**

References:

Braithwaite, R.J. and Raper, S.C.B., 2010. Estimating equilibrium-line altitude (ELA) from glacier inventory data. Annals of Glaciology, 50(53), pp.127-132.

Reviewer 2

5   Authors have now well-taken into account most comments from my first and second reviews. I still
    fully disagree with the statement that the ELA corresponds to 0 elevation change. This is not a key
    point/finding of this paper but it is not glaciologically acceptable. The publications by Huss & Farinotti
    (cited by the authors to back up their assumption) do not agree with this statement. I reproduce below
    the Figure 1 of their 2012 JGR paper. Dh/dt does not cross the axis! I recommend that the authors use
10  the much more conventional hypothesis that the ELA = median altitude of the glacier (see for example
    http://www.the-cryosphere.net/9/2135/2015/    or    Figure    3    in    https://wwwcambridge-
    org.biblioplanets.gate.inist.fr/core/services/aop-
    cambridgecore/content/view/7A6999E78A3C73FDF340119462BFF313/S0260305500000239a.pdf/di
    v-classtitle-the-high-mountain-asia-glacier-contribution-to-sea-level-rise-from-2000-to-2050-div.pdf
15  ).

    **We have followed the reviewers above suggestion and recalculated glacier AARs and projected
    future AARs using a median altitude = ELA approach. We have subsequently amended Figure 7
    (below) and altered section 5.5.1 where the ELA estimates are discussed.**

[Figure]

    In the event where the authors would decide to keep this wrong statement in their paper (I really
    hope not and do not see what they would stick to it), authors need to explain how they tackle the
    glaciers for which the dh/dt curves do not cross the 0 elevation change line (this is the case for some
25  of the glaciers in Figure 5 and will be observed for many glaciers in a retreat state due to simple
    principles of glacier dynamics). More specifically, how do the authors quantitatively define the

"altitude where the curve closes 0"? It is a bit too vague for a scientific paper and lack reproducibility. Do they use a specific minimum absolute value for dh/dt to define where dh/dt become undistinguishable from 0 based on the uncertainty of their measurements?

Further, is this definition of the ELA used to apply different SRTM penetration bias correction for clean ice and snow/firn? This is currently not clearly specified.

**As we have now taken an alternative approach to estimating glacier ELA the above comments are no longer applicable.**

P13.25, authors state "Using those ELAs the accumulation area ratio (AAR) (Dyurgerov et al., 2009) can be estimated for each glacier and this is a parameter strongly related to long-term mass balance (König et al., 2014)". Significance of the ELA/AAR for debris covered glaciers is different than for clean ice glaciers. When the proportion of debris cover is large and the accumulation is partly made by avalanches the significance of the AAR and its relationship to mass balance / climate is dramatically change. See for example discussion this in (Iturrizaga, 2011), P211) for Karakoram glaciers. Authors need to acknowledge this additional complexity.

**We have now acknowledged the differing impacts of accumulation area contraction between typical alpine glaciers and those that are fed by avalanches in the Himalaya. We have cited the suggested reference and emphasised that, without data on avalanching rates for high-mountain glaciers such as those in our study, the impact of AAR reduction is slightly uncertain.**

I will now trust the editor in its ability to check carefully that this last round of comments is well taken into account by the authors.

Good finalization of the paper.

P8 L19. Eastern

**Text amended as suggested.**

P3 L28. New paragraph before "There"

**Text altered as suggested above.**

Some readers may wonder whether the WV DEM are available to others. Were they not restricted to 2015 post-earthquake studies as originally announced on the cryolist if I recall correctly (maybe to mention P5.10 or in the acknowledgments)?

**We have added a link to the SETSM DEM data download page in the acknowledgements section. On this web page it clearly states that the DEMs can be freely distributed.**

P12 L13 fix spacing between words

**The spacing between words on this page has been checked.**

Editor- Tobias Bolch

Dear authors,

The manuscript has significantly improved and there are only few issues left. Most important: Both
5 reviewers experessed their concern to use the elevation where dH/dT = 0 as an approximate for the
equilibrium line altitude (ELA) and suggest to use the median elevation instead. A very useful exercise
would be (but it is not a must) to compare the meadian elevation with the elevation where dH/dT = 0
as suggested by one reviewer. Please also address the few other comments by the reviewers in your
resubmission, provide a point to point reply and indicate the changes in the revised manuscript.

10 **We have adopted the alternative approach to estimate glacier ELA as suggested by both reviewers
and updated our glacier AAR and potential future AAR estimates accordingly. Figure 7 has also been
amended along with section 5.5.1 of the manuscript where these results are discussed. The table
below provides a comparison of the ELAs we had previously estimated and those derived using
median glacier altitude. We have not added this comparison to the manuscript to avoid any
15 confusion regarding which method we followed. Generally, median glacier altitude is much higher
than the altitude of zero surface lowering. This difference explains the lower AARs we have now
estimated using this new approach.**

Table below: A comparison of the median altitude of each glacier and the approximate altitude of zero
20 lowering used in previous version of the manuscript as an estimate of ELA.

| Glacier | Min alt. | Max alt. | Median alt. | Zero surface lowering | Catchment |
|---|---|---|---|---|---|
| Bamolelingja | 5013 | 6745 | 5879 | 5700 | TK |
| G1 | 4778 | 7045 | 5911 | 5800 | TK |
| Yanong | 4984 | 6377 | 5680 | 5500 | TK |
| Yanong North | 5025 | 6524 | 5774 | 5500 | TK |
| Erbu | 5020 | 7130 | 6075 | 5800 | TK |
| Drogpa Nagtsang | 5018 | 7031 | 6024 | 5800 | TK |
| Trakarding | 4561 | 6659 | 5610 | 5800 | TK |
| Ripimo Shar | 4600 | 6683 | 5641 | 5600 | TK |
| Shalong | 5301 | 6835 | 5933 | 5200 | TK |
| *Mean* | | | *5867* | *5633* | |
| Ngozumpa | 4686 | 8176 | 6431 | 5600 | DK |
| Imja | 5021 | 7998 | 6509 | 5600 | DK |
| Khumbu | 4915 | 8062 | 6456 | 6000 | DK |
| Lumbsamba | 4936 | 7258 | 6097 | 5500 | DK |
| Lhotse | 4821 | 6082 | 5451 | - | DK |
| Melung | 5271 | 6028 | 5649 | - | DK |
| Bhote Kosi | 4793 | 6679 | 5736 | 5700 | DK |

| | | | | | |
|---|---|---|---|---|---|
| Hungu | 5207 | 6942 | 6074 | 6000 | DK |
| Marala | 5366 | 5920 | 5643 | 5800 | DK |
| *Mean* | | | *6015* | *5742* | |
| Jiuda | 5405 | 7801 | 6603 | 6400 | TP |
| Gyachung | 5309 | 7853 | 6581 | 6400 | TP |
| Rongbuk East | 5640 | 8361 | 7000 | 6500 | TP |
| Rongbuk | 5153 | 8258 | 6705 | 6200 | TP |
| Ayi | 5313 | 6863 | 6088 | 5800 | TP |
| Tibet 1 | 5138 | 7085 | 6111 | 6100 | TP |
| Gyabrag | 5095 | 8182 | 6638 | 6000 | TP |
| Longmojian | 5348 | 6788 | 6068 | 5800 | TP |
| Duiya | 5480 | 7201 | 6340 | 6200 | TP |
| *Mean* | | | *6459* | *6155* | |
| Duosangpuxi | 5561 | 6992 | 6276 | 6300 | TP clean |
| Siguang | 5652 | 6866 | 6259 | 6200 | TP clean |
| Duosangpudong | 5502 | 6925 | 6213 | 6300 | TP clean |
| G08 | 5726 | 6475 | 6100 | 6200 | TP clean |
| G06 | 5545 | 6926 | 6235 | 5900 | TP clean |
| *Mean* | | | *6216* | *6180* | |

Do not hesitate to ask in case you have any questions.

Best regards,

Tobias

References cited:

[revised manuscript text omitted]